# Private Federated Learning Without a Trusted Server: Optimal Algorithms for Convex Losses

**Andrew Lowy**
Department of Mathematics
University of Southern California
Los Angeles, CA 90089, USA
`lowya@usc.edu`

**Meisam Razaviyayn**
Department of Industrial & Systems Engineering
University of Southern California
Los Angeles, CA 90089, USA
`razaviya@usc.edu`

## Abstract

This paper studies federated learning (FL)—especially cross-silo FL—with data from people who do not trust the server or other silos. In this setting, each silo (e.g. hospital) has data from different people (e.g. patients) and must maintain the privacy of each person's data (e.g. medical record), even if the server or other silos act as adversarial eavesdroppers. This requirement motivates the study of *Inter-Silo Record-Level Differential Privacy* (ISRL-DP), which requires silo $i$'s *communications* to satisfy record/item-level differential privacy (DP). ISRL-DP ensures that the data of each person (e.g. patient) in silo $i$ (e.g. hospital $i$) cannot be leaked. ISRL-DP is different from well-studied privacy notions. Central and user-level DP assume that people trust the server/other silos. On the other end of the spectrum, local DP assumes that people do not trust anyone at all (even their own silo). Sitting between central and local DP, ISRL-DP makes the realistic assumption (in cross-silo FL) that *people trust their own silo, but not the server or other silos*. In this work, we provide *tight* (up to logarithms) *upper and lower bounds* for ISRL-DP FL with convex/strongly convex loss functions and homogeneous (i.i.d.) silo data. Remarkably, we show that similar bounds are attainable for smooth losses with *arbitrary heterogeneous silo data distributions*, via an accelerated ISRL-DP algorithm. We also provide tight upper and lower bounds for ISRL-DP federated empirical risk minimization, and use acceleration to attain the optimal bounds in fewer rounds of communication than the state-of-the-art. Finally, with a secure "shuffler" to anonymize silo messages (but without a trusted server), our algorithm attains the optimal *central DP* rates under more practical trust assumptions. Numerical experiments show favorable privacy-accuracy tradeoffs for our algorithm in classification and regression tasks.

## 1 Introduction

Machine learning tasks often involve data from different "silos" (e.g. cell-phone users or organizations such as hospitals) containing sensitive information (e.g. location or health records). In federated learning (FL), each silo (a.k.a. "client") stores its data locally and a central server coordinates updates among different silos to achieve a global learning objective (Kairouz et al., 2019). One of the primary reasons for the introduction of FL was to offer greater privacy (McMahan et al., 2017). However, storing data locally is not sufficient to prevent data leakage. Model parameters or updates can still reveal sensitive information (e.g. via model inversion attacks or membership inference attacks) (Fredrikson et al., 2015; He et al., 2019; Song et al., 2020; Zhu & Han, 2020).

*Differential privacy* (DP) (Dwork et al., 2006) protects against privacy attacks. Different notions of DP have been proposed for FL. The works of Jayaraman & Wang (2018); Truex et al. (2019); Noble et al. (2022) considered *central DP* (CDP) FL, which protects the privacy of silos' *aggregated* data against an *external adversary* who observes the *final trained model*.[1] There are two major issues with

---

[1] We abbreviate central differential privacy by CDP. This is different than the *concentrated differential privacy* notion in Bun & Steinke (2016), for which the same abbreviation is sometimes used in other works.

CDP FL: 1) it does not guarantee privacy for *each specific silo*; and 2) it does not guarantee data privacy when an adversarial eavesdropper has access to other silos or the server. To address the first issue, McMahan et al. (2018); Geyer et al. (2017); Jayaraman & Wang (2018); Gade & Vaidya (2018); Wei et al. (2020a); Zhou & Tang (2020); Levy et al. (2021); Ghazi et al. (2021) considered *user-level DP* (a.k.a. client-level DP). User-level DP guarantees privacy of each silo's *full local data set*. This is a practical notion for *cross-device FL*, where each silo/client corresponds to a single person (e.g. cell-phone user) with many records (e.g. text messages). However, user-level DP still suffers from the second critical shortcoming of CDP: it *allows silo data to be leaked to an untrusted server or to other silos*. Furthermore, user-level DP is less suitable for *cross-silo FL*, where silos are typically organizations (e.g. hospitals, banks, or schools) that contain data from many different people (e.g. patients, customers, or students). In cross-silo FL, each person has a record (a.k.a. "item") that may contain sensitive data. Thus, an appropriate notion of DP for cross-silo FL should protect the privacy of *each individual record* ("item-level DP") in silo $i$, rather than silo $i$'s full aggregate data.

Another notion of DP is *local DP* (LDP) (Kasiviswanathan et al., 2011; Duchi et al., 2013). While central and user-level DP assume that people trust all of the silos and the server, LDP assumes that individuals (e.g. patients) do not trust *anyone* else with their sensitive data, *not even their own silo* (e.g. hospital). Thus, LDP would require each person (e.g. patient) to randomize her report (e.g. medical test results) before releasing it (e.g. to their own doctor/hospital). Since patients/customers/students usually trust their *own* hospital/bank/school, LDP may be unnecessarily stringent, *hindering performance/accuracy*.

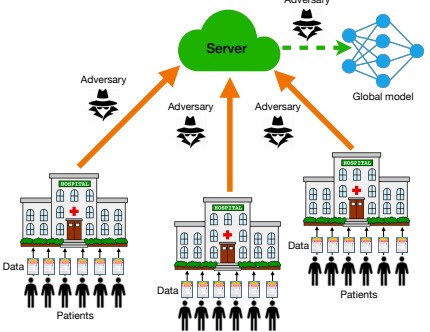

In this work, we consider a privacy notion called *inter-silo record-level differential privacy* (ISRL-DP), which requires that all of the communications of each silo satisfy (item-level) DP; see Fig. 1. By the post-processing property of DP, this also ensures that the the broadcasts by the server and the global model are DP. Privacy notions similar or identical to ISRL-DP have been considered in Truex

Figure 1: ISRL-DP protects the privacy of each patient's record regardless of whether the server/other silos are trustworthy, as long as the patient's *own hospital* is trusted. By contrast, user-level DP protects aggregate data of patients in hospital $i$ and does not protect against adversarial server/other silos.

et al. (2020); Huang et al. (2020); Huang & Gong (2020); Wu et al. (2019); Wei et al. (2020b); Dobbe et al. (2020); Zhao et al. (2020); Arachchige et al. (2019); Seif et al. (2020); Liu et al. (2022). We provide a rigorous definition of ISRL-DP in Definition 2 and Appendix B.

**Why ISRL-DP?** ISRL-DP is the natural notion of DP for cross-silo FL, where each silo contains data from many individuals who trust their own silo but may not trust the server or other silos (e.g., hospitals in Fig. 1). The item-level privacy guarantee that ISRL-DP provides for each silo (e.g. hospital) ensures that no person's record can be leaked. In contrast to central DP and user-level DP, the protection of ISRL-DP is guaranteed *even against an adversary with access to the server and/or other silos* (e.g. hospitals). This is because each silo's *communications* are DP with respect to their own data records and cannot leak information to any adversarial eavesdropper. On the other hand, since individuals (e.g. patients) trust their own silo (e.g. hospital), ISRL-DP does not require individuals to randomize their own data reports (e.g. health records). Thus, ISRL-DP leads to better performance/accuracy than local DP by relaxing the strict local DP requirement. Another benefit of ISRL-DP is that each silo $i$ can set its own $(\epsilon_i, \delta_i)$ item-level DP budget depending on its privacy needs; see Appendix H and also Liu et al. (2022); Aldaghri et al. (2021).

In addition, ISRL-DP can be useful in *cross-device* FL without a trusted server: If the ISRL privacy parameters are chosen sufficiently small, then *ISRL-DP implies user-level DP* (see Appendix C). Unlike user-level DP, ISRL-DP does not allow data to be leaked to the untrusted server/other users.

Another intermediate trust model between the low-trust local model and the high-trust central/user-level models is the *shuffle model* of DP (Bittau et al., 2017; Cheu et al., 2019). In this model, a secure shuffler receives noisy reports from the silos and randomly permutes them before the reports are sent to the untrusted server.[2] An algorithm is *Shuffle Differentially Private* (SDP) if silos' shuffled

---

[2]Assume that the reports can be decrypted by the server, but not by the shuffler Erlingsson et al. (2020a); Feldman et al. (2020b).

messages are CDP; see Definition 3. Fig. 2 summarizes which parties are assumed to be trustworthy (from the perspective of a person contributing data to a silo) in each of the described privacy notions.

**Problem setup:** Consider a FL setting with $N$ silos, each containing a local data set with $n$ samples:[3] $X_i = (x_{i,1}, \cdots, x_{i,n})$ for $i \in [N] := \{1, \ldots, N\}$. In each round of communication $r$, silos download the global model $w_r$ and use their local data to improve the model. Then, silos send local updates to the server (or other silos, in peer-to-peer FL), who updates the global model to $w_{r+1}$. For each silo $i$, let $\mathcal{D}_i$ be an unknown probability distribution on a data universe $\mathcal{X}_i$ (i.e. $X_i \in \mathcal{X}_i^n$). Let $\mathcal{X} := \bigcup_{i=1}^{N} \mathcal{X}_i$. Given a loss function $f : \mathcal{W} \times \mathcal{X} \to \mathbb{R}$, define silo $i$'s local objective as

$$F_i(w) := \mathbb{E}_{x_i \sim \mathcal{D}_i}[f(w, x_i)], \tag{1}$$

where $\mathcal{W} \subset \mathbb{R}^d$ is a parameter domain. Our goal is to find a model parameter that performs well for all silos, by solving the FL problem

$$\min_{w \in \mathcal{W}} \left\{ F(w) := \frac{1}{N} \sum_{i=1}^{N} F_i(w) \right\}, \tag{2}$$

while maintaining the privacy of each silo's local data. At times, we will focus on *empirical risk minimization* (ERM), where $\widehat{F}_i(w) := \frac{1}{n} \sum_{j=1}^{n} f(w, x_{i,j})$ is silo $i$'s local objective. Thus, in the ERM case, our goal is to solve $\min_{w \in \mathcal{W}} \widehat{F}_{\mathbf{X}}(w) := \frac{1}{N} \sum_{i=1}^{N} \widehat{F}_i(w)$, while maintaining privacy. When $F_i$ takes the form (1) (not necessarily ERM), we may refer to the problem as *stochastic convex optimization* (SCO) for emphasis. For ERM, we make no assumptions on the data; for SCO, we assume the samples $\{x_{i,j}\}_{i \in [N], j \in [n]}$ are drawn independently. For SCO, we say problem (2) is "i.i.d." or "homogeneous" if $\mathcal{X}_i = \mathcal{X}$ and $\mathcal{D}_i = \mathcal{D}, \forall i$. The *excess risk* of an algorithm $\mathcal{A}$ for solving (2) is $\mathbb{E}F(\mathcal{A}(\mathbf{X})) - F^*$, where $F^* = \inf_{w \in \mathcal{W}} F(w)$ and the expectation is taken over both the random draw of $\mathbf{X} = (X_1, \ldots, X_N)$ and the randomness of $\mathcal{A}$. For ERM, the *excess empirical risk* of $\mathcal{A}$ is $\mathbb{E}\widehat{F}_{\mathbf{X}}(\mathcal{A}(\mathbf{X})) - \widehat{F}_{\mathbf{X}}^*$, where the expectation is taken solely over the randomness of $\mathcal{A}$. Thus, a fundamental

Figure 2: Trust assumptions of DP FL notions: We put "trust" in quotes because the shuffler is assumed to be secure and silo messages must already satisfy (at least a weak level of) ISRL-DP in order to realize SDP: anonymization alone cannot "create" DP (Dwork & Roth, 2014).

question in FL is about the *minimum achievable excess risk while maintaining privacy*. In this work, we specifically study the following questions for FL with convex and strongly convex loss functions:[4]

> **Question 1.** What is the minimum achievable excess risk for solving (2) with inter-silo record-level DP?
>
> **Question 2.** With a trusted shuffler (but no trusted server), can the optimal central DP rates be attained?

**Contributions:** Our first contribution is a complete answer to **Question 1** when silo data is *i.i.d.*: we give tight upper and lower bounds in Section 2. The ISRL-DP rates sit between the local DP and central DP rates: higher trust allows for higher accuracy. Further, we show that the ISRL-DP rates nearly match the optimal non-private rates if $d \lesssim n\epsilon_0^2$, where $\epsilon_0$ is the ISRL-DP parameter ("privacy for free"). As a corollary of our analysis, we also derive tight upper and lower bounds for FL algorithms that satisfy *both ISRL-DP and user-level DP simultaneously*, which could be useful in cross-device settings where (e.g. cell phone) users don't trust the server or other users with their sensitive data (e.g. text messages): see Appendix D.3.5.

Second, we give a complete answer to **Question 1** when $F = \widehat{F}_{\mathbf{X}}$ is an *empirical* loss in Section 4.[5] While (Girgis et al., 2021) provided a tight upper bound for the (non-strongly) convex case, we use a

---

[3]In Appendix H, we consider the more general setting where data set sizes $n_i$ and ISRL-DP parameters $(\epsilon_i, \delta_i)$ may vary across silos, and the weights $p_i$ on each silo's loss $F_i$ in (2) may differ (i.e. $p_i \neq 1/N$).

[4]Function $g : \mathcal{W} \to \mathbb{R}$ is $\mu$-*strongly convex* ($\mu \geqslant 0$) if $g(w) \geqslant g(w') + \langle \partial g(w'), w - w' \rangle + \frac{\mu}{2}\|w - w'\|^2 \, \forall \, w, w' \in \mathcal{W}$ and all sub-gradients $\partial g(w')$. If $\mu = 0$, $g$ is *convex*.

[5]ERM is a special case of the FL problem (2): if $\mathcal{D}_i$ is the empirical distribution on $X_i$, then $F = \widehat{F}_{\mathbf{X}}$.

novel *accelerated* algorithm to achieve this upper bound in fewer communication rounds. Further, we obtain matching lower bounds. We also cover the strongly convex case.

Third, we give a partial answer to **Question 1** when silo data is *heterogeneous* (non-i.i.d.), providing algorithms for *smooth* $f(\cdot, x)$ that *nearly* achieve the optimal i.i.d. rates in Section 3. For example, if $f(\cdot, x)$ is $\mu$-strongly convex and $\beta$-smooth, then the excess risk of our algorithm nearly matches the i.i.d. lower bound up to a multiplicative factor of $\widetilde{\mathcal{O}}(\beta/\mu)$. Our algorithm is significantly more effective (in terms of excess risk) than existing ISRL-DP FL algorithms (e.g. Arachchige et al. (2019); Dobbe et al. (2020); Zhao et al. (2020)): see Appendix A for a thorough discussion of related work.

Fourth, we address **Question 2** in Section 5: We give a positive answer to **Question 2** when silo data is i.i.d. Further, with heterogeneous silo data, the optimal central DP rates are *nearly* achieved without a trusted server, if the loss function is smooth. We summarize our results in Fig. 3.

## 1.1 PRELIMINARIES

**Differential Privacy:** Let $\mathbb{X} = \mathcal{X}_1^n \times \cdots \mathcal{X}^n$ and $\rho : \mathbb{X}^2 \to [0, \infty)$ be a distance between databases. Two databases $\mathbf{X}, \mathbf{X}' \in \mathbb{X}$ are $\rho$-*adjacent* if $\rho(\mathbf{X}, \mathbf{X}') \leqslant 1$. DP ensures that (with high probability) an adversary cannot distinguish between the outputs of algorithm $\mathcal{A}$ when it is run on adjacent databases:

**Definition 1** (Differential Privacy). *Let* $\epsilon \geqslant 0, \; \delta \in [0, 1)$. *A randomized algorithm* $\mathcal{A} : \mathbb{X} \to \mathcal{W}$ *is* $(\epsilon, \delta)$-*differentially private (DP) (with respect to* $\rho$*) if for all* $\rho$-*adjacent data sets* $\mathbf{X}, \mathbf{X}' \in \mathbb{X}$ *and all measurable subsets* $S \subseteq \mathcal{W}$, *we have*

$$\mathbb{P}(\mathcal{A}(\mathbf{X}) \in S) \leqslant e^\epsilon \mathbb{P}(\mathcal{A}(\mathbf{X}') \in S) + \delta. \quad (3)$$

**Definition 2** (Inter-Silo Record-Level Differential Privacy). *Let* $\rho_i : \mathcal{X}_i^2 \to [0, \infty)$, $\rho_i(X_i, X_i') := \sum_{j=1}^n \mathbb{1}_{\{x_{i,j} \neq x_{i,j}'\}}, \; i \in [N]$. *A randomized algorithm* $\mathcal{A}$ *is* $(\epsilon_0, \delta_0)$-*ISRL-DP if for all* $i \in [N]$ *and all* $\rho_i$-*adjacent silo data sets* $X_i, X_i'$, *the full transcript of silo* $i$*'s sent messages satisfies* (3) *for any fixed settings of other silos' messages and data.*

**Definition 3** (Shuffle Differential Privacy (Bittau et al., 2017; Cheu et al., 2019)). *A randomized algorithm* $\mathcal{A}$ *is* $(\epsilon, \delta)$-*shuffle DP (SDP) if for all* $\rho$-*adjacent databases* $\mathbf{X}, \mathbf{X}' \in \mathbb{X}$ *and all measurable subsets* $S$, *the collection of all uniformly randomly permuted messages that are sent by the shuffler satisfies* (3), *with* $\rho(\mathbf{X}, \mathbf{X}') := \sum_{i=1}^N \sum_{j=1}^n \mathbb{1}_{\{x_{i,j} \neq x_{i,j}'\}}$.

| FL problem | Loss Func. | Excess Risk Upper Bound | | Excess Risk Lower Bound | |
|---|---|---|---|---|---|
| ISRL-DP i.i.d. SCO | Convex | $\frac{1}{\sqrt{nN}} + \frac{\sqrt{d}}{\epsilon_0 n \sqrt{N}}$ | (Thm. 2.1) | $\frac{1}{\sqrt{nN}} + \frac{\sqrt{d}}{\epsilon_0 n \sqrt{N}}$ | (Thm. 2.2) |
| | Strongly Convex | $\frac{1}{nN} + \frac{d}{\epsilon_0^2 n^2 N}$ | (Thm. 2.1) | $\frac{1}{nN} + \frac{d}{\epsilon_0^2 n^2 N}$ | (Thm. 2.2) |
| ISRL-DP non-i.i.d. SCO | Smooth Convex | $\frac{1}{\sqrt{nN}} + \left(\frac{\sqrt{d}}{\epsilon_0 n \sqrt{N}}\right)^{4/5}$ | (Thm. 3.1) | $\frac{1}{\sqrt{nN}} + \frac{\sqrt{d}}{\epsilon_0 n \sqrt{N} C^2}$ | (Thm. 2.2*) |
| | Smooth strongly | $\frac{1}{nN} + \frac{\sqrt{\kappa d}}{\epsilon_0^2 n^2 N}$ | (Thm. 3.1) | $\frac{1}{nN} + \frac{d}{\epsilon_0^2 n^2 N \kappa}$ | (Thm. 2.2*) |
| ISRL-DP ERM | Convex | $\frac{\sqrt{d}}{\epsilon_0 n \sqrt{N}}$ | (Girgis et al., 2021) + (Thm. 4.1) | $\frac{\sqrt{d}}{\epsilon_0 n \sqrt{N}}$ | (Thm. 4.2) |
| | Strongly Convex | $\frac{d}{\epsilon_0^2 n^2 N}$ | (Thm. 4.1) | $\frac{d}{\epsilon_0^2 n^2 N}$ | (Thm. 4.2) |
| SDP i.i.d. SCO | Convex | $\frac{1}{\sqrt{nN}} + \frac{\sqrt{d}}{\epsilon n N}$ | (Thm. 5.1) | $\frac{1}{\sqrt{nN}} + \frac{\sqrt{d}}{\epsilon n N}$ | (Bassily et al., 2019) |
| | Strongly Convex | $\frac{1}{nN} + \frac{d}{\epsilon^2 n^2 N^2}$ | (Thm. 5.1) | $\frac{1}{nN} + \frac{d}{\epsilon^2 n^2 N^2}$ | (Bassily et al., 2019) |
| SDP non-i.i.d. SCO | Smooth Convex | $\frac{1}{\sqrt{nN}} + \left(\frac{\sqrt{d}}{\epsilon n N}\right)^{4/5}$ | (Thm. 5.2) | $\frac{1}{\sqrt{nN}} + \frac{\sqrt{d}}{\epsilon n N}$ | (Bassily et al., 2019) |
| | Smooth strongly | $\frac{1}{nN} + \frac{\sqrt{\kappa d}}{\epsilon^2 n^2 N^2}$ | (Thm. 5.2) | $\frac{1}{nN} + \frac{d}{\epsilon^2 n^2 N^2}$ | (Bassily et al., 2019) |

Figure 3: We fix $M = N$, omit logs, and $C^2 := (\epsilon_0 n \sqrt{N}/\sqrt{d})^{2/5}$. Round complexity in Theorem 4.1 improves on (Girgis et al., 2021). *For our non-i.i.d. algorithm, Theorem 2.2 only applies if $\epsilon_0 = \mathcal{O}(1/n)$ or $N = \mathcal{O}(1)$: see Appendix D.3.

**Notation and Assumptions:** Let $\| \cdot \|$ be the $\ell_2$ norm and $\Pi_\mathcal{W}(z) := \operatorname{argmin}_{w \in \mathcal{W}} \|w - z\|^2$ denote the projection operator. Function $h : \mathcal{W} \to \mathbb{R}^m$ is *L-Lipschitz* if $\|h(w) - h(w')\| \leqslant L\|w - w'\|$, $\forall w, w' \in \mathcal{W}$. A differentiable function $h(\cdot)$ is $\beta$-*smooth* if its derivative $\nabla h$ is $\beta$-Lipschitz. If $h$ is $\beta$-smooth and $\mu$-strongly convex, denote its *condition number* by $\kappa := \beta/\mu$. For differentiable (w.r.t. $w$) $f(w, x)$, we denote its gradient w.r.t. $w$ by $\nabla f(w, x)$. Write $a \lesssim b$ if $\exists C > 0$ such that $a \leqslant Cb$. Write $a = \widetilde{\mathcal{O}}(b)$ if $a \lesssim \log^2(\theta)b$ for some parameters $\theta$. Assume the following throughout:

**Assumption 1.** *1. $\mathcal{W} \subset \mathbb{R}^d$ is closed, convex, and $\|w - w'\| \leqslant D, \; \forall w, w' \in \mathcal{W}$.*

2. *$f(\cdot, x)$ is L-Lipschitz and convex for all $x \in \mathcal{X}$. In some parts of the paper, we assume $f(\cdot, x)$ is $\mu$-strongly convex.*

3. *$\sup_{w \in \mathcal{W}} \mathbb{E}_{x_i \sim \mathcal{D}_i} \|\nabla f(w, x_i) - \nabla F_i(w)\|^2 \leqslant \phi^2$ for all $i \in [N]$.*

4. *In each round $r$, a uniformly random subset $S_r$ of $M_r \in [N]$ silos is available to communicate with the server, where $\{M_r\}_{r \geqslant 0}$ are independent random variables with $\frac{1}{M} := \mathbb{E}(\frac{1}{M_r})$.*

In Assumption 1 part 4, the network determines $M_r$: it is not a design parameter. This assumption is more general (and realistic for cross-device FL (Kairouz et al., 2019)) than most (DP) FL works, which usually assume $M = N$ or $M_r = M$ is deterministic. On the other hand, in cross-silo FL, typically all silos can reliably communicate in each round, i.e. $M = N$ (Kairouz et al., 2019).

## 2 INTER-SILO RECORD-LEVEL DP FL WITH HOMOGENEOUS SILO DATA

In this section, we provide tight (up to logarithms) upper and lower bounds on the excess risk of ISRL-DP algorithms for FL with i.i.d. silo data. For consistency of presentation, we assume that there is an untrusted server. However, our algorithms readily extend to peer-to-peer FL (no server), by having silos send private messages directly to each other and perform model updates themselves.

### 2.1 UPPER BOUNDS VIA NOISY DISTRIBUTED MINIBATCH SGD

We begin with our upper bounds, obtained via *Noisy Distributed Minibatch SGD (MB-SGD)*: In each round $r$, all $M_r$ available silos receive $w_r$ from the server and send noisy stochastic gradients to the server: $\widetilde{g}_r^i := \frac{1}{K} \sum_{j=1}^{K} \nabla f(w_r, x_{i,j}^r) + u_i$, where $u_i \sim \mathcal{N}(0, \sigma^2 \mathbf{I}_d)$ and $x_{i,j}^r$ are drawn uniformly from $X_i$ (and then replaced). The server averages these $M_r$ reports and updates $w_{r+1} := \Pi_{\mathcal{W}}[w_r - \frac{\eta_r}{M_r} \sum_{i \in S_r} \widetilde{g}_r^i]$. After $R$ rounds, a weighted average of the iterates is returned: $\widehat{w}_R = \frac{1}{\Gamma_R} \sum_{r=0}^{R-1} \gamma_r w_r$ with $\Gamma_R := \sum_{r=0}^{R-1} \gamma_r$. With proper choices of $\{\eta_r, \gamma_r\}_{r=0}^{R-1}$, $\sigma^2$, $K$, and $R$, we have:

**Theorem 2.1** (Informal). *Let $\epsilon_0 \leqslant 2\ln(2/\delta_0), \delta_0 \in (0, 1)$. Then Noisy MB-SGD is $(\epsilon_0, \delta_0)$-ISRL-DP. Moreover:*
*1. If $f(\cdot, x)$ is convex, then*

$$\mathbb{E}F(\widehat{w}_R) - F^* = \widetilde{\mathcal{O}}\left(\frac{LD}{\sqrt{M}}\left(\frac{1}{\sqrt{n}} + \frac{\sqrt{d\ln(1/\delta_0)}}{\epsilon_0 n}\right)\right). \tag{4}$$

*2. If $f(\cdot, x)$ is $\mu$-strongly convex, then*

$$\mathbb{E}F(\widehat{w}_R) - F^* = \widetilde{\mathcal{O}}\left(\frac{L^2}{\mu M}\left(\frac{1}{n} + \frac{d\ln(1/\delta_0)}{\epsilon_0^2 n^2}\right)\right). \tag{5}$$

The first terms in each of (4) and (5) ($LD/\sqrt{Mn}$ for convex and $L^2/\mu Mn$ for strongly convex) are bounds on the *uniform stability* (Bousquet & Elisseeff, 2002) of Noisy MB-SGD. We use these uniform stability bounds to bound the generalization error of our algorithm. Our stability bound for $\mu > 0$ in Lemma D.2 is novel even for $N = 1$. The second terms in (4) and (5) are bounds on the empirical risk of the algorithm. We use Nesterov smoothing (Nesterov, 2005) to extend our bounds to non-smooth losses. This requires us to choose a different stepsize and $R$ from Bassily et al. (2019) (for $N = 1, \mu = 0$), which eliminates the restriction on the smoothness parameter that appears in (Bassily et al., 2019, Theorem 3.2).[6] Privacy of Noisy MB-SGD follows from the Gaussian mechanism (Dwork & Roth, 2014, Theorem A.2), privacy amplification by subsampling (Ullman, 2017), and advanced composition (Dwork & Roth, 2014, Theorem 3.20) or moments accountant (Abadi et al., 2016, Theorem 1). See Appendix D.2 for the detailed proof.

### 2.2 LOWER BOUNDS

We provide excess risk lower bounds for the case $M = N$, establishing the optimality of Noisy MB-SGD for two function classes: $\mathcal{F}_{L,D} := \{f \mid f(\cdot, x)$ is convex, $L$-Lipschitz, $\forall x \in \mathcal{X}$ and $\|w - w'\| \leqslant D \ \forall w, w' \in \mathcal{W}\}$; and $\mathcal{G}_{\mu, L, D} := \{f \in \mathcal{F}_{L,D} \mid f(\cdot, x)$ is $\mu$-strongly convex, $\forall x \in \mathcal{X}\}$. We restrict attention to distributions $\mathcal{D}$ satisfying Assumption 1, part 3. The $(\epsilon_0, \delta_0)$-ISRL-DP algorithm class $\mathbb{A}_{(\epsilon_0, \delta_0), C}$ contains all *sequentially interactive* algorithms and all *fully interactive*, $C$-*compositional* (defined in Appendix D.3) algorithms.[7] If $\mathcal{A}$ is sequentially interactive or $\mathcal{O}(1)$-compositional, write $\mathcal{A} \in \mathbb{A}_{(\epsilon_0, \delta_0)}$. The vast majority of DP algorithms in the literature are $C$-compositional. For example, any algorithm that uses the strong composition theorems of (Dwork & Roth, 2014; Kairouz et al., 2015; Abadi et al., 2016) for its privacy analysis is 1-compositional: see Appendix D.3.2. In particular, Noisy MB-SGD is 1-compositional, hence it is in $\mathbb{A}_{(\epsilon_0, \delta_0)}$.

---

[6]In (Bassily et al., 2019), the number of iterations is denoted by $T$, rather than $R$.

[7]*Sequentially interactive* algorithms can query silos adaptively in sequence, but cannot query any one silo more than once. *Fully interactive* algorithms can query each silo any number of times. See (Joseph et al., 2019) for further discussion.

**Theorem 2.2** (Informal). *Let* $\epsilon_0 \in (0, \sqrt{N}]$, $\delta_0 = o(1/nN)$, $M = N$, *and* $\mathcal{A} \in \mathbb{A}_{(\epsilon_0, \delta_0), C}$. *Then:*
*1. There exists a* $f \in \mathcal{F}_{L,D}$ *and a distribution* $\mathcal{D}$ *such that for* $\mathbf{X} \sim \mathcal{D}^{nN}$, *we have:*

$$\mathbb{E}F(\mathcal{A}(\mathbf{X})) - F^* = \widetilde{\Omega}\left(\frac{\phi D}{\sqrt{Nn}} + LD\min\left\{1, \frac{\sqrt{d}}{\epsilon_0 n\sqrt{N}C^2}\right\}\right).$$

*2. There exists a* $\mu$*-smooth* $f \in \mathcal{G}_{\mu, L, D}$ *and distribution* $\mathcal{D}$ *such that for* $\mathbf{X} \sim \mathcal{D}^{nN}$, *we have*

$$\mathbb{E}F(\mathcal{A}(\mathbf{X})) - F^* = \widetilde{\Omega}\left(\frac{\phi^2}{\mu nN} + LD\min\left\{1, \frac{d}{\epsilon_0^2 n^2 NC^4}\right\}\right).$$

*Further, if* $\mathcal{A} \in \mathbb{A}$*, then the above lower bounds hold with* $C = 1$.

The lower bounds for $\mathbb{A}_{(\epsilon_0, \delta_0)}$ are nearly tight[8] by Theorem 2.1. The first term in each of the lower bounds is the optimal non-private rate; the second terms are proved in Appendix D.3.4. In particular, if $d \lesssim \epsilon_0^2 n$, then the non-private term in each bound is dominant, so the ISRL-DP rates match the respective non-private rates, resulting in "privacy for free" (Nemirovskii & Yudin, 1983). The ISRL-DP rates sit between the rates for LDP and CDP: higher trust allows for higher accuracy. For example, for $\mathcal{F}_{L,D}$, the LDP rate is $\Theta(LD\min\{1, \sqrt{d}/\epsilon_0\sqrt{N}\})$ (Duchi et al., 2013), and the CDP rate is $\Theta(\phi^2/\sqrt{Nn} + LD\min\{1, \sqrt{d}/\epsilon_0 nN\})$ (Bassily et al., 2019).

Theorem 2.2 is more generally applicable than existing LDP and CDP lower bounds. When $n = 1$, ISRL-DP is equivalent to LDP and Theorem 2.2 recovers the LDP lower bounds (Duchi et al., 2013; Smith et al., 2017). However, Theorem 2.2 holds for a wider class of algorithms than the lower bounds of (Duchi et al., 2013; Smith et al., 2017), which were limited to *sequentially interactive* LDP algorithms. When $N = 1$, ISRL-DP is equivalent to CDP and Theorem 2.2 recovers the CDP lower bounds (Bassily et al., 2019).

Obtaining our more general lower bounds under the more complex notion of ISRL-DP is challenging. The lower bound approaches of (Duchi et al., 2013; Smith et al., 2017; Duchi & Rogers, 2019) are heavily tailored to LDP and sequentially interactive algorithms. Further, the applicability of the standard CDP lower bound framework (e.g. (Bassily et al., 2014; 2019)) to ISRL-DP FL is unclear.

In light of these challenges, we take a different approach to proving Theorem 2.2: We first analyze the *central* DP guarantees of $\mathcal{A}$ when silo data sets $X_1, \cdots, X_N$ are shuffled in each round, showing that CDP amplifies to $\epsilon = \widetilde{O}(\frac{\epsilon_0}{\sqrt{N}})$. We could not have concluded this from existing amplification results (Erlingsson et al., 2020b; Feldman et al., 2020b; Balle et al., 2019; Cheu et al., 2019; Balle et al., 2020) since these results are all limited to sequentially interactive LDP algorithms and $n = 1$. Thus, we prove *the first privacy amplification by shuffling bound for fully interactive ISRL-DP FL algorithms*. Then, we apply the CDP lower bounds of Bassily et al. (2019) to $\mathcal{A}_s$, the "shuffled" version of $\mathcal{A}$. This implies that the shuffled algorithm $\mathcal{A}_s$ has excess population loss that is lower bounded as in Theorem 2.2. Finally, we observe that the i.i.d. assumption implies that $\mathcal{A}_s$ and $\mathcal{A}$ have the same expected population loss. Note that our proof techniques can also be used to obtain ISRL-DP lower bounds for other problems in which a CDP lower bound is known.

## 3 INTER-SILO RECORD-LEVEL DP FL WITH HETEROGENEOUS SILO DATA

Consider the **non-i.i.d.** FL problem, where $F_i(w)$ takes the form (1) for some unknown distributions $\mathcal{D}_i$ on $\mathcal{X}_i, \forall i$. The uniform stability approach that we used to obtain our i.i.d. upper bounds does not work in this setting.[9] Instead, we directly minimize $F$ by modifying Noisy MB-SGD as follows:
**1.** We draw disjoint batches of $K$ local samples *without replacement* from each silo and set $R = \lfloor \frac{n}{K} \rfloor$. Thus, stochastic gradients are independent across iterations, so our bounds apply to $F$.
**2.** We use *acceleration* (Ghadimi & Lan, 2012) to increase the convergence rate.
**3.** To provide ISRL-DP, we re-calibrate $\sigma^2$ and apply *parallel composition* (McSherry, 2009).
After these modifications, we call the resulting algorithm *One-Pass Accelerated Noisy MB-SGD*. It is described in Algorithm 1. In the strongly convex case, we use a *multi-stage implementation* of One-Pass Accelerated Noisy MB-SGD (inspired by (Ghadimi & Lan, 2013)) to further expedite convergence: see Appendix E.1 for details. Carefully tuning step sizes, $\sigma^2$, and $K$ yields:

---

[8]Up to logarithms, a factor of $\phi/L$, and for strongly convex case–a factor of $\mu D/L$. If $d > \epsilon_0^2 n^2 N$, then the ISRL-DP algorithm that outputs any $w_0 \in \mathcal{W}$ attains the matching upper bound $\mathcal{O}(LD)$.

[9]Specifically, Lemma D.1 in Appendix D does not apply without the i.i.d. assumption.

**Theorem 3.1** ($M = N$ case). *Let $f(\cdot, x)$ be $\beta$-smooth for all $x \in \mathcal{X}$. Assume $\epsilon_0 \leqslant 8 \ln(1/\delta_0)$, $\delta_0 \in (0, 1)$. Then One-Pass Accelerated Noisy MB-SGD is $(\epsilon_0, \delta_0)$-ISRL-DP. Moreover, if $M = N$, then:*
*1. For convex $f(\cdot, x)$, we have*

$$\mathbb{E}F(w_R^{ag}) - F^* \lesssim \frac{\phi D}{\sqrt{Nn}} + \left( \frac{\beta^{1/4} L D^{3/2} \sqrt{d \ln(1/\delta_0)}}{\epsilon_0 n \sqrt{N}} \right)^{4/5}. \tag{6}$$

*2. For $\mu$-strongly convex $f(\cdot, x)$ with $\kappa = \frac{\beta}{\mu}$, we have*

$$\mathbb{E}F(w_R^{ag}) - F^* = \widetilde{\mathcal{O}}\left( \frac{\phi^2}{\mu nN} + \sqrt{\kappa} \frac{L^2}{\mu} \frac{d \ln(1/\delta_0)}{\epsilon_0^2 n^2 N} \right). \tag{7}$$

Remarkably, the bound (7) nearly matches the optimal *i.i.d.* bound (5) up to the factor $\widetilde{\mathcal{O}}(\sqrt{\kappa})$. In particular, *for well-conditioned loss functions, our algorithm achieves the optimal i.i.d. rates even when silo data is arbitrarily heterogeneous*. The gap between the bound (6) and the i.i.d. bound (4) is $\mathcal{O}((\sqrt{d}/\epsilon_0 n \sqrt{N})^{1/5})$. Closing the gaps between the non-i.i.d. upper bounds in Theorem 3.1 and the i.i.d. lower bounds in Theorem 2.2 is left as an open problem. Compared to previous upper bounds, Theorem 3.1 is a major improvement: see Appendix A for details.

---

**Algorithm 1** Accelerated Noisy MB-SGD

---

1: **Input:** Data $X_i \in \mathcal{X}_i^n$, $i \in [N]$, strong convexity modulus $\mu \geqslant 0$, noise parameter $\sigma^2$, iteration number $R \in \mathbb{N}$, batch size $K \in [n]$, step size parameters $\{\eta_r\}_{r \in [R]}$, $\{\alpha_r\}_{r \in [R]}$.
2: Initialize $w_0^{ag} = w_0 \in \mathcal{W}$ and $r = 1$.
3: **for** $r \in [R]$ **do**
4:     Server updates and broadcasts $w_r^{md} = \frac{(1-\alpha_r)(\mu+\eta_r)}{\eta_r + (1-\alpha_r^2)\mu} w_{r-1}^{ag} + \frac{\alpha_r[(1-\alpha_r)\mu+\eta_r]}{\eta_r + (1-\alpha_r^2)\mu} w_{r-1}$
5:     **for** $i \in S_r$ **in parallel do**
6:         Silo $i$ draws $\{x_{i,j}^r\}_{j=1}^k$ from $X_i$ (*replace samples for ERM*) and noise $u_i \sim \mathcal{N}(0, \sigma^2 \mathbf{I}_d)$.
7:         Silo $i$ computes $\widetilde{g}_r^i := \frac{1}{K} \sum_{j=1}^K \nabla f(w_r^{md}, x_{i,j}^r) + u_i$.
8:     **end for**
9:     Server aggregates $\widetilde{g}_r := \frac{1}{M_r} \sum_{i \in S_r} \widetilde{g}_r^i$ and updates $w_r := \text{argmin}_{w \in \mathcal{W}} \left\{ \alpha_r \left[ \langle \widetilde{g}_r, w \rangle + \frac{\mu}{2} \|w_r^{md} - w\|^2 \right] + \left[ (1 - \alpha_r)\frac{\mu}{2} + \frac{\eta_r}{2} \right] \|w_{r-1} - w\|^2 \right\}$.
10:     Server updates and broadcasts $w_r^{ag} = \alpha_r w_r + (1 - \alpha_r) w_{r-1}^{ag}$.
11: **end for**
12: **Output:** $w_R^{ag}$.

---

In Appendix E.2, we provide a general version (and proof) of Theorem 3.1 for $M \leqslant N$. If $M < N$ but $M$ is sufficiently large or silo heterogeneity sufficiently small, then the same bounds in Theorem 3.1 hold with $N$ replaced by $M$. Intuitively, the $M < N$ case is harder when data is highly heterogeneous, since stochastic estimates of $\nabla F$ will have larger variance. In Lemma E.3 (Appendix E.2), we use a combinatorial argument to bound the variance of stochastic gradients.

## 4   INTER-SILO RECORD-LEVEL DP FEDERATED ERM

In this section, we provide an ISRL-DP FL algorithm, *Accelerated Noisy MB-SGD*, with optimal excess *empirical* risk. The difference between our proposed algorithm and *One-Pass* Accelerated Noisy MB-SGD is that silo $i$ now samples from $X_i$ *with replacement* in each round: see line 6 in Algorithm 1. This allows us to a) amplify privacy via local subsampling and advanced composition/moments accountant, allowing for smaller $\sigma^2$; and b) run more iterations of our algorithm to better optimize $\widehat{F}_{\mathbf{X}}$. These modifications are necessary for obtaining the optimal rates for federated ERM. We again employ Nesterov smoothing (Nesterov, 2005) to extend our results to non-smooth $f$.

**Theorem 4.1** (Informal). *Let $\epsilon_0 \leqslant 2 \ln(2/\delta_0), \delta_0 \in (0, 1)$. Then, there exist algorithmic parameters such that Algorithm 1 is $(\epsilon_0, \delta_0)$-ISRL-DP and:*
*1. If $f(\cdot, x)$ is convex, then*

$$\mathbb{E}\widehat{F}_{\mathbf{X}}(w_R^{ag}) - \widehat{F}_{\mathbf{X}}^* = \widetilde{\mathcal{O}}\left( LD \frac{\sqrt{d \ln(1/\delta_0)}}{\epsilon_0 n \sqrt{M}} \right). \tag{8}$$

*2. If $f(\cdot, x)$ is $\mu$-strongly convex then*

$$\mathbb{E}\widehat{F}_{\mathbf{X}}(w_R^{ag}) - \widehat{F}_{\mathbf{X}}^* = \widetilde{\mathcal{O}}\left( \frac{L^2}{\mu} \frac{d \ln(1/\delta_0)}{\epsilon_0^2 n^2 M} \right). \tag{9}$$

See Appendix F.1 for the formal statement and proof. With non-random $M_r = M$, Girgis et al. (2021) provides an upper bound for (non-strongly) convex ISRL-DP ERM that nearly matches the one we provide in (8). However, Algorithm 1 achieves the upper bounds for convex and strongly convex loss in fewer rounds of communication than (Girgis et al., 2021): see Appendix F.1 for details. In Appendix F.2, we get matching lower bounds, establishing the optimality of Algorithm 1 for ERM.

## 5    SHUFFLE DP FEDERATED LEARNING

Assume access to a secure shuffler and fix $M_r = M$. In each round $r$, the shuffler receives reports $(Z_r^{(1)}, \cdots Z_r^{(M)})$ from active silos (we assume $S_r = [M]$ here for concreteness), draws a uniformly random permutation of $[M]$, $\pi$, and then sends $(Z_r^{(\pi(1))}, \cdots, Z_r^{(\pi(M))})$ to the server for aggregation. When this shuffling procedure is combined with ISRL-DP Noisy Distributed MB-SGD, we obtain:

**Theorem 5.1** (i.i.d.). *Let $\epsilon \leq \ln(2/\delta)$, $\delta \in (0, 1)$. Then there are choices of algorithmic parameters such that Shuffled Noisy MB-SGD is $(\epsilon, \delta)$-SDP. Moreover:*
*1. If $f(\cdot, x)$ is convex, then*

$$\mathbb{E}F(\widehat{w}_R) - F^* = \widetilde{\mathcal{O}}\left( LD\left( \frac{1}{\sqrt{nM}} + \frac{\sqrt{d\ln(1/\delta)}}{\epsilon nN} \right) \right).$$

*2. If $f(\cdot, x)$ is $\mu$-strongly convex, then*

$$\mathbb{E}F(\widehat{w}_R) - F^* = \widetilde{\mathcal{O}}\left( \frac{L^2}{\mu}\left( \frac{1}{nM} + \frac{d\ln(1/\delta)}{\epsilon^2 n^2 N^2} \right) \right).$$

See Appendix G.1 for details and proof. When $M = N$, the rates in Theorem 5.1 match the *optimal central DP rates* (Bassily et al., 2019), and are attained *without a trusted server*. Thus, with shuffling, *Noisy MB-SGD is simultaneously optimal for i.i.d. FL in the inter-silo and central DP models.*

If silo data is *heterogeneous*, then we use a shuffle DP variation of *One-Pass Accelerated Noisy MB-SGD*, described in Appendix G.2, to get:

**Theorem 5.2** (Non-i.i.d.). *Assume $f(\cdot, x)$ is $\beta$-smooth $\forall x \in \mathcal{X}$. Let $\epsilon \leq 15, \delta \in (0, \frac{1}{2})$. Then, there is an $(\epsilon, \delta)$-SDP variation of One-Pass Accelerated Noisy MB-SGD such that for $M = N$, we have:*
*1. If $f(\cdot, x)$ is convex, then*

$$\mathbb{E}F(w_R^{ag}) - F^* \lesssim \frac{\phi D}{\sqrt{Nn}} + \left( \frac{\beta^{1/4} L D^{3/2}\sqrt{d}\ln(d/\delta)}{\epsilon nN} \right)^{4/5}. \tag{10}$$

*2. If $f(\cdot, x)$ is $\mu$-strongly convex with $\kappa = \frac{\beta}{\mu}$, then*

$$\mathbb{E}F(w_R^{ag}) - F^* = \widetilde{\mathcal{O}}\left( \frac{\phi^2}{\mu nN} + \sqrt{\kappa}\frac{L^2}{\mu}\frac{d\ln(1/\delta)}{\epsilon^2 n^2 N^2} \right). \tag{11}$$

The bound (11) matches the optimal *i.i.d.*, *central DP* bound (Bassily et al., 2019) up to $\widetilde{\mathcal{O}}(\sqrt{\kappa})$. Hence, if $f$ is not ill-conditioned, then (11) shows that *it is not necessary to have either i.i.d. data or a trusted server to attain the optimal CDP rates*. See Appendix G.2 for proof and the $M < N$ case.

## 6    NUMERICAL EXPERIMENTS

We validate our theoretical findings with three sets of experiments. Our results indicate that ISRL-DP MB-SGD yields accurate, private models in practice. Our method performs well even relative to *non-private* Local SGD, a.k.a. FedAvg (McMahan et al., 2017), and outperforms ISRL-DP Local SGD for most privacy levels. Appendix I contains details of experiments, and additional results.[10]

**Binary Logistic Regression with MNIST:** Following (Woodworth et al., 2020b), we consider binary logistic regression on MNIST (LeCun & Cortes, 2010). The task is to classify digits as odd or even. Each of 25 odd/even digit pairings is assigned to a silo ($N = 25$). Fig. 4 shows that *ISRL-DP MB-SGD outperforms (non-private) Local SGD for $\epsilon \geq 12$ and outperforms ISRL-DP Local SGD.*

**Linear Regression with Health Insurance Data:** We divide the data set (Choi, 2018) ($d = 7, n \times N = 1338$) into $N$ silos based on the level of the target: patient medical costs. Fig. 5 shows that ISRL-DP MB-SGD outperforms ISRL-DP Local SGD, especially in the high privacy regime $\epsilon \leq 2$. *For $\epsilon \geq 2$, ISRL-DP MB-SGD is in line with (non-private) Local SGD.*

---

[10]We also describe the ISRL-DP Local SGD baseline in Appendix I.

**Multiclass Logistic Regression with Obesity Data:** We train a softmax regression model for a 7-way classification task with an obesity data set (Palechor & de la Hoz Manotas, 2019). We divide the data into $N = 7$ heterogeneous silos based on the value of the target variable, obesity level, which takes 7 values: Insufficient Weight, Normal Weight, Overweight Level I, Overweight Level II, Obesity Type I, Obesity Type II and Obesity Type III. As shown in Fig. 6, *our algorithm significantly outperforms ISRL-DP Local SGD by* $10 − 30\%$ *across all privacy levels.*

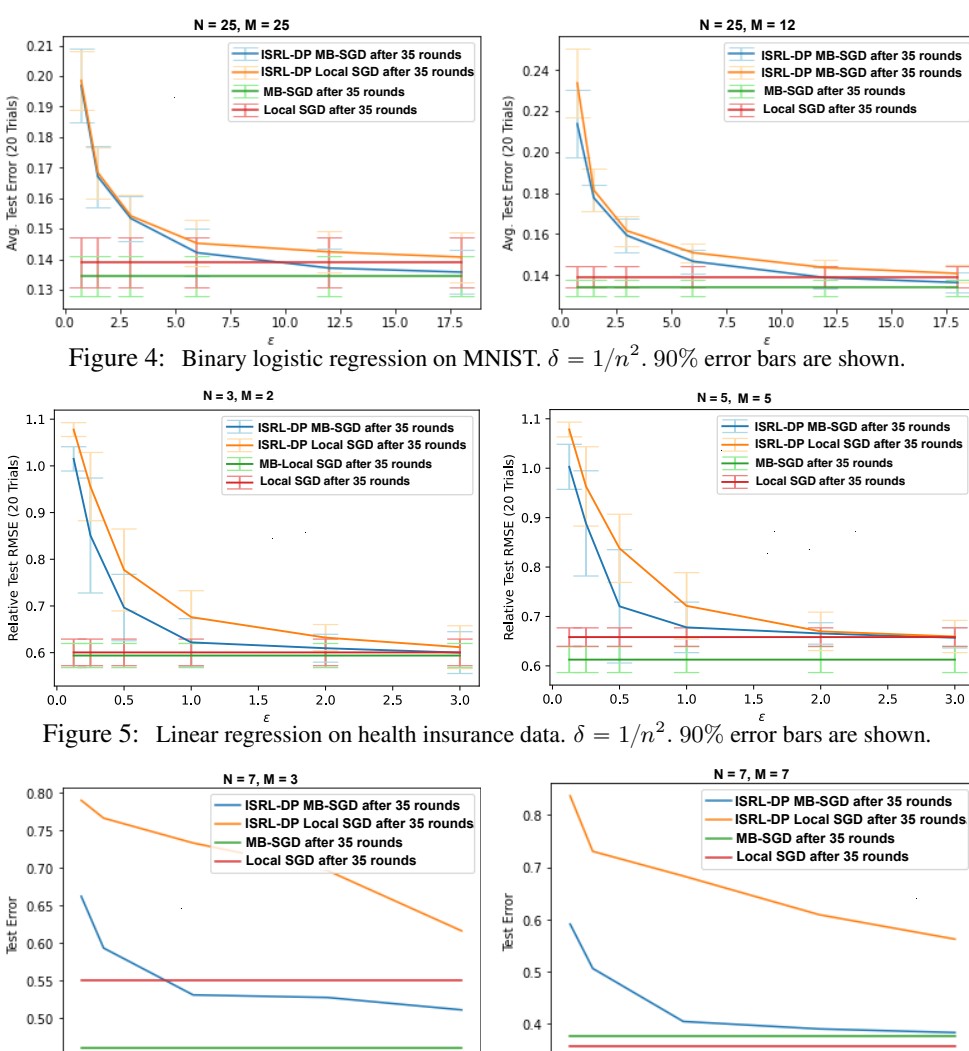

Figure 4: Binary logistic regression on MNIST. $\delta = 1/n^2$. 90% error bars are shown.

Figure 5: Linear regression on health insurance data. $\delta = 1/n^2$. 90% error bars are shown.

Figure 6: Softmax regression on obesity data. $\delta = 1/n^2$.

## 7 CONCLUDING REMARKS AND OPEN QUESTIONS

This paper considered FL without a trusted server and advocated for inter-silo record-level DP (ISRL-DP) as a practical privacy notion in this setting, particularly in cross-silo applications. We provided optimal ISRL-DP algorithms for convex/strongly convex FL in both the *i.i.d.* and *ERM* settings when all $M = N$ clients are able to communicate. The i.i.d. rates sit between the rates for the stringent "no trust" local DP and relaxed "high trust" central DP notions, and allow for "privacy for free" when $d \lesssim \epsilon_0^2 n$. As a side result, in Appendix D.3.5, we established the optimal rates for *cross-device* FL with *user-level DP* in the absence of a trusted server. Additionally, we devised an accelerated ISRL-DP algorithm to obtain state-of-the-art upper bounds for *heterogeneous* FL. We also gave a *shuffle DP* algorithm that (nearly) attains the optimal central DP rates for (non-i.i.d.) i.i.d. FL. An open problem is to close the gap between our i.i.d. lower bounds and non-i.i.d. upper bounds: e.g. can $\sqrt{\kappa}$ in (7) be removed? Also, when $M < N$, are our upper bounds tight? Finally, what performance is possible for *non-convex* ISRL-DP FL?

ACKNOWLEDGMENTS

This work was partly supported by a gift from the USC-Meta Center for Research and Education in AI and Learning.

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

APPENDIX

To ease navigation, we provide a high-level table of contents for this Appendix:

**Contents**

## A  THOROUGH DISCUSSION OF RELATED WORK

**Federated Learning:** In the absence of differential privacy constraints, federated learning (FL) has received a lot of attention from researchers in recent years. Among these, the most relevant works to us are (Koloskova et al., 2020; Li et al., 2020; Karimireddy et al., 2020; Woodworth et al., 2020a;b; Yuan & Ma, 2020), which have proved bounds on the convergence rate of FL algorithms. From an algorithmic standpoint, all of these works propose and analyze either Minibatch SGD (MB-SGD), FedAvg/Local SGD (McMahan et al., 2017), or an extension or accelerated/variance-reduced variation of one of these (e.g. SCAFFOLD (Karimireddy et al., 2020)). Notably, Woodworth et al. (2020b) proves tight upper and lower bounds that establish the near optimality of accelerated MB-SGD for the heterogeneous SCO problem with non-random $M_r = M = N$ in a fairly wide parameter regime. Lobel & Ozdaglar (2010); Touri & Gharesifard (2015); Nedic et al. (2017) provide convergence results with random connectivity graphs. Our upper bounds describe the effect of the mean of $1/M_r$ on DP FL.

**DP Optimization:** In the centralized setting, DP ERM and SCO is well-understood for convex and strongly convex loss functions (Bassily et al., 2014; Wang et al., 2017; Bassily et al., 2019; Feldman et al., 2020a; Lowy & Razaviyayn, 2021; Asi et al., 2021).

Tight excess risk bounds for local DP SCO were provided in Duchi et al. (2013); Smith et al. (2017).

A few works have also considered Shuffle DP ERM and SCO. Girgis et al. (2021); Erlingsson et al. (2020a) showed that the optimal CDP convex ERM rate can be attained in the lower trust (relative to the central model) shuffle model of DP. The main difference between our treatment of shuffle DP and Cheu et al. (2021) is that our results are much more general than Cheu et al. (2021). For example, Cheu et al. (2021) does not consider FL: instead they consider the simpler problem of stochastic convex optimization (SCO). SCO is a simple special case of FL in which each silo has only $n = 1$ sample. Additionally, Cheu et al. (2021) only considers the i.i.d. case, but not the more challenging non-i.i.d. case. Further, Cheu et al. (2021) assumes perfect communication ($M = N$), while we also analyze the case when $M < N$ and some silos are unavailable in certain rounds (e.g. due to internet issues). Note that our bounds in Theorem 5.1 recover the results in Cheu et al. (2021) in the special case considered in their work.

**DP Federated Learning:** More recently, there have been many proposed attempts to ensure the privacy of individuals' data during and after the FL process. Some of these have used secure multi-party computation (MPC) (Chen et al., 2018; Ma et al., 2018), but this approach leaves users vulnerable to inference attacks on the trained model and does not provide the rigorous guarantee of DP. Others (McMahan et al., 2018; Geyer et al., 2017; Jayaraman & Wang, 2018; Gade & Vaidya, 2018; Wei et al., 2020a; Zhou & Tang, 2020; Levy et al., 2021; Ghazi et al., 2021; Noble et al., 2022) have used user-level DP or central DP (CDP), which rely on a trusted third party, or hybrid DP/MPC approaches (Jayaraman & Wang, 2018; Truex et al., 2019). The work of (Jayaraman & Wang, 2018) proves CDP empirical risk bounds and high probability guarantees on the population loss when the data is i.i.d. across silos. However, ISRL-DP and SDP are not considered, nor is heterogeneous (non-i.i.d.) FL. It is also worth mentioning that (Geyer et al., 2017) considers random $M_r$ but does not prove any bounds.

Despite this progress, prior to our present work, very little was known about the excess risk potential of ISRL-DP FL algorithms, except in the two extreme corner cases of $N = 1$ and $n = 1$. When $N = 1$, ISRL-DP and CDP are essentially equivalent; tight ERM (Bassily et al., 2014) and i.i.d. SCO (Bassily et al., 2019; Feldman et al., 2020a) bounds are known for this case. In addition, for LDP i.i.d. SCO when $n = 1$ and $M_r = N$, (Duchi et al., 2013) establishes the minimax optimal rate for the class of sequentially interactive algorithms and non-strongly convex loss functions. To the best of our knowledge, all previous works examining the general ISRL-DP FL problem with arbitrary $n, M, N \geqslant 1$ either focus on ERM and/or do not provide excess risk bounds that scale with both $M$ and $n$, making the upper bounds provided in the present work significantly tighter. Furthermore, none of the existing works on ISRL-DP FL provide lower bounds or upper bounds for the case of random $M_r$. We discuss each of these works in turn below:

(Truex et al., 2020) gives a ISRL-DP FL algorithm, but no risk bounds are provided in their work.

The works of (Huang et al., 2020) and (Huang & Gong, 2020) use ISRL-DP ADMM algorithms for smooth convex Federated ERM. However, the utility bounds in their works are stated in terms of an

average of the silo functions evaluated at different points, so it is not clear how to relate their result to the standard performance measure for learning (which we consider in this paper): expected excess risk at the point $\widehat{w}$ output by the algorithm. Also, no lower bounds are provided for their performance measure. Therefore, the sub-optimality gap of their result is not clear.

(Wu et al., 2019, Theorem 2) provides an $(\epsilon, 0)$-ISRL-DP ERM bound for fixed $M_r = M = N$ of $\mathcal{O}\left(\kappa \frac{L^2}{\mu} \frac{d}{nN\epsilon^2} + \epsilon\right)$ for $\mu$-strongly convex, $\beta$-smooth $f$ with condition number $\kappa = \beta/\mu$, and $1/\epsilon^2$ is an average of $1/\epsilon_i^2$. The additive $\epsilon$ term is clearly problematic: e.g. if $\epsilon = \Theta(1)$, then the bound becomes trivial. Ignoring this term, the first term in their bound is still looser than the bound that we provide in Theorem 4.1. Namely, for $\epsilon = \epsilon_0$, our bound in part 2 of Theorem 4.1 is tighter by a factor of $\mathcal{O}\left(\frac{\ln(1/\delta_0)}{\kappa n}\right)$ and does not require $\beta$-smoothness of the loss. Additionally, the bounds in (Wu et al., 2019) require $R$ "large enough" and do not come with communication complexity guarantees. In the convex case, the ISRL-DP ERM bound reported in (Wu et al., 2019, Theorem 3) is difficult to interpret because the unspecified "constants" in the upper bound on $\mathbb{E}\widehat{F}_{\mathbf{X}}(\widehat{w}_R) - \widehat{F}_{\mathbf{X}}^*$ are said to be allowed to depend on $R$.

(Wei et al., 2020b, Theorems 2-3) provide convergence rates for smooth convex Polyak-Łojasiewicz (a generalization of strong convexity) ISRL-DP ERM, which are complicated non-monotonic functions of $R$. Since they do not prescribe a choice of $R$, it is unclear what excess loss and communication complexity bounds are attainable with their algorithm.

(Dobbe et al., 2020) proposes a ISRL-DP Inexact Alternating Minimization Algorithm (IAMA) with Laplace noise; their result (Dobbe et al., 2020, Theorem 3.11) gives a convergence rate for smooth, strongly convex ISRL-DP FL of order $\mathcal{O}\left(\frac{\Theta \sum_{i \in [M]} \sigma_i^2}{R}\right)$ ignoring smoothness and strong convexity factors, where $\Theta$ is a parameter that is only upper bounded in special cases (e.g. quadratic objective). Thus, the bounds given in (Dobbe et al., 2020) are not complete for general strongly convex loss functions. Even in the special cases where a bound for $\Theta$ is provided, our bounds are tighter. Assuming that $\Theta = 1$ and (for simplicity of exposition) that parameters are the same across silos, (Dobbe et al., 2020, Theorem 3.11) implies taking $\sigma^2 = 1/\epsilon^2$ to ensure $(\epsilon, 0)$-ISRL-DP. The resulting convergence rate is then $O(M/\epsilon^2)$, which does not scale with $n$ and is *increasing* with $M$. Also, the dependence of their rate on the dimension $d$ is unclear, as it does not appear explicitly in their theorem.[11] Ignoring this issue, the dependence on $M$ and $n$ in the bound of (Dobbe et al., 2020) is still looser than all of the excess loss bounds that we provide in the present work.

(Zhao et al., 2020) and (Arachchige et al., 2019) apply the ISRL-DP FL framework to Internet of Things, and (Seif et al., 2020) uses noisy (full-batch) GD for ISRL-DP wireless channels in the FL (smooth strongly convex) ERM setting. The bounds in these works do not scale with the number of data points $n$, however (only with the number of silos $N$). Therefore, the bounds that we provide in the present work are tighter, and apply to general convex FL problems besides wireless channels and Internet of Things.

## B RIGOROUS DEFINITION OF INTER-SILO RECORD-LEVEL DP

Recall:

**Definition 4.** *(Differential Privacy) Let $\epsilon \geqslant 0$, $\delta \in [0, 1)$. A randomized algorithm $\mathcal{A} : \mathbb{X} \to \mathcal{W}$ is $(\epsilon, \delta)$-DP if for all $\rho$-adjacent data sets $\mathbf{X}, \mathbf{X}' \in \mathbb{X}$ and all measurable subsets $S \subset \mathcal{W}$, we have*

$$\mathbb{P}(\mathcal{A}(\mathbf{X}) \in S) \leqslant e^\epsilon \mathbb{P}(\mathcal{A}(\mathbf{X}') \in S) + \delta. \tag{12}$$

If (12) holds for all measurable subsets $S$, then we denote this property by $\mathcal{A}(\mathbf{X}) \underset{(\epsilon,\delta)}{\simeq} \mathcal{A}(\mathbf{X}')$. An $R$-round fully interactive randomized algorithm $\mathcal{A}$ for FL is characterized in every round $r \in [R]$ by $N$ local silo functions called *randomizers* $\mathcal{R}_r^{(i)} : \mathcal{Z}^{(r-1) \times N} \times \mathcal{X}_i^n \to \mathcal{Z}$ $(i \in [N])$ and an aggregation mechanism. (See below for an example that further clarifies the terminology used in this paragraph.) The randomizers send messages $Z_r^{(i)} := \mathcal{R}_r^{(i)}(\mathbf{Z}_{1:r-1}, X_i)$, to the server or other silos. The messages

---

[11]Note that in order for their result to be correct, by (Bassily et al., 2014) when $N = M = 1$, their bound must scale at least as $d^2/\epsilon^2 n^2$, unless their bound is trivial ($\geqslant LD$).

$Z_r^{(i)}$ may depend on silo data $X_i$ and the outputs $\mathbf{Z}_{1:r-1} := \{Z_t^{(j)}\}_{j \in [N], t \in [r-1]}$ of silos' randomizers in prior rounds.[12] Then, the server (or silos, for peer-to-peer FL) updates the global model. We consider the output of $\mathcal{A} : \mathbb{X} \to \mathcal{Z}^{R \times N}$ to be the *transcript* of all silos' communications: i.e. the collection of all $N \times R$ messages $\{Z_r^{(i)}\}$. Algorithm $\mathcal{A}$ is $(\epsilon_0, \delta_0)$-ISRL-DP if for all silos $i$, the full transcript $\{Z_r^{(i)}\}_{r \in [R]}$ is $(\epsilon_0, \delta_0)$-DP, for any fixed settings of the other silos' messages and data. More precisely:

**Definition 5.** *(Inter-Silo Record-Level Differential Privacy) Let* $\rho_i(X_i, X_i') := \sum_{j=1}^n \mathbb{1}_{\{x_{i,j} \neq x'_{i,j}\}}$, $i \in [N]$. *A randomized algorithm* $\mathcal{A}$ *is* $(\epsilon_0, \delta_0)$*-ISRL-DP if for all silos* $i$ *and all* $\rho_i$*-adjacent* $X_i, X_i'$,

$$(\mathcal{R}_1^{(i)}(X_i), \mathcal{R}_2^{(i)}(\mathbf{Z}_1, X_i), \cdots, \mathcal{R}_R^{(i)}(\mathbf{Z}_{1:R-1}, X_i)) \underset{(\epsilon_0, \delta_0)}{\simeq} (\mathcal{R}_1^{(i)}(X_i'), \mathcal{R}_2^{(i)}(\mathbf{Z}_1', X_i'), \cdots, \mathcal{R}_R^{(i)}(\mathbf{Z}_{1:R-1}', X_i')),$$

*where* $\mathbf{Z}_r := \{\mathcal{R}_r^{(i)}(\mathbf{Z}_{1:r-1}, X_i)\}_{i=1}^N$ *and* $\mathbf{Z}_r' := \{\mathcal{R}_r^{(i)}(\mathbf{Z}_{1:r-1}', X_i')\}_{i=1}^N$.

**Example clarifying the terminology used in the definition of ISRL-DP given above:** Assume all $M_r = N$ silos are available in every round and consider $\mathcal{A}$ to be the minibatch SGD algorithm, $w_{r+1} := w_r - \eta g_r$, where $g_r = \frac{1}{NK} \sum_{i=1}^N \sum_{j=1}^K \nabla f(w_r, x_{i,j}^r)$ for $\{x_{i,j}^r\}_{j=1}^K$ drawn randomly from $X_i$. Then the *randomizers* $\mathcal{R}_r^{(i)} : \mathcal{Z}^{(r-1) \times N} \times \mathcal{X}_i^n \to \mathcal{Z}$ of silo $i$ are its stochastic gradients: $Z_r^{(i)} = \mathcal{R}_r^{(i)}(\mathbf{Z}_{1:r-1}, X_i) = \frac{1}{K} \sum_{j=1}^K \nabla f(w_r, x_{i,j}^r)$ for $\{x_{i,j}^r\}_{j=1}^K$ drawn randomly from $X_i$. Note that the output of these randomizers depends on $w_r$, which is a function of previous stochastic gradients of all silos $\mathbf{Z}_{1:r-1} = \{Z_t^{(i)}\}_{i \in [N], t \in [r-1]}$. The *aggregation mechanism* outputs $g_r$ by simply averaging the outputs of silos' randomizers: $g_r = \frac{1}{N} \sum_{i=1}^N Z_r^{(i)}$. We view the output of $\mathcal{A} : \mathbb{X} \to \mathcal{Z}^{R \times N}$ to be the *transcript* of all silos' communications, which in this case is the collection of all $N \times R$ stochastic minibatch gradients $\{Z_r^{(i)}\}_{i \in [N], r \in [R]}$. Note that in practice, the algorithm $\mathcal{A}$ does not truly output a list of gradients, but rather outputs $\widehat{w} \in \mathcal{W}$ that is some convex combination of the iterates $\{w_r\}_{r \in [R]}$, which themselves are functions of $\{Z_r^{(i)}\}_{i \in [N], r \in [R]}$. However, by the post-processing property of DP (Dwork & Roth, 2014, Proposition 2.1), the privacy of $\widehat{w}$ will be guaranteed if the silo transcripts are DP. Thus, here we simply consider the output of $\mathcal{A}$ to be the silo transcripts. Clearly, minibatch SGD is not ISRL-DP. To make it ISRL-DP, it would be necessary to introduce additional randomness to make sure that each silo's collection of stochastic gradients is DP, conditional on the messages and data of all other silos. For example, Noisy MB-SGD is a ISRL-DP variation of (projected) minibatch SGD.

## C  RELATIONSHIPS BETWEEN NOTIONS OF DP

### C.1  ISRL-DP IS STRONGER THAN CDP

Assume $\mathcal{A}$ is $(\epsilon_0, \delta_0)$-ISRL-DP. Let $\mathbf{X}, \mathbf{X}'$ be adjacent databases in the CDP sense; i.e. there exists a unique $i \in [N]$, $j \in [n]$ such that $x_{i,j} \neq x'_{i,j}$. Then for all $r \in [R]$, $l \neq i$, $X_l = X_l'$, so the conditional distributions of $\mathcal{R}_r^{(l)}(\mathbf{Z}_{1:r-1}, X_l)$ and $\mathcal{R}_r^{(l)}(\mathbf{Z}_{1:r-1}', X_l')$ given $Z_{1:r-1}^{(l' \neq l)} = z_{1:r-1}^{(l' \neq l)}$ are identical for all $z_{1:r-1}^{(l' \neq l)} \in \mathcal{Z}^{(r-1) \times (N-1)}$. Integrating both sides of this equality with respect to the joint density of $Z_{1:r-1}^{(l' \neq l)}$ shows that $\mathcal{R}_r^{(l)}(\mathbf{Z}_{1:r-1}, X_l) = \mathcal{R}_r^{(l)}(\mathbf{Z}_{1:r-1}', X_l')$ (unconditional equality of distributions). Hence the full transcript of silo $l$ is (unconditionally) $(0, 0)$-CDP for all $l \neq i$. A similar argument (using the inequality (3) instead of equality) shows that silo $i$'s full transcript is unconditionally $(\epsilon_0, \delta_0)$-CDP. Therefore, by the basic composition theorem for DP (Dwork & Roth, 2014), the full combined transcript of all $N$ silos is $(\epsilon_0, \delta_0)$-CDP, which implies that $\mathcal{A}$ is $(\epsilon_0, \delta_0)$-CDP.

Conversely, $(\epsilon, \delta)$-CDP does not imply $(\epsilon', \delta')$-ISRL-DP for any $\epsilon' > 0, \delta' \in (0, 1)$. This is because a CDP algorithm may send non-private updates to the server and rely on the server to randomize, completely violating the requirement of LDP.

---

[12]We assume that $\mathcal{R}_r^{(i)}(\mathbf{Z}_{1:r-1}, X_i)$ is not dependent on $X_j$ ($j \neq i$) given $\mathbf{Z}_{1:r-1}$ and $X_i$; that is, the distribution of $\mathcal{R}_r^{(i)}$ is completely characterized by $\mathbf{Z}_{1:r-1}$ and $X_i$. Therefore, the randomizers of $i$ cannot "eavesdrop" on another silo's data, which aligns with the local data principle of FL. We allow for $Z_t^{(i)}$ to be empty/zero if silo $i$ does not output anything to the server in round $t$.

## C.2 ISRL-DP Implies User-Level DP for small $(\epsilon_0, \delta_0)$

Precisely, we claim that if $\mathcal{A}$ is $(\epsilon_0, \delta_0)$-ISRL-DP then $\mathcal{A}$ is $(n\epsilon, ne^{(n-1)\epsilon}\delta)$ user-level DP; but conversely $(\epsilon, \delta)$-user-level DP does not imply $(\epsilon', \delta')$-ISRL-DP for any $\epsilon' > 0, \delta' \in (0, 1)$. The first part of the claim is due to group privacy (see (Kamath, 2020, Theorem 10) ) and the argument used above in Appendix C.1 to get rid of the "conditional". The second part of the claim is true because a user-level DP algorithm may send non-private updates to the server and rely on the server to randomize, completely violating the requirement of ISRL-DP.

Therefore, if $\epsilon_0 = \mathcal{O}(1/n)$ and $\delta_0 \ll 1/n$, then any $(\epsilon_0, \delta_0)$-ISRL-DP algorithm also provides a strong user-level privacy guarantee.

# D  Proofs and Supplementary Material for Section 2

## D.1  Pseudocode of Noisy MB-SGD

We present pseudocode for Noisy MB-SGD in Algorithm 2:

---
**Algorithm 2** Noisy ISRL-DP MB-SGD

1: **Input:** $N, d, R \in \mathbb{N}, \sigma^2 \geq 0, X_i \in \mathcal{X}_i^n$ for $i \in [N]$, loss function $f(w, x), K \in [n], \{\eta_r\}_{r \in [R]}$ and $\{\gamma_r\}_{r \in [R]}$.
2: Initialize $w_0 \in \mathcal{W}$.
3: **for** $r \in \{0, 1, \cdots, R - 1\}$ **do**
4:     **for** $i \in S_r$ **in parallel do**
5:         Server sends global model $w_r$ to silo $i$.
6:         Silo $i$ draws $K$ samples $x_{i,j}^r$ uniformly from $X_i$ (for $j \in [K]$) and noise $u_i \sim \mathcal{N}(0, \sigma^2 \mathbf{I}_d)$.
7:         Silo $i$ computes $\widetilde{g}_r^i := \frac{1}{K} \sum_{j=1}^{K} \nabla f(w_r, x_{i,j}^r) + u_i$ and sends to server.
8:     **end for**
9:     Server aggregates $\widetilde{g}_r := \frac{1}{M_r} \sum_{i \in S_r} \widetilde{g}_r^i$.
10:     Server updates $w_{r+1} := \Pi_{\mathcal{W}}[w_r - \eta_r \widetilde{g}_r]$.
11: **end for**
12: **Output:** $\widehat{w}_R = \frac{1}{\Gamma_R} \sum_{r=0}^{R-1} \gamma_r w_r$, where $\Gamma_R := \sum_{r=0}^{R-1} \gamma_r$.

---

## D.2  Proof of Theorem 2.1

We begin by proving Theorem 2.1 for $\beta$-smooth $f(\cdot, x)$ and then extend our result to the non-smooth case via Nesterov smoothing (Nesterov, 2005). We will require some preliminaries. We begin with the following definition from (Bousquet & Elisseeff, 2002):

**Definition 6.** *(Uniform Stability) A randomized algorithm $\mathcal{A} : \mathcal{W} \times \mathcal{X}^{\widetilde{N}}$ is said to be $\alpha$-uniformly stable (w.r.t. loss function $f : \mathcal{W} \times \mathcal{X}$) if for any pair of adjacent data sets $\mathbf{X}, \mathbf{X}' \in \mathcal{X}^{\widetilde{N}}, |\mathbf{X}\Delta\mathbf{X}'| \leq 2$, we have*

$$\sup_{x \in \mathcal{X}} \mathbb{E}_{\mathcal{A}}[f(\mathcal{A}(\mathbf{X}), x) - f(\mathcal{A}(\mathbf{X}'), x)] \leq \alpha.$$

In our context, $\widetilde{N} = N \times n$. The following lemma, which is well known, allows us to easily pass from empirical risk to population loss when the algorithm in Question 1s uniformly stable:

**Lemma D.1.** *Let $\mathcal{A} : \mathcal{X}^{\widetilde{N}} \to \mathcal{W}$ be $\alpha$-uniformly stable w.r.t. convex loss function $f : \mathcal{W} \times \mathcal{X} \to \mathbb{R}$. Let $\mathcal{D}$ be any distribution over $\mathcal{X}$ and let $\mathbf{X} \sim \mathcal{D}^{\widetilde{N}}$. Then the excess population loss is upper bounded by the excess expected empirical loss plus $\alpha$:*

$$\mathbb{E}[F(\mathcal{A}(\mathbf{X}), \mathcal{D}) - F^*] \leq \alpha + \mathbb{E}[\widehat{F}_{\mathbf{X}}(\mathcal{A}(\mathbf{X})) - \min_{w \in \mathcal{W}} \widehat{F}_{\mathbf{X}}(w)],$$

*where the expectations are over both the randomness in $\mathcal{A}$ and the sampling of $\mathbf{X} \sim \mathcal{D}^{\widetilde{N}}$. Here we denote the empirical loss by $\widehat{F}_{\mathbf{X}}(w)$ and the population loss by $F(w, \mathcal{D})$ for additional clarity, and $F^* := \min_{w \in \mathcal{W}} F(w, \mathcal{D}) = \min_{w \in \mathcal{W}} F(w)$.*

*Proof.* By Theorem 2.2 in (Hardt et al., 2016),

$$\mathbb{E}[F(\mathcal{A}(\mathbf{X}), \mathcal{D}) - \widehat{F}_{\mathbf{X}}(\mathcal{A}(\mathbf{X}))] \leqslant \alpha.$$

Hence

$$\mathbb{E}[F(\mathcal{A}(\mathbf{X}), \mathcal{D}) - F^*] = \mathbb{E}[F(\mathcal{A}(\mathbf{X}), \mathcal{D}) - \widehat{F}_{\mathbf{X}}(\mathcal{A}(\mathbf{X})) + \widehat{F}_{\mathbf{X}}(\mathcal{A}(\mathbf{X}))$$
$$- \min_{w \in \mathcal{W}} \widehat{F}_{\mathbf{X}}(w) + \min_{w \in \mathcal{W}} \widehat{F}_{\mathbf{X}}(w) - F^*]$$
$$\leqslant \alpha + \mathbb{E}[\widehat{F}_{\mathbf{X}}(\mathcal{A}(\mathbf{X})) - \min_{w \in \mathcal{W}} \widehat{F}_{\mathbf{X}}(w)],$$

since $\mathbb{E} \min_{w \in \mathcal{W}} \widehat{F}_{\mathbf{X}}(w) \leqslant \min_{w \in \mathcal{W}} \mathbb{E}\left[\widehat{F}_{\mathbf{X}}(w, X)\right] = \min_{w \in \mathcal{W}} F(w, \mathcal{D}) = F^*$. $\qquad \square$

The next step is to bound the uniform stability of Algorithm 2.

**Lemma D.2.** *Let $f(\cdot, x)$ be convex, $L$-Lipschitz, and $\beta$-smooth loss for all $x \in \mathcal{X}$. Let $n := \min\{n\}_{i=1}^N$. Then under Assumption 3, Noisy MB-SGD with constant stepsize $\eta \leqslant \frac{1}{\beta}$ and averaging weights $\gamma_r = \frac{1}{R}$ is $\alpha$-uniformly stable with respect to $f$ for $\alpha = \frac{2L^2 R \eta}{nM}$. If, in addition $f(\cdot, x)$ is $\mu$-strongly convex, for all $x \in \mathcal{X}$, then Noisy MB-SGD with constant step size $\eta_r = \eta \leqslant \frac{1}{\beta}$ and any averaging weights $\{\gamma_r\}_{r=1}^R$ is $\alpha$-uniformly stable with respect to $f$ for $\alpha = \frac{4L^2}{\mu(Mn-1)}$ (assuming $\min\{M, n\} > 1$).*

*Proof of Lemma D.2.* The proof of the convex case extends techniques and arguments used in the proofs of (Hardt et al., 2016, Theorem 3.8), (Feldman & Vondrak, 2019, Lemma 4.3), and (Bassily et al., 2019, Lemma 3.4) to the ISRL-DP FL setting; the strongly convex bound requires additional work to get a tight bound. For now, fix the randomness of $\{M_r\}_{r \geqslant 0}$. Let $\mathbf{X}, \mathbf{X}' \in \mathcal{X}^{\widetilde{N}}$ be two data sets, denoted $\mathbf{X} = (X_1, \cdots, X_N)$ for $X_i \in \mathcal{X}^n$ for all $i \in [N]$ and similarly for $\mathbf{X}'$, and assume $|\mathbf{X}\Delta\mathbf{X}'| = 2$. Then there is a unique $a \in [N]$ and $b \in [n]$ such that $x_{a,b} \neq x'_{a,b}$. For $t \in \{0, 1, \cdots, R\}$, denote the $t$-th iterates of Algorithm 2 on these two data sets by $w_t = w_t(\mathbf{X})$ and $w'_t = w_t(\mathbf{X}')$ respectively. We claim that

$$\mathbb{E}\left[\|w_t - w'_t\| \,|\, \{M_r\}_{0 \leqslant r \leqslant t}\right] \leqslant \frac{2L\eta}{n} \sum_{r=0}^{t} \frac{1}{M_r} \tag{13}$$

for all $t$. We prove the claim by induction. It is trivially true when $t = 0$. Suppose (13) holds for all $t \leqslant \tau$. Denote the samples in each local mini-batch at iteration $\tau$ by $\{x_{i,j}\}_{i \in [N], j \in [K]}$ (dropping the $\tau$ for brevity). Assume WLOG that $S_\tau = [M_\tau]$. First condition on the randomness due to minibatch sampling and due to the Gaussian noise, which we denote by $\bar{u} = \frac{1}{M_\tau} \sum_{i \in S_\tau} u_i$. Also, denote (for $t \geqslant 0$) $\tilde{g}_t := \frac{1}{M_t K} \sum_{i \in S_t, j \in [K]} \nabla f(w_t, x_{i,j}) + \bar{u}$ and $\tilde{g}_t := \frac{1}{M_t K} \sum_{i \in S_t, j \in [K]} \nabla f(w'_t, x'_{i,j}) + \bar{u}$. Then by the same argument used in Lemma 3.4 of (Bassily et al., 2019), we can effectively ignore the noise in our analysis of step $\tau + 1$ of the algorithm since the same (conditionally non-random) update is performed on $X$ and $X'$, implying that the noises cancel out. More precisely, by non-expansiveness of projection and gradient descent step for $\eta \leqslant \frac{2}{\beta}$ (see Lemma 3.7 in (Hardt et al., 2016)), we have

$$\|w_{\tau+1} - w'_{\tau+1}\|$$
$$= \left\|\Pi_{\mathcal{W}}(w_\tau - \eta_\tau \tilde{g}_\tau) - \Pi_{\mathcal{W}}(w'_\tau - \eta_\tau \tilde{g}'_\tau)\right\|$$
$$\leqslant \left\|(w_\tau - \eta_\tau \tilde{g}_\tau) - (w'_\tau - \eta_\tau \tilde{g}'_\tau)\right\|$$
$$\leqslant \left\|w_\tau - \eta_\tau \left(\left(\frac{1}{M_\tau K} \sum_{(i,j) \neq (a,b)} \nabla f(w_\tau, x_{i,j})\right) + \bar{u}\right)\right.$$
$$\left. - \left(w'_\tau - \eta_\tau \left(\left(\frac{1}{M_\tau K} \sum_{(i,j) \neq (a,b)} \nabla f(w'_\tau, x_{i,j})\right) + \bar{u}\right)\right)\right\|$$
$$+ \frac{q_\tau \eta_\tau}{M_\tau K} \left\|\nabla f(w_\tau, x_{a,b}) - \nabla f(w'_\tau, x'_{a,b})\right\|$$
$$\leqslant \|w_\tau - w'_\tau\| + \frac{q_\tau \eta_\tau}{M_\tau K} \left\|\nabla f(w_\tau, x_{a,b}) - \nabla f(w'_\tau, x'_{a,b})\right\|,$$

where $q_\tau \in \{0, 1, \cdots, K\}$ is a realization of the random variable $Q_\tau$ that counts the number of times index $b$ occurs in worker $a$'s local minibatch at iteration $\tau$. (Recall that we sample uniformly with replacement.) Now $Q_\tau$ is a sum of $K$ independent Bernoulli($\frac{1}{n_a}$) random variables, hence $\mathbb{E}Q_\tau = \frac{K}{n_a}$. Then using the inductive hypothesis and taking expected value over the randomness of the Gaussian noise and the minibatch sampling proves the claim. Next, taking expectation with respect to the randomness of $\{M_r\}_{r \in [t]}$ implies

$$\mathbb{E}\|w_t - w_t'\| \leqslant \frac{2Lt}{nM},$$

since the $M_r$ are i.i.d. with $\mathbb{E}(\frac{1}{M_1}) = \frac{1}{M}$. Then Jensen's inequality and Lipschitz continuity of $f(\cdot, x)$ imply that for any $x \in \mathcal{X}$,

$$\mathbb{E}[f(\bar{w}_R, x) - f(\bar{w}_R', x')] \leqslant L\mathbb{E}\|\bar{w}_R - \bar{w}_R'\|$$
$$\leqslant \frac{L}{R} \sum_{t=0}^{R-1} \mathbb{E}\|w_t - w_t'\|$$
$$\leqslant \frac{2L^2\eta}{RMn} \frac{R(R+1)}{2} = \frac{L^2\eta(R+1)}{Mn},$$

completing the proof of the convex case.

Next suppose $f$ is $\mu$-strongly convex. The proof begins identically to the convex case. We condition on $M_r$, $u_i$, and $S_r$ as before and (keeping the same notation used there) get for any $r \geqslant 0$

$$\|w_{r+1} - w_{r+1}'\|$$
$$\leqslant \left\| w_r - \eta_r \left( \left( \frac{1}{M_r K} \sum_{(i,j) \neq (a,b)} \nabla f(w_r, x_{i,j}) \right) + \bar{u} \right) \right.$$
$$\left. - \left( w_r' - \eta_r \left( \left( \frac{1}{M_r K} \sum_{(i,j) \neq (a,b)} \nabla f(w_r', x_{i,j}) \right) + \bar{u} \right) \right) \right\|$$
$$+ \frac{q_r \eta_r}{M_r K} \|\nabla f(w_r, x_{a,b}) - \nabla f(w_r', x_{a,b}')\|.$$

We will need the following tighter estimate of the non-expansiveness of the gradient updates to bound the first term on the right-hand side of the inequality above:

**Lemma D.3.** *(Hardt et al., 2016) Let $G : \mathcal{W} \to \mathbb{R}^d$ be $\mu$-strongly convex and $\beta$-smooth. Assume $\eta \leqslant \frac{2}{\beta+\mu}$ Then for any $w, v \in \mathcal{W}$, we have*

$$\|(w - \eta\nabla G(w)) - (v - \eta\nabla G(v))\| \leqslant \left(1 - \frac{\eta\beta\mu}{\beta+\mu}\right) \|v - w\| \leqslant \left(1 - \frac{\eta\mu}{2}\right) \|v - w\|.$$

Note that $G_r(w_r) := \frac{1}{M_r K} \sum_{(i,j) \neq (a,b),(i,j) \in [M_r] \times [K]} f(w_r, x_{i,j}^r)$ is $(1 - \frac{q_r}{M_r K})\beta$-smooth and $(1 - \frac{q_r}{M_r K})\mu$-strongly convex and hence so is $G_r(w_r) + \bar{u}$. Therefore, invoking Lemma D.3 and the assumption $\eta_r = \eta \leqslant \frac{1}{\beta}$, as well as Lipschitzness of $f(\cdot, x) \forall x \in \mathcal{X}$, yields

$$\|w_{r+1} - w_{r+1}'\| \leqslant \left(1 - \frac{\eta\mu(1 - \frac{q_r}{M_r K})}{2}\right) \|w_r - w_r'\| + \frac{2q_r \eta L}{M_r K}.$$

Next, taking expectations over the $M_r$ (with mean $\mathbb{E}(\frac{1}{M_r}) = \frac{1}{M}$), the minibatch sampling (recall $\mathbb{E}q_r = \frac{K}{n_a}$), and the Gaussian noise implies

$$\mathbb{E}\|w_{r+1} - w_{r+1}'\| \leqslant \left(1 - \frac{\eta\mu(1 - \frac{1}{n_a M})}{2}\right) \mathbb{E}\|w_r - w_r'\| + \frac{2\eta L}{n_a M}.$$

One can then prove the following claim by an inductive argument very similar to the one used in the proof of the convex part of Lemma D.2: for all $t \geqslant 0$,

$$\mathbb{E}\|w_t - w_t'\| \leqslant \frac{2\eta L}{n_a M} \sum_{r=0}^{t} (1 - b)^r,$$

where $b := \frac{\mu\eta}{2}\left(\frac{n_a M - 1}{n_a M}\right) < 1$. The above claim implies that

$$\mathbb{E}\|w_t - w_t'\| \leqslant \frac{2\eta L}{n_a M}\left(\frac{1 - (1-b)^{t+1}}{b}\right)$$

$$\leqslant \frac{4L}{\mu(n_a M - 1)}$$

$$\leqslant \frac{4L}{\mu(nM - 1)}.$$

Finally, using the above bound together with Lipschitz continuity of $f$ and Jensen's inequality, we obtain that for any $x \in \mathcal{X}$,

$$\mathbb{E}[f(\widehat{w}_R, x) - f(\widehat{w}_R', x)] \leqslant L\mathbb{E}\|\widehat{w}_R - \widehat{w}_R'\|$$

$$= L\mathbb{E}\left\|\frac{1}{\Gamma_R}\sum_{r=0}^{R-1}\gamma_r(w_r - w_r')\right\|$$

$$\leqslant L\mathbb{E}\left[\frac{1}{\Gamma_R}\sum_{r=0}^{R-1}\gamma_r\|w_r - w_r'\|\right]$$

$$\leqslant L\left[\frac{1}{\Gamma_R}\sum_{r=0}^{R-1}\gamma_r\left(\frac{4L}{\mu(nM-1)}\right)\right]$$

$$= \frac{4L^2}{\mu(nM-1)}.$$

This completes the proof of Lemma D.2. $\qquad\square$

Next, we will bound the empirical loss of Noisy MB-SGD (Algorithm 2). We will need the following two lemmas for the proof of Lemma D.6 (and hence Theorem 2.1):

**Lemma D.4.** *(Projection lemma) Let $\mathcal{W} \subset \mathbb{R}^d$ be a closed convex set. Then $\|\Pi_{\mathcal{W}}(a) - b\|^2 \leqslant \|a - b\|^2$ for any $a \in \mathbb{R}^d, b \in \mathcal{W}$.*

**Lemma D.5.** *(Stich, 2019) Let $b > 0$, let $a, c \geqslant 0$, and $\{\eta_t\}_{t\geqslant 0}$ be non-negative step-sizes such that $\eta_t \leqslant \frac{1}{g}$ for all $t \geqslant 0$ for some parameter $g \geqslant a$. Let $\{r_t\}_{t\geqslant 0}$ and $\{s_t\}_{t\geqslant 0}$ be two non-negative sequences of real numbers which satisfy*

$$r_{t+1} \leqslant (1 - a\eta_t)r_t - b\eta_t s_t + c\eta_t^2$$

*for all $t \geqslant 0$. Then there exist particular choices of step-sizes $\eta_t \leqslant \frac{1}{g}$ and averaging weights $\gamma_t \geqslant 0$ such that*

$$\frac{b}{\Gamma_T}\sum_{t=0}^{T}s_t\gamma_t + ar_{T+1} = \tilde{\mathcal{O}}\left(gr_0\exp\left(\frac{-aT}{g}\right) + \frac{c}{aT}\right),$$

*where $\Gamma_T := \sum_{t=0}^{T}\gamma_t$. In fact, we can choose $\eta_t$ and $\gamma_t$ as follows:*

$$\eta_t = \eta = \min\left\{\frac{1}{g}, \frac{\ln\left(\max\left\{2, a^2 r_0 T^2/c\right\}\right)}{aT}\right\}, \gamma_t = (1 - a\eta)^{-(t+1)}.$$

We give the excess empirical risk guarantee of ISRL-DP MB-SGD below:

**Lemma D.6.** *Let $f : \mathcal{W} \times \mathcal{X} \to \mathbb{R}$ be $\mu$-strongly convex (with $\mu = 0$ for convex case), $L$-Lipschitz, and $\beta$-smooth in $w$ for all $x \in \mathcal{X}$, where $\mathcal{W}$ is a closed convex set in $\mathbb{R}^d$ s.t. $\|w - w'\| \leqslant D$ for all $w, w' \in \mathcal{W}$. Let $\mathbf{X} \in \mathbb{X}$. Then Noisy MB-SGD (Algorithm 2) with $\sigma^2 = \frac{256 L^2 R \ln(\frac{2.5R}{\delta_0})\ln(2/\delta_0)}{n^2\epsilon_0^2}$ attains the following empirical loss bounds as a function of step size and the number of rounds:*
*1. (Convex) For any $\eta \leqslant 1/\beta$ and $R \in \mathbb{N}$, $\gamma_r := 1/R$, we have*

$$\mathbb{E}\widehat{F}_{\mathbf{X}}(\widehat{w}_R) - \widehat{F}_{\mathbf{X}}^* \leqslant \frac{D^2}{2\eta R} + \frac{\eta L^2}{2}\left(\frac{256 dR\ln(\frac{2.5R}{\delta_0})\ln(2/\delta_0)}{Mn^2\epsilon_0^2} + 1\right).$$

2.  *(Strongly Convex) There exists a constant stepsize $\eta_r = \eta \leqslant 1/\beta$ such that if $R \geqslant 2\kappa \ln \left( \frac{\mu M \epsilon_0^2 n^2 \beta D}{L^2 d} \right)$, then*

$$\mathbb{E}\widehat{F}_{\mathbf{X}}(\widehat{w}_R) - \widehat{F}_{\mathbf{X}}^* = \widetilde{\mathcal{O}}\left( \frac{L^2}{\mu} \left( \frac{d \ln(1/\delta_0)}{M\epsilon_0^2 n^2} \right) + \frac{1}{R} \right). \tag{14}$$

*Proof.* First, condition on the random $M_r$ and consider $M_r$ as fixed. Let $w^* \in \operatorname{argmin}_{w \in \mathcal{W}} \widehat{F}_{\mathbf{X}}(w)$ be any minimizer of $\widehat{F}_{\mathbf{X}}$, and denote the average of the i.i.d. Gaussian noises across all silos in one round by $\bar{u}_r := \frac{1}{M_r} \sum_{i \in S_r} u_i$. Note that $\bar{u}_r \sim N\left(0, \frac{\sigma^2}{M_r} \mathbf{I}_d\right)$ by independence of the $\{u_i\}_{i \in [N]}$ and hence $\mathbb{E}\|\bar{u}_r\|^2 = \frac{d\sigma^2}{M_r}$. Then for any $r \geqslant 0$, conditional on $M_r$, we have that

$$
\begin{aligned}
&\mathbb{E}\left[ \left\| w_{r+1} - w^* \right\|^2 \Big| M_r \right] \\
={}& \mathbb{E}\left[ \left\| \Pi_{\mathcal{W}} \left[ w_r - \eta_r \left( \frac{1}{M_r} \sum_{i \in S_r} \frac{1}{K} \sum_{j=1}^{K} \nabla f(w_r, x_{i,j}^r) - u_i \right) \right] - w^* \right\|^2 \Big| M_r \right] \\
\leqslant{}& \mathbb{E}\left[ \left\| w_r - \eta_r \left( \frac{1}{M_r} \sum_{i \in S_r} \frac{1}{K} \sum_{j=1}^{K} \nabla f(w_r, x_{i,j}^r) - u_i \right) - w^* \right\|^2 \Big| M_r \right] \\
={}& \mathbb{E}\left[ \left\| w_r - w^* \right\|^2 \Big| M_r \right] - 2\eta_r \mathbb{E}\left[ \langle \nabla \widehat{F}_{\mathbf{X}}(w_r) + \bar{u}_r, w_r - w^* \rangle \Big| M_r \right] \\
&+ \eta_r^2 \mathbb{E}\left[ \left\| \bar{u}_r + \frac{1}{M_r} \sum_{i \in S_r} \frac{1}{K} \sum_{j=1}^{K} \nabla f(w_r, x_{i,j}^r) \right\|^2 \Big| M_r \right] \\
\leqslant{}& (1 - \mu\eta_r) \mathbb{E}\left[ \left\| w_r - w^* \right\|^2 \Big| M_r \right] - 2\eta_r \mathbb{E}[\widehat{F}_{\mathbf{X}}(w_r) - \widehat{F}_{\mathbf{X}}^* | M_r] \\
&+ \eta_r^2 \mathbb{E}\left[ \left\| \bar{u}_r + \frac{1}{M_r} \sum_{i \in S_r} \frac{1}{K} \sum_{j=1}^{K} \nabla f(w_r, x_{i,j}^r) \right\|^2 \Big| M_r \right] \\
\leqslant{}& (1 - \mu\eta_r) \mathbb{E}\left[ \left\| w_r - w^* \right\|^2 \Big| M_r \right] - 2\eta_r \mathbb{E}[\widehat{F}_{\mathbf{X}}(w_r) - \widehat{F}_{\mathbf{X}}^* | M_r] + \eta_r^2 \left( \frac{d\sigma^2}{M_r} + L^2 \right), \tag{15}
\end{aligned}
$$

where we used Lemma D.4 in the first inequality, $\mu$-strong convexity of $\widehat{F}$ (for $\mu \geqslant 0$) and the fact that $\bar{u}_r$ is independent of the gradient estimate and mean zero in the next inequality, and the fact that $f(\cdot, x)$ is $L$-Lipschitz for all $x$ in the last inequality (together with independence of the noise and the data again). Now we consider the convex ($\mu = 0$) and strongly convex ($\mu > 0$) cases separately.

**Convex ($\mu = 0$) case:** Re-arranging (15), we get

$$\mathbb{E}[\widehat{F}_{\mathbf{X}}(w_r) - \widehat{F}_{\mathbf{X}}^* | M_r] \leqslant \frac{1}{2\eta_r} \left( \mathbb{E}[\|w_r - w^*\|^2 - \|w_{r+1} - w^*\|^2] \right) + \frac{\eta_r}{2} \left( \frac{d\sigma^2}{M_r} + L^2 \right)$$

and hence

$$\mathbb{E}[\widehat{F}_{\mathbf{X}}(w_r) - \widehat{F}_{\mathbf{X}}^*] \leqslant \frac{1}{2\eta_r} \left( \mathbb{E}[\|w_r - w^*\|^2 - \|w_{r+1} - w^*\|^2] \right) + \frac{\eta_r}{2} \left( \frac{d\sigma^2}{M} + L^2 \right)$$

by taking total expectation and using $\mathbb{E}[1/M_r] = 1/M$. Then for $\eta_r = \eta$, the average iterate $\bar{w}_R$ satisfies:

$$\mathbb{E}[\widehat{F}_{\mathbf{X}}(\bar{w}_R) - \widehat{F}_{\mathbf{X}}^*] \leqslant \frac{1}{R} \sum_{r=0}^{R-1} \mathbb{E}[\widehat{F}_{\mathbf{X}}(w_r) - \widehat{F}_{\mathbf{X}}^*]$$

$$\leqslant \frac{1}{R} \sum_{r=0}^{R-1} \frac{1}{2\eta} (\mathbb{E}[\|w_r - w^*\|^2 - \|w_{r+1} - w^*\|])$$

$$+ \frac{\eta}{2} \left( \frac{d\sigma^2}{M} + L^2 \right)$$

$$\leqslant \frac{\|w_0 - w^*\|^2}{\eta R} + \frac{\eta_r}{2} \left( \frac{d\sigma^2}{M} + L^2 \right).$$

Plugging in $\sigma^2$ finishes the proof of the convex case.

**Strongly convex ($\mu > 0$) case:** Recall from (15) that

$$\mathbb{E}[\|w_{t+1} - w^*\|^2] \leqslant (1 - \mu\eta_t)\mathbb{E}[\|w_t - w^*\|^2] - 2\eta_t \mathbb{E}[\widehat{F}_{\mathbf{X}}(w_t) - \widehat{F}_{\mathbf{X}}^*]$$

$$+ \eta_t^2 \left( \frac{d\sigma^2}{M} + L^2 \right) \tag{16}$$

for all $t \geqslant 0$ (upon taking expectation over $M_t$). Now, (16) satisfies the conditions for Lemma D.5, with sequences

$$r_t = \mathbb{E}\|w_t - w^*\|^2, s_t = \mathbb{E}[\widehat{F}_{\mathbf{X}}(w_t) - \widehat{F}_{\mathbf{X}}^*]$$

and parameters

$$a = \mu, \; b = 2, \; c = \frac{d\sigma^2}{M} + L^2, \; g = 2\beta, \; T = R.$$

Then Lemma D.5 and Jensen's inequality imply

$$\mathbb{E}\widehat{F}_{\mathbf{X}}(\widehat{w}_R) - \widehat{F}_{\mathbf{X}}^* = \widetilde{\mathcal{O}} \left( \beta D^2 \exp \left( \frac{-R}{2\kappa} \right) + \frac{L^2}{\mu} \left( \frac{1}{R} + \frac{d}{M\epsilon_0^2 n^2} \right) \right).$$

Finally, plugging in $R$ and $\sigma^2$ completes the proof. $\qquad\square$

We are now prepared to prove Theorem 2.1 in the $\beta$-smooth case:

**Theorem D.1.** *Let $f(w, x)$ be $\beta$-smooth in $w$ for all $x \in \mathcal{X}$. Assume $\epsilon_0 \leqslant 2\ln(2/\delta_0)$, choose $\sigma^2 = \frac{256L^2 R \ln(\frac{2.5R}{\delta_0}) \ln(2/\delta_0)}{n^2 \epsilon_0^2}$ and $K \geqslant \frac{\epsilon_0 n}{4\sqrt{2R\ln(2/\delta_0)}}$. Then Algorithm 2 is $(\epsilon_0, \delta_0)$-ISRL-DP. Moreover, there are choices of $\{\eta_r\}_{r=1}^R$ such that Algorithm 2 achieves the following excess loss bounds:*

*1. If $f(\cdot, x)$ is convex, then setting $R = \frac{\beta D\sqrt{M}}{L} \min \left\{ \sqrt{n}, \frac{\epsilon_0 n}{\sqrt{d}} \right\} + \min \left\{ nM, \frac{\epsilon_0^2 n^2 M}{d} \right\}$ yields*

$$\mathbb{E}F(\widehat{w}_R) - F^* = \widetilde{\mathcal{O}} \left( \frac{LD}{\sqrt{M}} \left( \frac{1}{\sqrt{n}} + \frac{\sqrt{d\ln(1/\delta_0)}}{\epsilon_0 n} \right) \right). \tag{17}$$

*2. If $f(\cdot, x)$ is $\mu$-strongly convex, then setting $R = \max \left( \frac{2\beta}{\mu} \ln \left( \frac{\beta D^2 \mu M \epsilon_0^2 n^2}{dL^2} \right), \min \left\{ Mn, \frac{M\epsilon_0^2 n^2}{d} \right\} \right)$ yields*

$$\mathbb{E}F(\widehat{w}_R) - F^* = \widetilde{\mathcal{O}} \left( \frac{L^2}{\mu M} \left( \frac{1}{n} + \frac{d\ln(1/\delta_0)}{\epsilon_0^2 n^2} \right) \right). \tag{18}$$

*Proof.* **Privacy:** By independence of the Gaussian noise across silos, it suffices to show that transcript of silo $i$'s interactions with the server is DP for all $i \in [N]$ (conditional on the transcripts of all other silos). WLOG consider $i = 1$. By the advanced composition theorem (Theorem 3.20 in (Dwork & Roth, 2014)), it suffices to show that each of the $R$ rounds of the algorithm is $(\widetilde{\epsilon}, \widetilde{\delta})$-ISRL-DP, where $\widetilde{\epsilon} = \frac{\epsilon_0}{2\sqrt{2R\ln(2/\delta_0)}}$ (we used the assumption $\epsilon_0 \leqslant 2\ln(2/\delta_0)$ here) and $\widetilde{\delta} = \frac{\delta_0}{2R}$. First, condition on the randomness due to local sampling of the local data point

$x_{1,1}^r$ (line 4 of Algorithm 2). Now, the $L_2$ sensitivity of each local step of SGD is bounded by $\Delta := \sup_{|X_1 \Delta X_1'| \leqslant 2, w \in \mathcal{W}} \| \frac{1}{K} \sum_{j=1}^{K} \nabla f(w, x_{1,j}) - \nabla f(w, x_{1,j}') \| \leqslant 2L/K$, by $L$-Lipschitzness of $f$. Thus, the standard privacy guarantee of the Gaussian mechanism (see (Dwork & Roth, 2014, Theorem A.1)) implies that (conditional on the randomness due to sampling) taking $\sigma_1^2 \geqslant \frac{8L^2 \ln(1.25/\tilde{\delta})}{\tilde{\epsilon}^2 K^2}$ suffices to ensure that round $r$ (in isolation) is $(\tilde{\epsilon}, \tilde{\delta})$-ISRL-DP. Now we invoke the randomness due to sampling: (Ullman, 2017) implies that round $r$ (in isolation) is $(\frac{2\tilde{\epsilon}K}{n}, \tilde{\delta})$-ISRL-DP. The assumption on $K$ ensures that $\epsilon' := \frac{n}{2K} \frac{\epsilon_0}{2\sqrt{2R\ln(2/\delta_0)}} \leqslant 1$, so that the privacy guarantees of the Gaussian mechanism and amplification by subsampling stated above indeed hold. Therefore, with sampling, it suffices to take $\sigma^2 \geqslant \frac{32L^2 \ln(1.25/\tilde{\delta})}{n^2 \tilde{\epsilon}^2} = \frac{256L^2 R \ln(2.5R/\delta_0) \ln(2/\delta_0)}{n^2 \epsilon_0^2}$ to ensure that round $r$ (in isolation) is $(\tilde{\epsilon}, \tilde{\delta})$-ISRL-DP for all $r$ and hence that the full algorithm ($R$ rounds) is $(\epsilon_0, \delta_0)$-ISRL-DP.

**Excess loss:** 1. First suppose $f$ is merely convex ($\mu = 0$). By Lemma D.2, Lemma D.1, and Lemma D.6, we have:

$$\mathbb{E}F(\hat{w}_R) - F^* \leqslant \alpha + \mathbb{E}\hat{F}_{\mathbf{X}}(\hat{w}_R) - \hat{F}_{\mathbf{X}}^*$$

$$\leqslant \frac{2L^2 R\eta}{nM} + \frac{D^2}{2\eta R} + \frac{\eta L^2}{2M} \left( \frac{256dR\ln(\frac{2.5R}{\delta_0})\ln(2/\delta_0)}{n^2 \epsilon_0^2} + 1 \right)$$

for any $\eta \leqslant 1/\beta$. Choosing $\eta = \min\left(\frac{1}{\beta}, \frac{D}{L\sqrt{R}} \min\left\{1, \frac{\epsilon_0 n\sqrt{M}}{\sqrt{Rd}}, \sqrt{\frac{nM}{R}}\right\}\right)$ implies

$$\mathbb{E}F(\hat{w}_R) - F^* \lesssim \frac{\beta D^2}{R} + \frac{LD}{\sqrt{R}} \left( 1 + \frac{\sqrt{dR\ln(R/\delta_0)}}{\epsilon_0 n\sqrt{M}} + \sqrt{\frac{R}{nM}} \right).$$

Finally, one verifies that the prescribed choice of $R$ yields (17).
2. Now suppose $f$ is $\mu$-strongly convex. Then for the $\eta \leqslant 1/\beta$ used in the proof of Lemma D.6 and $R \geqslant \frac{2\beta}{\mu} \ln\left(\frac{\beta D^2 \mu M \epsilon_0^2 n^2}{dL^2}\right)$, we have

$$\mathbb{E}\hat{F}_{\mathbf{X}}(\hat{w}_R) - \hat{F}_{\mathbf{X}}^* = \tilde{\mathcal{O}}\left( \frac{L^2}{\mu} \left( \frac{d\ln(1/\delta_0)}{M\epsilon_0^2 n^2} + \frac{1}{R} + \frac{1}{Mn} \right) \right),$$

by Lemma D.6, Lemma D.2, and Lemma D.1. Hence (18) follows by our choice of $R \geqslant \min\left(Mn, \frac{M\epsilon_0^2 n^2}{d}\right)$. $\qquad\qquad\square$

We use Theorem D.1 to prove Theorem 2.1 via Nesterov smoothing (Nesterov, 2005), similar to how (Bassily et al., 2019) proceeded for CDP SCO with non-strongly convex loss and $N = 1$. That is, for non-smooth $f$, we run ISRL-DP Noisy MB-SGD on the smoothed objective (a.k.a. $\beta$-Moreau envelope) $f_\beta(w) := \min_{v \in \mathcal{W}} \left( f(v) + \frac{\beta}{2}\|w - v\|^2 \right)$, where $\beta > 0$ is a design parameter that we will optimize for. The following key lemma allows us to easily extend Theorem D.1 to non-smooth $f$:

**Lemma D.7** (Nesterov (2005)). *Let $f : \mathcal{W} \to \mathbb{R}^d$ be convex and $L$-Lipschitz and let $\beta > 0$. Then the $\beta$-Moreau envelope $f_\beta(w) := \min_{v \in \mathcal{W}} \left( f(v) + \frac{\beta}{2}\|w - v\|^2 \right)$ satisfies:*
*1. $f_\beta$ is convex, $2L$-Lipschitz, and $\beta$-smooth.*
*2. $\forall w, f_\beta(w) \leqslant f(w) \leqslant f_\beta(w) + \frac{L^2}{2\beta}$.*

Now let us re-state the precise version of Theorem 2.1 before providing its proof:

**Theorem D.2** (Precise version of Theorem 2.1). *Let $\epsilon_0 \leqslant 2\ln(2/\delta_0)$ and choose $\sigma^2 = \frac{256L^2 R \ln(\frac{2.5R}{\delta_0})\ln(2/\delta_0)}{n^2 \epsilon_0^2}$, $K \geqslant \frac{\epsilon_0 n}{4\sqrt{2R\ln(2/\delta_0)}}$. Then Algorithm 2 is $(\epsilon_0, \delta_0)$-ISRL-DP. Further, there exist choices of $\beta > 0$ such that running Algorithm 2 on $f_\beta(w, x) := \min_{v \in \mathcal{W}} \left( f(v, x) + \frac{\beta}{2}\|w - v\|^2 \right)$ yields:*

*1. If $f(\cdot, x)$ is convex, then setting $R = \frac{\beta D\sqrt{M}}{L} \min\left\{\sqrt{n}, \frac{\epsilon_0 n}{\sqrt{d}}\right\} + \min\left\{nM, \frac{\epsilon_0^2 n^2 M}{d}\right\}$ yields*

$$\mathbb{E}F(\hat{w}_R) - F^* = \tilde{\mathcal{O}}\left( \frac{LD}{\sqrt{M}} \left( \frac{1}{\sqrt{n}} + \frac{\sqrt{d\ln(1/\delta_0)}}{\epsilon_0 n} \right) \right). \qquad (19)$$

2. *If $f(\cdot, x)$ is $\mu$-strongly convex, then setting $R = \max\left(\frac{2\beta}{\mu}\ln\left(\frac{\beta D^2 \mu M \epsilon_0^2 n^2}{dL^2}\right), \min\left\{Mn, \frac{M\epsilon_0^2 n^2}{d}\right\}\right)$ yields*

$$\mathbb{E}F(\widehat{w}_R) - F^* = \widetilde{\mathcal{O}}\left(\frac{L^2}{\mu M}\left(\frac{1}{n} + \frac{d\ln(1/\delta_0)}{\epsilon_0^2 n^2}\right)\right). \tag{20}$$

*Proof.* **Privacy:** ISRL-DP is immediate from post-processing (Dwork & Roth, 2014, Proposition 2.1), since we already showed that Algorithm 2 (applied to $f_\beta$) is ISRL-DP.

**Excess risk:** We have $\mathbb{E}F(\widehat{w}_R) - F^* \leq \mathbb{E}F_\beta(\widehat{w}_R) - F_\beta^* + \frac{L^2}{2\beta}$, by part 2 of Lemma D.7. Moreover, by part 1 of Lemma D.7 and Theorem D.1, we have:

1. $\mathbb{E}F_\beta(\widehat{w}_R) - F_\beta^* = \widetilde{\mathcal{O}}\left(\frac{LD}{\sqrt{M}}\left(\frac{1}{\sqrt{n}} + \frac{\sqrt{d\ln(1/\delta_0)}}{\epsilon_0 n}\right)\right)$

for convex $f$, and

2. $\mathbb{E}F_\beta(\widehat{w}_R) - F_\beta^* = \widetilde{\mathcal{O}}\left(\frac{L^2}{\mu M}\left(\frac{1}{n} + \frac{d\ln(1/\delta_0)}{\epsilon_0^2 n^2}\right)\right),$

for $\mu$-strongly convex $f$. Thus, choosing $\beta_1$ such that $L^2/\beta_1 \leq \frac{LD}{\sqrt{M}}\left(\frac{1}{\sqrt{n}} + \frac{\sqrt{d\ln(1/\delta_0)}}{\epsilon_0 n}\right)$ and $\beta_2$ such that $L^2/\beta_2 \leq \frac{L^2}{\mu M}\left(\frac{1}{n} + \frac{d\ln(1/\delta_0)}{\epsilon_0^2 n^2}\right)$ completes the proof. $\square$

### D.3  LOWER BOUNDS FOR ISRL-DP FL: SUPPLEMENTAL MATERIAL AND PROOFS

This section requires familiarity with the notation introduced in the rigorous definition of ISRL-DP in Appendix B.

#### D.3.1  $C$-COMPOSITIONALITY

The $(\epsilon_0, \delta_0)$-ISRL-DP algorithm class $\mathbb{A}_{(\epsilon_0, \delta_0), C}$ contains all sequentially interactive and fully interactive, $C$-*compositional* algorithms.

**Definition 7** (Compositionality). *Let $\mathcal{A}$ be an $R$-round $(\epsilon_0, \delta_0)$-ISRL-DP FL algorithm with data domain $\mathcal{X}$. Let $\{(\epsilon_0^r, \delta_0^r)\}_{r=1}^R$ denote the minimal (non-negative) parameters of the local randomizers $\mathcal{R}_r^{(i)}$ selected at round $r$ such that $\mathcal{R}_r^{(i)}(\mathbf{Z}_{(1:r-1)}, \cdot)$ is $(\epsilon_0^r, \delta_0^r)$-DP for all $i \in [N]$ and all $\mathbf{Z}_{(1:r-1)}$. For $C > 0$, we say that $\mathcal{A}$ is $C$-compositional if $\sqrt{\sum_{r \in [R]}(\epsilon_0^r)^2} \leq C\epsilon_0$. If such $C$ is an absolute constant, we simply say $\mathcal{A}$ is compositional.*

Definition 7 is an extension of the definition in Joseph et al. (2019) to $\delta_0 > 0$.

#### D.3.2  ALGORITHMS WHOSE PRIVACY FOLLOWS FROM ADVANCED COMPOSITION THEOREM ARE 1-COMPOSITIONAL

Suppose $\mathcal{A}$ is $(\epsilon_0, \delta_0)$-ISRL-DP by the advanced composition theorem (Dwork & Roth, 2014, Theorem 3.20). Then

$$\epsilon_0 = \sqrt{2\sum_{r=1}^R (\epsilon_0^r)^2 \ln(1/\delta')} + \sum_{r=1}^R \epsilon_0^r (e^{\epsilon_0^r} - 1),$$

and $\delta_0 = \sum_{r=1}^R \delta_0^r + \delta'$ for any $\delta' \in (0, 1)$. Assume $\delta' \in (0, 1/3)$ without loss of generality: otherwise the privacy guarantee of $\mathcal{A}$ is essentially meaningless (see Remark D.1). Then $\epsilon_0 \geq \sqrt{2\sum_{r=1}^R (\epsilon_0^r)^2 \ln(1/\delta')} \geq \sqrt{2}\sqrt{\sum_{r=1}^R (\epsilon_0^r)^2} \geq \sqrt{\sum_{r=1}^R (\epsilon_0^r)^2}$, so that $\mathcal{A}$ is 1-compositional. Note that even if $\delta' > 1/3$, $\mathcal{A}$ would still be compositional, but the constant $C$ may be larger than 1.

### D.3.3 EXAMPLE OF LDP ALGORITHM THAT IS NOT COMPOSITIONAL

This example is a simple modification of Example 2.2 in (Joseph et al., 2019) (adapted to our definition of compositionality for $\delta_0 > 0$). Given any $C > 0$, set $d := 2C^2$ and let $\mathcal{X} = \{e_1, \cdots e_d\} \subset \{0, 1\}^d$ be the standard basis for $\mathbb{R}^d$. Let $n = 1$ and $\mathbf{X} = (x_1, \cdots, x_N) \in \mathcal{X}^N$. For all $i \in [N]$ let $\mathcal{Q}^{(i)} : \mathcal{X} \to \mathcal{X}$ be the randomized response mechanism that outputs $\mathcal{Q}^{(i)}(x_i) = x_i$ with probability $\frac{e^{\epsilon_0}}{e^{\epsilon_0} + d - 1}$ and otherwise outputs a uniformly random element of $\mathcal{X} \setminus \{x_i\}$. Note that $\mathcal{Q}^{(i)}$ is $\epsilon_0$-DP, hence $(\epsilon_0, \delta_0)$-DP for any $\delta_0 > 0$. Consider the $d$-round algorithm $\mathcal{A} : \mathcal{X}^N \to \mathcal{Z}^{d \times N}$ in Algorithm 3, where $\mathcal{Z} = \mathbb{R}^d$

---

**Algorithm 3** LDP Algorithm that is not $C$-compositional

1: **for** $r \in [d]$ **do**
2:  **for** $i \in N$ **do**
3:   **if** $x_i = e_r$ **then**
4:    $\mathcal{R}_r^{(i)}(x_i) := \mathcal{Q}^{(i)}(x_i)$.
5:   **else**
6:    $\mathcal{R}_r^{(i)}(x_i) := 0 \in \mathbb{R}^d$.
7:   **end if**
8:  **end for**
9: **end for**
10: **Output:** $\{\mathcal{R}_r^{(i)}(x_i)\}_{i \in [N], r \in [d]}$.

---

Since each silo's data is only referenced once and $\mathcal{Q}^{(i)}$ is $\epsilon_0$-DP, we have $\epsilon_0^r = \epsilon_0$ and $\mathcal{A}$ is $(\epsilon_0, \delta_0)$-DP. However, $\sqrt{\sum_{r=1}^d (\epsilon_0^r)^2} = \sqrt{d\epsilon_0^2} = \sqrt{2}C\epsilon_0 > C\epsilon_0$, so that $\mathcal{A}$ is not $C$-compositional.

Also, note that our One-Pass Accelerated Noisy MB-SGD is only $C$-compositional for $C \geqslant \sqrt{R}$, since $\epsilon_0 = \epsilon_0^r$ for this algorithm. Thus, substituting the $R$ that is used to prove the upper bounds in Theorem 3.1 (see Appendix E.2) and plugging $C = \sqrt{R}$ into Theorem 2.2 explains where the non-i.i.d. lower bounds in Fig. 3 come from.

### D.3.4 PROOF OF THEOREM 2.2

First, let us state the complete, formal version of Theorem 2.2, which uses notation from Appendix B:

**Theorem D.3** (Complete Version of Theorem 2.2). *Let $\epsilon_0 \in (0, \sqrt{N}]$, $\delta_0 = o(1/nN)$, and $\mathcal{A} \in \mathbb{A}_{(\epsilon_0, \delta_0), C}$. Suppose that in each round $r$, the local randomizers are all $(\epsilon_0^r, \delta_0^r)$-DP, for $\epsilon_0^r \lesssim \frac{1}{n}$, $\delta_0^r = o(1/nNR)$, $N \geqslant 16 \ln(2/\delta_0^r n)$. Then:*
*1. There exists a $f \in \mathcal{F}_{L,D}$ and a distribution $\mathcal{D}$ such that for $\mathbf{X} \sim \mathcal{D}^{nN}$, we have:*

$$\mathbb{E}F(\mathcal{A}(\mathbf{X})) - F^* = \widetilde{\Omega}\left(\frac{\phi D}{\sqrt{Nn}} + LD\min\left\{1, \frac{\sqrt{d}}{\epsilon_0 n \sqrt{N} C^2}\right\}\right).$$

*2. There exists a $\mu$-smooth $f \in \mathcal{G}_{\mu, L, D}$ and distribution $\mathcal{D}$ such that for $\mathbf{X} \sim \mathcal{D}^{nN}$, we have*

$$\mathbb{E}F(\mathcal{A}(\mathbf{X})) - F^* = \widetilde{\Omega}\left(\frac{\phi^2}{\mu nN} + LD\min\left\{1, \frac{d}{\epsilon_0^2 n^2 N C^4}\right\}\right).$$

*Further, if $\mathcal{A} \in \mathbb{A}$, then the above lower bounds hold with $C = 1$.*

Before we proceed to the proof of Theorem 2.2, we recall the simpler characterization of ISRL-DP for sequentially interactive algorithms. A sequentially interactive algorithm $\mathcal{A}$ with randomizers $\{\mathcal{R}^{(i)}\}_{i=1}^N$ is $(\epsilon_0, \delta_0)$-ISRL-DP if and only if for all $i \in [N]$, $\mathcal{R}^{(i)}(\cdot, Z^{(1:i-1)}) : \mathcal{X}_i^n \times \mathcal{Z}$ is $(\epsilon_0, \delta_0)$-DP for all $Z^{(1:i-1)} \in \mathcal{Z}^{i-1}$. In what follows, we will fix $\mathcal{X}_i = \mathcal{X}$ for all $i$. We now turn to the proof.

**Step 1: Privacy amplification by shuffling.** We begin by stating and proving the amplification by shuffling result that we will leverage to obtain Theorem 2.2:

**Theorem D.4.** *Let $\mathcal{A} \in \mathbb{A}_{(\epsilon_0,\delta_0),C}$ such that $\epsilon_0 \in (0,\sqrt{N}]$ and $\delta_0 \in (0,1)$. Assume that in each round, the local randomizers $\mathcal{R}_r^{(i)}(\mathbf{Z}_{(1:r-1)}, \cdot) : \mathcal{X}^n \to \mathcal{Z}$ are $(\epsilon_0^r, \delta_0^r)$-DP for all $i \in [N]$, $r \in [R]$, $\mathbf{Z}_{(1:r-1)} \in \mathcal{Z}^{r-1 \times N}$ with $\epsilon_0^r \leqslant \frac{1}{n}$. Assume $N \geqslant 16 \ln(2/\delta_0^r n)$. If $\mathcal{A}$ is $C$-compositional, then assume $\delta_0^r \leqslant \frac{1}{14nNR}$ and denote $\delta := 14Nn \sum_{r=1}^R \delta_0^r$; if instead $\mathcal{A}$ is sequentially interactive, then assume $\delta_0 = \delta_0^r \leqslant \frac{1}{7Nn}$ and denote $\delta := 7Nn\delta_0$. Let $\mathcal{A}_s : \mathbb{X} \to \mathcal{W}$ be the same algorithm as $\mathcal{A}$ except that in each round $r$, $\mathcal{A}_s$ draws a random permutation $\pi_r$ of $[N]$ and applies $\mathcal{R}_r^{(i)}$ to $X_{\pi_r(i)}$ instead of $X_i$. Then, $\mathcal{A}_s$ is $(\epsilon, \delta)$-CDP, where $\epsilon = \mathcal{O}\left(\frac{\epsilon_0 \ln\left(1/nN\delta_0^{\min}\right)C^2}{\sqrt{N}}\right)$, and $\delta_0^{\min} := \min_{r \in [R]} \delta_0^r$.*

*In particular, if $\mathcal{A} \in \mathbb{A}$, then $\epsilon = \mathcal{O}\left(\frac{\epsilon_0 \ln\left(1/nN\delta_0^{\min}\right)}{\sqrt{N}}\right)$. Note that for sequentially interactive $\mathcal{A}$, $\delta_0^{\min} = \delta_0$.*

To the best of our knowledge, the restriction on $\epsilon_0^r$ is needed to obtain $\epsilon = \widetilde{O}(\epsilon_0/\sqrt{N})$ in all works that have analyzed privacy amplification by shuffling (Erlingsson et al., 2020b; Feldman et al., 2020b; Balle et al., 2019; Cheu et al., 2019; Balle et al., 2020), but these works focus on the sequentially interactive case with $n = 1$, so the restriction amounts to $\epsilon_0 \lesssim 1$ (or $\epsilon_0 = \widetilde{O}(1)$). The non-sequential $C$-compositional part of Theorem D.4 will follow as a corollary (Corollary D.1) of the following result which analyzes the privacy amplification in each round:

**Theorem D.5** (Single round privacy amplification by shuffling)**.** *Let $\epsilon_0^r \leqslant \ln\left(\frac{N}{16 \ln(2/\delta^r)}\right)/n$, $r \in \mathbb{N}$ and let $\mathcal{R}_r^{(i)}(\mathbf{Z}, \cdot) : \mathcal{X}^n \to \mathcal{Z}$ be an $(\epsilon_0^r, \delta_0^r)$-DP local randomizer for all $\mathbf{Z} = Z_{(1:r-1)}^{(1:N)} \in \mathcal{Z}^{(r-1) \times N}$ and $i \in [N]$, where $\mathcal{X}$ is an arbitrary set. Given a distributed data set $\mathbf{X} = (X_1, \cdots, X_N) \in \mathcal{X}^{N \times n}$ and $\mathbf{Z} = Z_{(1:r-1)}^{(1:N)}$, consider the shuffled algorithm $\mathcal{A}_s^r : \mathcal{X}^{n \times N} \times \mathcal{Z}^{(r-1) \times N} \to \mathcal{Z}^N$ that first samples a random permutation $\pi$ of $[N]$ and then computes $Z_r = (Z_r^{(1)}, \cdots, Z_r^{(N)})$, where $Z_r^{(i)} := \mathcal{R}_r^{(i)}(\mathbf{Z}, X_{\pi(i)})$. Then, $\mathcal{A}_s^r$ is $(\epsilon^r, \widetilde{\delta^r})$-CDP, where*

$$\epsilon^r := \ln\left[1 + \left(\frac{e^{\epsilon_0^r} - 1}{e^{\epsilon_0^r} + 1}\right)\left(\frac{8\sqrt{e^{n\epsilon_0^r}\ln(4/\delta^r)}}{\sqrt{N}} + \frac{8e^{n\epsilon_0^r}}{N}\right)\right], \tag{21}$$

*and $\widetilde{\delta^r} := \delta^r + 2Nne^{(n-1)\epsilon_0^r}\delta_0^r$. In particular, if $\epsilon_0^r = \mathcal{O}\left(\frac{1}{n}\right)$, then*

$$\epsilon^r = \mathcal{O}\left(\frac{\epsilon_0^r\sqrt{\ln(1/\delta^r)}}{\sqrt{N}}\right). \tag{22}$$

*Further, if $\epsilon_0^r \leqslant 1/n$, then setting $\delta^r := Nn\delta_0^r$ implies that*

$$\epsilon^r = \mathcal{O}\left(\frac{\epsilon_0^r\sqrt{\ln(1/nN\delta_0^r)}}{\sqrt{N}}\right) \tag{23}$$

*and $\widetilde{\delta^r} \leqslant 7Nn\delta_0^r$, which is in $(0,1)$ if we assume $\delta_0^r \in (0, \frac{1}{7Nn}]$.*

We sometimes refer to the algorithm $\mathcal{A}_s^r$ as the "shuffled algorithm derived from the randomizers $\{\mathcal{R}_r^{(i)}\}$." From Theorem D.5, we obtain:

**Corollary D.1** (R-round privacy amplification for $C$-compositional algorithms)**.** *Let $\mathcal{A} : \mathcal{X}^{n \times N} \to \mathcal{Z}^{R \times N}$ be an $R$-round $(\epsilon_0, \delta_0)$-ISRL-DP and $C$-compositional algorithm such that $\epsilon_0 \in (0, \sqrt{N}]$ and $\delta_0 \in (0,1)$, where $\mathcal{X}$ is an arbitrary set. Assume that in each round, the local randomizers $\mathcal{R}_r^{(i)}(\mathbf{Z}_{(1:r-1)}, \cdot) : \mathcal{X}^n \to \mathcal{Z}$ are $(\epsilon_0^r, \delta_0^r)$-DP for $i \in [N], r \in [R]$, where $N \geqslant 16 \ln(2/\delta_0^r n)$, $\epsilon_0^r \leqslant \frac{1}{n}$, and $\delta_0^r \leqslant \frac{1}{14nNR}$. Then, the shuffled algorithm $\mathcal{A}_s : \mathcal{X}^{n \times N} \to \mathcal{Z}^{R \times N}$ derived from $\{\mathcal{R}_r^{(i)}(\mathbf{Z}_{(1:r-1)}, .)\}_{i \in [N], r \in [R]}$ (i.e. $\mathcal{A}_s$ is the composition of the $R$ shuffled algorithms $\mathcal{A}_s^r$ defined in Theorem D.5) is $(\epsilon, \delta)$-CDP, where $\delta \leqslant 14Nn \sum_{r=1}^R \delta_0^r$ and $\epsilon = \mathcal{O}\left(\frac{\epsilon_0 \ln\left(1/nN\delta_0^{\min}\right)C^2}{\sqrt{N}}\right)$, where $\delta_0^{\min} := \min_{r \in [R]} \delta_0^r$. In particular, if $\mathcal{A} \in \mathbb{A}$ is compositional, then $\epsilon = \mathcal{O}\left(\frac{\epsilon_0 \ln\left(1/nN\delta_0^{\min}\right)}{\sqrt{N}}\right)$.*

*Proof.* Let $\delta' := \sum_r Nn\delta_0^r$ and $\delta^r := Nn\delta_0^r$. Then the (central) privacy loss of the full $R$-round shuffled algorithm is bounded as

$$\epsilon \leqslant 2\sum_r (\epsilon^r)^2 + \sqrt{2\sum_r (\epsilon^r)^2 \ln(1/\delta')}$$

$$= \mathcal{O}\left(\sum_r \left(\frac{(\epsilon_0^r)^2 \ln(1/\delta^r)}{N}\right) + \sqrt{\sum_r \frac{(\epsilon_0^r)^2 \ln(1/\delta^r) \ln(1/\delta')}{N}}\right)$$

$$= \mathcal{O}\left(\frac{C^2 \epsilon_0 \ln(1/nN\delta_0^{\min})}{\sqrt{N}}\right),$$

where the three (in)equalities follow in order from the Advanced Composition Theorem (Dwork & Roth, 2014), (23) in Theorem D.5, and $C$-compositionality of $\mathcal{A}$ combined with the assumption $\epsilon_0 \lesssim \sqrt{N}$. Also, $\delta = \delta' + \sum_r \widetilde{\delta}^r$ by the Advanced Composition Theorem, where $\widetilde{\delta}_r \leqslant 7Nn\delta_0^r$ by Theorem D.5. Hence $\delta \leqslant 14Nn\sum_r \delta_0^r$. In particular, if $\mathcal{A}$ is compositional, then $C = \mathcal{O}(1)$, so $\epsilon = \mathcal{O}\left(\frac{\epsilon_0 \ln\left(1/nN\delta_0^{\min}\right)}{\sqrt{N}}\right)$. $\qquad\square$

**Remark D.1.** *The upper bounds assumed on $\delta_0^r$ and $\delta^r$ in Theorem 2.2 ensure that $\delta \in (0, 1)$ and that the lower bounds of (Bassily et al., 2014) apply (see Theorem D.6). These assumptions are not very restrictive in practice, since $\delta_0^r, \delta_0 \ll 1/n$ is needed for meaningful privacy guarantees (see e.g. Chapter 2 in (Dwork & Roth, 2014)) and $R$ must be polynomial for the algorithm to run. To quote (Dwork & Roth, 2014), "typically we are interested in values of $\delta$ that are less than the inverse of any polynomial in the size of the database" (page 18). For larger $\delta$ (e.g. $\delta = \Omega(1/n)$), there are examples of algorithms that satisfy the definition of DP but clearly violate any reasonable notion of privacy. For instance, an algorithm that outputs $\delta_0 n$ random samples from each silo's data set is $(0, \delta_0)$-ISRL-DP, but completely violates the privacy of at least one person in each silo if $\delta_0 \geqslant 1/n$. Also, since $N \gg 1$ is the regime of interest (otherwise if $N = \widetilde{O}(1)$, the CDP lower bounds of (Bassily et al., 2019) already match our upper bounds up to logarithms), the requirement that $N$ be larger than $16\ln(2/\delta_0^{\min}n)$ is unimportant.*[13]

The sequentially interactive part of Theorem D.4 will be clear directly from the proof of Theorem D.5. We now turn to the proof of Theorem D.5, which uses the techniques from (Feldman et al., 2020b). First, we'll need some more notation. The privacy relation in (3) between random variables $P$ and $Q$ can be characterized by the *hockey-stick divergence*: $D_{e^\epsilon}(P\|Q) := \int \max\{0, p(x) - e^\epsilon q(x)\}dx$, where $p$ and $q$ denote the probability density or mass functions of $P$ and $Q$ respectively. Then $P \underset{(\epsilon, \delta)}{\simeq} Q$ iff $\max\{D_{e^\epsilon}(P\|Q), D_{e^\epsilon}(Q\|P)\} \leqslant \delta$. Second, recall the *total variation distance* between $P$ and $Q$ is given by $TV(P, Q) = \frac{1}{2}\int_\mathbb{R} |p(x) - q(x)|dx$. Third, we recall the notion of group privacy:

**Definition 8** (Group DP). *A randomized algorithm $\mathcal{A} : \mathcal{X}^\mathcal{N} \to \mathcal{Z}$ is $(\epsilon, \delta)$ group DP for groups of size $\mathcal{N}$ if $\mathcal{A}(\mathbf{X}) \underset{(\epsilon, \delta)}{\simeq} \mathcal{A}(\mathbf{X}')$ for all $\mathbf{X}, \mathbf{X}' \in \mathcal{X}^\mathcal{N}$.*

We'll also need the following stronger version of a decomposition from (Kairouz et al., 2015) and Lemma 3.2 of (Murtagh & Vadhan, 2016).

**Lemma D.8** ((Kairouz et al., 2015)). *Let $\mathcal{R}_0, \mathcal{R}_1 : \mathcal{X}^n \to \mathcal{Z}$ be local randomizers such that $\mathcal{R}_0(X_0)$ and $\mathcal{R}_1(X_1)$ are $(\epsilon, 0)$ indistinguishable. Then, there exists a randomized algorithm $U : \{X_0, X_1\} \to \mathcal{Z}$ such that $\mathcal{R}_0(X_0) = \frac{e^\epsilon}{e^\epsilon+1}U(X_0) + \frac{1}{e^\epsilon+1}U(X_1)$ and $\mathcal{R}_1(X_1) = \frac{1}{e^\epsilon+1}U(X_0) + \frac{e^\epsilon}{e^\epsilon+1}U(X_1)$.*

Lemma D.8 follows from the proof of Lemma 3.2 in (Murtagh & Vadhan, 2016), noting that the weaker hypothesis assumed in Lemma D.8 sufficient for all steps to go through.

**Definition 9** (Deletion Group DP). *Algorithm $\mathcal{R} : \mathcal{X}^n \to \mathcal{Z}$ is $(\epsilon, \delta)$ deletion group DP for groups of size $n$ if there exists a reference distribution $\rho$ such that $\mathcal{R}(X) \underset{(\epsilon, \delta)}{\simeq} \rho$ for all $X \in \mathcal{X}^n$.*

It's easy to show that if $\mathcal{R}$ is $(\epsilon, \delta)$-*deletion* group DP for groups of size $n$, then $\mathcal{R}$ is $(2\epsilon, (1 + e^\epsilon)\delta)$ group DP for groups of size $n$. In addition, we have the following result:

---

[13]Technically, this assumption on $N$ is needed to ensure that the condition on $\epsilon_0^r$ in Theorem D.5 is satisfied. A similar restriction appears in Theorem 3.8 of (Feldman et al., 2020b).

**Lemma D.9.** *Let $X_0 \in \mathcal{X}^n$. If $\mathcal{R} : \mathcal{X}^n \to \mathcal{Z}$ is an $(\epsilon, \delta)$-DP local randomizer, then $\mathcal{R}$ is $(n\epsilon, ne^{(n-1)\epsilon}\delta)$ deletion group DP for groups of size $n$ with reference distribution $\mathcal{R}(X_0)$ (i.e. $\mathcal{R}(X) \underset{(\widetilde{\epsilon}, \widetilde{\delta})}{\simeq} \mathcal{R}(X_0)$ for all $X \in \mathcal{X}^n$, where $\widetilde{\epsilon} = n\epsilon$ and $\widetilde{\delta} = ne^{(n-1)\epsilon}\delta$).*

*Proof.* By group privacy (see e.g. Theorem 10 in (Kamath, 2020)), and the assumption that $\mathcal{R}$ is $(\epsilon, \delta)$-DP, it follows that $\mathcal{R}(X)$ and $\mathcal{R}(X')$ are $(n\epsilon, ne^{(n-1)\epsilon}\delta)$ indistinguishable for all $X, X' \in \mathcal{X}^n$. In particular, taking $X' := X_0$ completes the proof. $\quad\square$

**Lemma D.10.** *Let $\mathcal{R}^{(i)} : \mathcal{X}^n \to \mathcal{Z}$ be randomized algorithms ($i \in [N]$) and let $\mathcal{A}_s : \mathcal{X}^{n \times N} \to \mathcal{Z}^N$ be the shuffled algorithm $\mathcal{A}_s(\mathbf{X}) := (\mathcal{R}^{(1)}(X_{\pi(1)}), \cdots \mathcal{R}^{(N)}(X_{\pi(N)}))$ derived from $\{\mathcal{R}^{(i)}\}_{i \in [N]}$ for $\mathbf{X} = (X_1, \cdots, X_N)$, where $\pi$ is a uniformly random permutation of $[N]$. Let $\mathbf{X}_0 = (X_1^0, X_2, \cdots, X_N)$ and $\mathbf{X}_1 = (X_1^1, X_2, \cdots, X_N)$, $\delta \in (0, 1)$ and $p \in [\frac{16 \ln(2/\delta)}{N}, 1]$. Suppose that for all $i \in [N], X \in \mathcal{X}^n \backslash \{X_1^1, X_1^0\}$, there exists a distribution $LO^{(i)}(X)$ such that*

$$\mathcal{R}^{(i)}(X) = \frac{p}{2}\mathcal{R}^{(i)}(X_1^0) + \frac{p}{2}\mathcal{R}^{(i)}(X_1^1) + (1 - p)LO^{(i)}(X).$$

*Then $\mathcal{A}_s(\mathbf{X}_0) \underset{(\epsilon, \delta)}{\simeq} \mathcal{A}_s(\mathbf{X}_1)$, where*

$$\epsilon \leqslant \ln\left(1 + \frac{8\sqrt{\ln(4/\delta)}}{\sqrt{pN}} + \frac{8}{pN}\right).$$

*Proof.* The proof mirrors the proof of Lemma 3.3 in (Feldman et al., 2020b) closely, replacing their notation with ours. Observe that the DP assumption in Lemma 3.3 of (Feldman et al., 2020b) is not actually needed in the proof. $\quad\square$

**Lemma D.11.** *Let $\mathcal{R} : \mathcal{X}^n \to \mathcal{Z}$ be $(\epsilon, \delta)$ deletion group DP for groups of size $n$ with reference distribution $\rho$. Then there exists a randomizer $\mathcal{R}' : \mathcal{X}^n \to \mathcal{Z}$ such that:*
*(i) $\mathcal{R}'$ is $(\epsilon, 0)$ deletion group DP for groups of size $n$ with reference distribution $\rho$; and*
*(ii) $TV(\mathcal{R}(X), \mathcal{R}'(X)) \leqslant \delta$.*
*In particular, $\mathcal{R}'$ is $(2\epsilon, 0)$ group DP for groups of size $n$ (by (i)).*

*Proof.* The proof is nearly identical to the proof of Lemma 3.7 in (Feldman et al., 2020b). $\quad\square$

We also need the following stronger version of Lemma 3.7 from (Feldman et al., 2020b):

**Lemma D.12.** *If $\mathcal{R}(X_1^0) \underset{(\epsilon_0, \delta_0)}{\simeq} \mathcal{R}(X_1^1)$, then there exists a randomizer $\mathcal{R}' : \mathcal{X}^n \to \mathcal{Z}$ such that $\mathcal{R}'(X_1^1) \underset{(\epsilon_0, 0)}{\simeq} \mathcal{R}(X_1^0)$ and $TV(\mathcal{R}'(X_1^1), \mathcal{R}(X_1^1)) \leqslant \delta_0$.*

*Proof.* The proof follows the same techniques as Lemma 3.7 in (Feldman et al., 2020b), noting that the weaker hypothesis in Lemma D.12 is sufficient for all the steps to go through and that the assumption of $n = 1$ in (Feldman et al., 2020b) is not needed in the proof. $\quad\square$

**Lemma D.13** ((Dwork & Roth, 2014), Lemma 3.17). *Given random variables $P, Q, P'$ and $Q'$, if $D_{e^\epsilon}(P', Q') \leqslant \delta$, $TV(P, P') \leqslant \delta'$, and $TV(Q, Q') \leqslant \delta'$, then $D_{e^\epsilon}(P, Q) \leqslant \delta + (e^\epsilon + 1)\delta'$.*

**Lemma D.14** ((Feldman et al., 2020b), Lemma 2.3). *Let $P$ and $Q$ be distributions satisfying $P = (1 - q)P_0 + qP_1$ and $Q = (1 - q)P_0 + qQ_1$ for some $q \in [0, 1]$. Then for any $\epsilon > 0$, if $\epsilon' = \log(1 + q(e^\epsilon - 1))$, then*

$$D_{e^{\epsilon'}}(P||Q) \leqslant q \max\{D_{e^\epsilon}(P_1||P_0), D_{e^\epsilon}(P_1||Q_1)\}.$$

We are now ready to prove Theorem D.5:

*Proof of Theorem D.5.* Let $\mathbf{X}_0, \mathbf{X}_1 \in \mathcal{X}^{n \times N}$ be adjacent (in the CDP sense) distributed data sets (i.e. $|\mathbf{X}_0 \Delta \mathbf{X}_1| \leqslant 1$). Assume WLOG that $\mathbf{X}_0 = (X_1^0, X_2, \cdots, X_N)$ and $\mathbf{X}_1 = (X_1^1, X_2, \cdots, X_N)$, where $X_1^0 = (x_{1,0}, x_{1,2}, \cdots, x_{1,n}) \neq (x_{1,1}, x_{1,2}, \cdots, x_{1,n})$. We can also assume WLOG that $X_j \notin \{X_1^0, X_1^1\}$ for all $j \in \{2, \cdots, N\}$ by re-defining $\mathcal{X}$ and $\mathcal{R}_r^{(i)}$ if necessary.

Fix $i \in [N], r \in [R], \mathbf{Z} = \mathbf{Z}_{1:r-1} = Z_{(1:r-1)}^{(1:N)} \in \mathcal{Z}^{(r-1) \times N}$, denote $\mathcal{R}(X) := \mathcal{R}_r^{(i)}(\mathbf{Z}, X)$ for $X \in \mathcal{X}^n$, and $\mathcal{A}_s(\mathbf{X}) := \mathcal{A}_s^r(\mathbf{Z}_{1:r-1}, \mathbf{X})$. Draw $\pi$ uniformly from the set of permutations of $[N]$. Now, since $\mathcal{R}$ is $(\epsilon_0, \delta_0)$-DP, $\mathcal{R}(X_1^1) \underset{(\epsilon_0^r, \delta_0^r)}{\simeq} \mathcal{R}(X_1^0)$, so by Lemma D.12, there exists a local randomizer $\mathcal{R}'$ such that $\mathcal{R}'(X_1^1) \underset{(\epsilon_0^r, 0)}{\simeq} \mathcal{R}(X_1^0)$ and $TV(\mathcal{R}'(X_1^1), \mathcal{R}(X_1^1)) \leqslant \delta_0^r$.

Hence, by Lemma D.8, there exist distributions $U(X_1^0)$ and $U(X_1^1)$ such that

$$\mathcal{R}(X_1^0) = \frac{e^{\epsilon_0^r}}{e^{\epsilon_0^r} + 1} U(X_1^0) + \frac{1}{e^{\epsilon_0^r} + 1} U(X_1^1) \tag{24}$$

and

$$\mathcal{R}'(X_1^1) = \frac{1}{e^{\epsilon_0^r} + 1} U(X_1^0) + \frac{e^{\epsilon_0^r}}{e^{\epsilon_0^r} + 1} U(X_1^1). \tag{25}$$

Denote $\widetilde{\epsilon_0} := n\epsilon_0^r$ and $\widetilde{\delta_0} := ne^{(n-1)\epsilon_0^r}\delta_0^r$. By convexity of hockey-stick divergence and the hypothesis that $\mathcal{R}$ is $(\epsilon_0^r, \delta_0^r)$-DP (hence $\mathcal{R}(X) \underset{(\widetilde{\epsilon_0}, \widetilde{\delta_0})}{\simeq} \mathcal{R}(X_1^0), \mathcal{R}(X_1^1)$ for all $X$ by Lemma D.9), we have $\mathcal{R}(X) \underset{(\widetilde{\epsilon_0}, \widetilde{\delta_0})}{\simeq} \frac{1}{2}(\mathcal{R}(X_1^0) + \mathcal{R}(X_1^1)) := \rho$ for all $X \in \mathcal{X}^n$. That is, $\mathcal{R}$ is $(\widetilde{\epsilon_0}, \widetilde{\delta_0})$ deletion group DP for groups of size $n$ with reference distribution $\rho$. Thus, Lemma D.11 implies that there exists a local randomizer $\mathcal{R}''$ such that $\mathcal{R}''(X)$ and $\rho$ are $(\widetilde{\epsilon_0}, 0)$ indistinguishable and $TV(\mathcal{R}''(X), \mathcal{R}(X)) \leqslant \widetilde{\delta_0}$ for all $X$. Then by the definition of $(\widetilde{\epsilon_0}, 0)$ indistinguishability, for all $X$ there exists a "left-over" distribution $LO(X)$ such that $\mathcal{R}''(X) = \frac{1}{e^{\widetilde{\epsilon_0}}} \rho + (1 - 1/e^{\widetilde{\epsilon_0}}) LO(X) = \frac{1}{2e^{\widetilde{\epsilon_0}}}(\mathcal{R}(X_1^0) + \mathcal{R}(X_1^1)) + (1 - 1/e^{\widetilde{\epsilon_0}}) LO(X)$.

Now, define a randomizer $\mathcal{L}$ by $\mathcal{L}(X_1^0) := \mathcal{R}(X_1^0)$, $\mathcal{L}(X_1^1) := \mathcal{R}'(X_1^1)$, and

$$\begin{aligned}\mathcal{L}(X) &:= \frac{1}{2e^{\widetilde{\epsilon_0}}} \mathcal{R}(X_1^0) + \frac{1}{2e^{\widetilde{\epsilon_0}}} \mathcal{R}'(X_1^1) + (1 - 1/e^{\widetilde{\epsilon_0}}) LO(X) \\ &= \frac{1}{2e^{\widetilde{\epsilon_0}}} U(X_1^0) + \frac{1}{2e^{\widetilde{\epsilon_0}}} U(X_1^1) + (1 - 1/e^{\widetilde{\epsilon_0}}) LO(X)\end{aligned} \tag{26}$$

for all $X \in \mathcal{X}^n \backslash \{X_1^0, X_1^1\}$. (The equality follows from (24) and (25).) Note that $TV(\mathcal{R}(X_1^0), \mathcal{L}(X_1^0)) = 0$, $TV(\mathcal{R}(X_1^1), \mathcal{L}(X_1^1)) \leqslant \delta_0^r$, and for all $X \in \mathcal{X}^n \backslash \{X_1^0, X_1^1\}$, $TV(\mathcal{R}(X), \mathcal{L}(X)) \leqslant TV(\mathcal{R}(X), \mathcal{R}''(X)) + TV(\mathcal{R}''(X), \mathcal{L}(X)) \leqslant \widetilde{\delta_0} + \frac{1}{2e^{\widetilde{\epsilon_0}}} TV(\mathcal{R}'(X_1^1), \mathcal{R}(X_1^1)) = (ne^{(n-1)\epsilon_0^r} + \frac{1}{2e^{n\epsilon_0^r}})\delta_0^r \leqslant (2ne^{(n-1)\epsilon_0^r})\delta_0^r = 2\widetilde{\delta_0}$.

Keeping $r$ fixed (omitting $r$ scripts everywhere), for any $i \in [N]$ and $\mathbf{Z} := \mathbf{Z}_{1:r-1} \in \mathcal{Z}^{(r-1) \times N}$, let $\mathcal{L}^{(i)}(\mathbf{Z}, \cdot)$, $U^{(i)}(\mathbf{Z}, \cdot)$, and $LO^{(i)}(\mathbf{Z}, \cdot)$ denote the randomizers resulting from the process described above. Let $\mathcal{A}_\mathcal{L} : \mathcal{X}^{n \times N} \to \mathcal{Z}^N$ be defined exactly the same way as $\mathcal{A}_s^r := \mathcal{A}_s$ (same $\pi$) but with the randomizers $\mathcal{R}^{(i)}$ replaced by $\mathcal{L}^{(i)}$. Since $\mathcal{A}_s$ applies each randomizer $\mathcal{R}^{(i)}$ exactly once and $\mathcal{R}^{(1)}(\mathbf{Z}, X_{\pi(1)}), \cdots \mathcal{R}^{(N)}(\mathbf{Z}, X_{\pi(N)})$ are independent (conditional on $\mathbf{Z} = \mathbf{Z}_{1:r-1}$) [14], we have $TV(\mathcal{A}_s(\mathbf{X}_0), \mathcal{A}_\mathcal{L}(\mathbf{X}_0)) \leqslant N(2ne^{(n-1)\epsilon_0^r})\delta_0^r$ and $TV(\mathcal{A}_s(\mathbf{X}_1), \mathcal{A}_\mathcal{L}(\mathbf{X}_1)) \leqslant N(2ne^{(n-1)\epsilon_0^r})\delta_0^r$ (see (Den Hollander, 2012)). Now we claim that $\mathcal{A}_\mathcal{L}(\mathbf{X}_0)$ and $\mathcal{A}_\mathcal{L}(\mathbf{X}_1)$ are $(\epsilon^r, \delta^r)$ indistinguishable for any $\delta^r \geqslant 2e^{-Ne^{-n\epsilon_0^r}/16}$. Observe that this claim implies that $\mathcal{A}_s(\mathbf{X}_0)$ and $\mathcal{A}_s(\mathbf{X}_1)$ are $(\epsilon^r, \widetilde{\delta^r})$ indistinguishable by Lemma D.13 (with $P' := \mathcal{A}_\mathcal{L}(\mathbf{X}_0), Q' := \mathcal{A}_\mathcal{L}(\mathbf{X}_1), P := \mathcal{A}_s(\mathbf{X}_0), Q := \mathcal{A}_s(\mathbf{X}_1)$.) Therefore, it remains to prove the claim, i.e. to show that $D_{e^{\epsilon^r}}(\mathcal{A}_\mathcal{L}(\mathbf{X}_0), \mathcal{A}_\mathcal{L}(\mathbf{X}_1)) \leqslant \delta^r$ for any $\delta^r \geqslant 2e^{-Ne^{-n\epsilon_0^r}/16}$.

---

[14]This follows from the assumption given in the lead up to Definition 5 that $\mathcal{R}^{(i)}(\mathbf{Z}_{1:r-1}, X)$ is conditionally independent of $X'$ given $\mathbf{Z}_{1:r-1}$ for all $\mathbf{Z}_{1:r-1}$ and $X \neq X'$.

Now, define $\mathcal{L}_U^{(i)}(\mathbf{Z}, X) := \begin{cases} U^{(i)}(\mathbf{Z}, X_1^0) & \text{if } X = X_1^0 \\ U^{(i)}(\mathbf{Z}, X_1^1) & \text{if } X = X_1^1 \\ \mathcal{L}^{(i)}(\mathbf{Z}, X) & \text{otherwise.} \end{cases}$ . For any inputs $\mathbf{Z}, \mathbf{X}$, let $\mathcal{A}_U(\mathbf{Z}, \mathbf{X})$ be

defined exactly the same as $\mathcal{A}_s(\mathbf{Z}, \mathbf{X})$ (same $\pi$) but with the randomizers $\mathcal{R}^{(i)}$ replaced by $\mathcal{L}_U^{(i)}$. Then by (24) and (25),

$$\mathcal{A}_{\mathcal{L}}(\mathbf{X}_0) = \frac{e^{\epsilon_0^r}}{e^{\epsilon_0^r} + 1} \mathcal{A}_U(\mathbf{X}_0) + \frac{1}{e^{\epsilon_0^r} + 1} \mathcal{A}_U(\mathbf{X}_1) \text{ and } \mathcal{A}_{\mathcal{L}}(\mathbf{X}_1) = \frac{1}{e^{\epsilon_0^r} + 1} \mathcal{A}_U(\mathbf{X}_0) + \frac{e^{\epsilon_0^r}}{e^{\epsilon_0^r} + 1} \mathcal{A}_U(\mathbf{X}_1).$$
(27)

Then by (26), for any $X \in \mathcal{X}^n \setminus \{X_1^0, X_1^1\}$ and any $\mathbf{Z} = \mathbf{Z}_{1:r-1} \in \mathcal{Z}^{(r-1) \times N}$, we have $\mathcal{L}_U^{(i)}(\mathbf{Z}, X) = \frac{1}{2e^{\widetilde{\epsilon_0}}} \mathcal{L}_U^{(i)}(\mathbf{Z}, X_1^0) + \frac{1}{2e^{\widetilde{\epsilon_0}}} \mathcal{L}_U^{(i)}(\mathbf{Z}, X_1^1) + (1 - e^{-\widetilde{\epsilon_0}}) LO^{(i)}(\mathbf{Z}, X)$. Hence, Lemma D.10 (with $p := e^{-\widetilde{\epsilon_0}} = e^{-n\epsilon_0^r}$) implies that $\mathcal{A}_U(\mathbf{X}_0)$ and $\mathcal{A}_U(\mathbf{X}_1)$) are

$$\left( \log \left( 1 + \frac{8\sqrt{e^{\widetilde{\epsilon_0}} \ln(4/\delta^r)}}{\sqrt{N}} + \frac{8e^{\widetilde{\epsilon_0}}}{N} \right), \delta^r \right)$$

indistinguishable for any $\delta^r \geqslant 2e^{-Ne^{-n\epsilon_0^r}/16}$. Applying Lemma D.14 with $P := \mathcal{A}_{\mathcal{L}}(\mathbf{X}_0)$, $Q = \mathcal{A}_{\mathcal{L}}(\mathbf{X}_1)$, $q = \frac{e^{\epsilon_0^r} - 1}{e^{\epsilon_0^r} + 1}$, $P_1 = \mathcal{A}_U(\mathbf{X}_0)$, $Q_1 = \mathcal{A}_U(\mathbf{X}_1)$, and $P_0 = \frac{1}{2}(P_1 + Q_1)$ and convexity of the hockey-stick divergence yields that $\mathcal{A}_{\mathcal{L}}(\mathbf{X}_0)$ and $\mathcal{A}_{\mathcal{L}}(\mathbf{X}_1)$ are $(\epsilon^r, \delta^r)$ indistinguishable, as desired. This proves the claim and hence (by Lemma D.13, as described earlier) the theorem. □

**Remark D.2.** *Notice that if $\mathcal{A}$ is sequentially interactive, then the proof of Theorem D.5 above almost immediately implies the sequentially interactive part of Theorem D.4. Essentially, just change notation: replace $\mathbf{Z}_{1:r-1}$ by $Z^{(1:i-1)}$, the collection of (single) reports sent by the first $i - 1$ silos; note that $\epsilon_0^r = \epsilon_0$, $\delta_0^r = \delta_0$; and view the $N$ reports as being sent in order instead of simultaneously. Alternatively, plug our techniques for $n > 1$ into the proof of Theorem 3.8 in (Feldman et al., 2020b), which is for sequentially interactive algorithms.*

**Step 2:** Combine Theorem D.4 with the following CDP SCO lower bounds which follow from (Bassily et al., 2014; 2019) and the non-private SCO lower bounds (Nemirovskii & Yudin, 1983; Agarwal et al., 2012) :

**Theorem D.6.** *(Bassily et al., 2019; 2014) Let $\mu, D, \epsilon > 0$, $L \geqslant \mu D$, and $\delta = o(1/nN)$. Consider $\mathcal{X} := \{\frac{-D}{\sqrt{d}}, \frac{D}{\sqrt{d}}\}^d \subset \mathbb{R}^d$ and $\mathcal{W} := B_2(0, D) \subset \mathbb{R}^d$. Let $\mathcal{A} : \mathcal{X}^{nN} \to \mathcal{W}$ be any $(\epsilon, \delta)$-CDP algorithm. Then:*
*1. There exists a ($\mu = 0$) convex, linear ($\beta$-smooth for any $\beta$), L-Lipschitz loss $f : \mathcal{W} \times \mathcal{X} \to \mathbb{R}$ and a distribution $\mathcal{D}$ on $\mathcal{X}$ such that if $\mathbf{X} \sim \mathcal{D}^{nN}$, then the expected excess loss of $\mathcal{A}$ is lower bounded as*

$$\mathbb{E}F(\mathcal{A}(\mathbf{X})) - F^* = \widetilde{\Omega} \left( \frac{\phi D}{\sqrt{Nn}} + LD \min \left\{ 1, \frac{\sqrt{d}}{\epsilon n N} \right\} \right).$$

*2. For $L \approx \mu D$, there exists a $\mu$-strongly convex and smooth, L-Lipschitz loss $f : \mathcal{W} \times \mathcal{X} \to \mathbb{R}$ and a distribution $\mathcal{D}$ on $\mathcal{X}$ such that if $\mathbf{X} \sim \mathcal{D}^{nN}$, then the expected excess loss of $\mathcal{A}$ is lower bounded as*

$$\mathbb{E}F(\mathcal{A}(\mathbf{X})) - F^* = \widetilde{\Omega} \left( \frac{\phi^2}{\mu Nn} + \frac{L^2}{\mu} \min \left\{ 1, \frac{d}{\epsilon^2 n^2 N^2} \right\} \right).$$

*For general $L, \mu, D$, the above strongly convex lower bound holds with the factor $\frac{L^2}{\mu}$ replaced by $LD$.*

Namely, if $\mathcal{A}$ is $(\epsilon_0, \delta_0)$-ISRL-DP, then (under the hypotheses of Theorem 2.2) $\mathcal{A}_s$ is $(\epsilon, \delta)$-CDP for $\epsilon = \widetilde{O}(\epsilon_0/\sqrt{N})$, so Theorem D.6 implies that the excess loss of $\mathcal{A}_s$ is lower bounded as in Theorem D.6 with $\epsilon$ replaced by $\epsilon_0/\sqrt{N}$.

**Step 3:** We simply observe that when the expectation is taken over the randomness in sampling $\mathbf{X} \sim \mathcal{D}^{n \times N}$, the expected excess population loss of $\mathcal{A}_s$ is identical to that of $\mathcal{A}$ since $X_i$ and $X_{\pi(i)}$ have the same distribution for all $i, \pi$ by the i.i.d. assumption. This completes the proof of Theorem 2.2.

### D.3.5 Tight Excess Risk Bounds for Cross-Device FL Without a Trusted Server

We explain how our ISRL-DP excess risk bounds also imply tight excess risk bounds for algorithms that satisfy *both ISRL-DP and user-level DP simultaneously*, which may be desirable for cross-device FL without a trusted server. Assume $M = N$ for simplicity and consider i.i.d. FL for concreteness.[15] Given $(\epsilon, \delta)$ with $\epsilon \leqslant 1$, let $\epsilon_0 = \epsilon/n$, $\delta_0 = \delta/4n \leqslant \delta/(ne^{(n-1)\epsilon_0}) = \delta/(ne^{(n-1)\epsilon/n})$ and run Noisy (ISRL-DP) MB-SGD with noise calibrated to $(\epsilon_0, \delta_0)$. Then the algorithm also satisfies $(\epsilon, \delta)$-user level DP by Appendix C. Thus, by Theorem 2.1, we obtain hybrid ISRL-DP/user-level DP excess risk upper bounds. For example, in the convex case, we get

$$
\begin{aligned}
\mathbb{E}F(\widehat{w}_R) - F^* &= \tilde{\mathcal{O}}\left( LD\left( \frac{\sqrt{d\ln(1/\delta_0)}}{\epsilon_0 n\sqrt{N}} + \frac{1}{\sqrt{Nn}} \right) \right) \\
&= \tilde{\mathcal{O}}\left( LD\left( \frac{\sqrt{d\ln(n/\delta)}}{\epsilon\sqrt{N}} + \frac{1}{\sqrt{Nn}} \right) \right) \\
&= \tilde{\mathcal{O}}\left( LD\frac{\sqrt{d\ln(n/\delta)}}{\epsilon\sqrt{N}} \right).
\end{aligned}
\tag{28}
$$

Regarding lower bounds: note that the semantics of the hybrid ISRL-DP/user-level DP notion are essentially identical to LDP, except that individual "records/items" are now thought of as datasets of size $n$. Thus, letting $n = 1$ in our ISRL-DP lower bounds (we think of each silo as having just one "record" even though that record is really a dataset) yields lower bounds matching (up to logarithms) the upper bounds attained above. For example, putting $n = 1$ in the convex lower bound in Theorem 2.2 yields a bound that matches (28), establishing the optimal FL rates (up to logarithmic factors) for this algorithm class. Note that the minimax risk bounds for ISRL-DP/user-level DP hybrid algorithms resemble the bounds for LDP algorithms (Duchi et al., 2013), scaling with $N$, but not with $n$.

## E  Proofs and Supplemental Material for Section 3

### E.1  Multi-Stage Implementation of Accelerated Noisy MB-SGD

Here we describe the multi-stage implementation of Accelerated Noisy MB-SGD that we will use to further expedite convergence for strongly convex loss, which builds on (Ghadimi & Lan, 2013). As before, for SCO, silos sample locally without replacement in each round and set $R = \lfloor n/K \rfloor$. Whereas for ERM, silos sample locally *with replacement*.

**Multi-stage implementation of Algorithm 1:** Inputs: $U \in [R]$ such that $\sum_{k=1}^{U} R_k \leqslant R$ for $R_k$ defined below; $w_0 \in \mathcal{W}, \Delta \geqslant F(w_0) - F^*, V > 0,$ and $q_0 = 0$.
For $k \in [U]$, do the following:

1. Run $R_k$ rounds of Algorithm 1 using $w_0 = q_{k-1}, \{\alpha_r\}_{r \geqslant 1}$ and $\{\eta_r\}_{r \geqslant 1}$, where

$$
R_k = \left\lceil \max\left\{ 4\sqrt{\frac{2\beta}{\mu}}, \frac{128V^2}{3\mu\Delta 2^{-(k+1)}} \right\} \right\rceil,
$$

$$
\alpha_r = \frac{2}{r+1}, \quad \eta_r = \frac{4\upsilon_k}{r(r+1)},
$$

$$
\upsilon_k = \max\left\{ 2\beta, \left[ \frac{\mu V^2}{3\Delta 2^{-(k-1)}R_k(R_k+1)(R_k+2)} \right]^{1/2} \right\}
$$

2. Set $q_k = w_{R_k}^{ag}$, where $w_{R_k}^{ag}$ is the output of Step 1 above. Then update $k \leftarrow k + 1$ and return to Step 1.

---

[15] It's easy to see that the same arguments we use in this subsection for i.i.d. SCO can also be used to establish tight federated ERM bounds.

### E.2 COMPLETE VERSION AND PROOF OF THEOREM 3.1

We will state and prove the theorem for general $M \in [N]$ under Assumption **3**. We first require some additional notation: Define the heterogeneity parameter

$$v^2 := \sup_{w \in \mathcal{W}} \frac{1}{N} \sum_{i=1}^{N} \|\nabla F_i(w) - \nabla F(w)\|^2, \tag{29}$$

which has appeared in (Khaled et al., 2019; Koloskova et al., 2020; Karimireddy et al., 2020; Woodworth et al., 2020b). $v^2 = 0$ iff $F_i = F + a_i$ for constants $a_i \in \mathbb{R}$, $i \in [N]$ ("homogeneous up to transaltion").

**Theorem E.1** (Complete version of Theorem 3.1). *Let $f(\cdot, x)$ be $\beta$-smooth for all $x \in \mathcal{X}$. Assume $\epsilon_0 \leqslant 8\ln(1/\delta_0)$, $\delta_0 \in (0, 1)$ Then, with $\sigma^2 = \frac{32L^2 \ln(1.25/\delta_0)}{\epsilon_0^2 K^2}$, One-Pass Accelerated Noisy MB-SGD is $(\epsilon_0, \delta_0)$-ISRL-DP. Moreover, there are choices of stepsize, batch size, and $\lambda > 0$ such that :*
*1. Running One-Pass Accelerated Noisy MB-SGD on $\tilde{f}(w, x) := f(w, x) + \frac{\lambda}{2}\|w - w_0\|^2$ (where $w_0 \in \mathcal{W}$) yields*

$$\mathbb{E}F(w_R^{ag}) - F^* = \tilde{\mathcal{O}}\left(\frac{\phi D}{\sqrt{nM}} + \left(\frac{\beta^{1/4}LD^{3/2}\sqrt{d\ln(1/\delta_0)}}{\epsilon_0 n\sqrt{M}}\right)^{4/5} + \sqrt{\frac{N-M}{N-1}}\mathbb{1}_{\{N>1\}}\frac{vL^{1/5}D^{4/5}}{\beta^{1/5}}\left(\frac{\sqrt{d\ln(1/\delta_0)}}{\epsilon_0 nM^3}\right)^{1/5}\right). \tag{30}$$

*2. If $f(\cdot, x)$ is $\mu$-strongly convex $\forall x \in \mathcal{X}$ and $\kappa = \frac{\beta}{\mu}$, then running the Multi-Stage Implementation of One-Pass Accelerated Noisy MB-SGD directly on $f$ yields with batch size $K$ yields*

$$\mathbb{E}F(w_R^{ag}) - F^* = \tilde{\mathcal{O}}\left(\frac{\phi^2}{nM} + \frac{L^2}{\mu}\frac{\sqrt{\kappa}d\ln(1/\delta_0)}{\epsilon_0^2 n^2 M} + \frac{v^2}{\mu\sqrt{\kappa}M}\left(1 - \frac{M-1}{N-1}\right)\mathbb{1}_{\{N>1\}}\right). \tag{31}$$

**Remark E.1.** *1. For convex $f$, if $M = N$ or*

$$v \lesssim \sqrt{\frac{N-1}{N-M}}\left[(L^3D^2\beta^2)^{1/5}\frac{M^{1/5}(d\ln(1/\delta_0))^{3/10}}{n^{3/5}\epsilon_0^{3/5}} + \phi\left(\frac{\beta D}{L}\right)^{1/5}\left(\frac{\epsilon_0^2 M}{n^3 d\ln(1/\delta_0)}\right)^{1/10}\right],$$

*then (30) recovers the bound (6) in Theorem 3.1 ($M = N$ version), with $N$ replaced by $M$.*
*2. For $\mu$-strongly convex $f$, if $M = N$ or*

$$v^2 \lesssim \left(\frac{N-1}{N-M}\right)\sqrt{\kappa}\left(\frac{\phi^2}{n} + \frac{\sqrt{\kappa}L^2 d\ln(1/\delta_0)}{\epsilon_0^2 n^2}\right),$$

*then (31) recovers the bound (7) in Theorem 3.1, with $N$ replaced by $M$.*

To prove Theorem E.1, we will need some preliminaries:

**Lemma E.1.** *(Woodworth et al., 2020b, Lemma 4) Let $F : \mathcal{W} \to \mathbb{R}^d$ be convex and $\beta$-smooth, and suppose that the unbiased stochastic gradients $\tilde{g}(w_t)$ at each iteration have bounded variance $\mathbb{E}\|\tilde{g}(w) - \nabla F(w)\|^2 \leqslant V^2$. If $\hat{w}_R^{ag}$ is computed by $R$ steps of Accelerated MB-SGD on the regularized objective $\tilde{F}(w) = F(w) + \frac{V}{2\|w_0 - w^*\|\sqrt{R}}\|w - w_0\|^2$, then*

$$\mathbb{E}F(\hat{w}_R^{ag}) - F^* \lesssim \frac{\beta\|w_0 - w^*\|^2}{R^2} + \frac{V\|w_0 - w^*\|}{\sqrt{R}}.$$

We then have the following bound for the multi-stage protocol:

**Lemma E.2.** *(Ghadimi & Lan, 2013, Proposition 7) Let $f : \mathcal{W} \to \mathbb{R}^d$ be $\mu$-strongly convex and $\beta$-smooth, and suppose that the unbiased stochastic gradients $\tilde{g}(w_r)$ at each iteration $r$ have bounded variance $\mathbb{E}\|\tilde{g}(w_r) - \nabla F(w_t)\|^2 \leqslant V^2$. If $\hat{w}_R^{ag}$ is computed by $R$ steps of the Multi-Stage Accelerated MB-SGD, then*

$$\mathbb{E}F(\hat{w}_R^{ag}) - F^* \lesssim \Delta\exp\left(-\sqrt{\frac{\mu}{\beta}}R\right) + \frac{V^2}{\mu R},$$

*where $\Delta = F(w_0) - F^*$.*

Of course, (Woodworth et al., 2020b, Lemma 4) and (Ghadimi & Lan, 2013, Proposition 7) are stated for the *non-private* Accelerated MB-SGD (AC-SA). However, we observe that the bounds in (Woodworth et al., 2020b, Lemma 4) and (Ghadimi & Lan, 2013, Proposition 7) depend only on the stochastic gradient oracle via its bias and variance. Hence these results also apply to our (multi-stage implementation of) Accelerated *Noisy* MB-SGD. Next, we bound the variance of our noisy stochastic gradient estimators:

**Lemma E.3.** *Let* $X_i \sim \mathcal{D}_i^n$, $\widetilde{g}_r := \frac{1}{M_r} \sum_{i \in S_r} \frac{1}{K} \sum_{j \in [K]} (\nabla f(w_r, x_{i,j}^\tau) + u_i)$, *where* $(x_{i,j}^\tau)_{j \in [K]}$ *are sampled from* $X_i$ *and* $u_i \sim \mathcal{N}(0, \sigma^2 \mathbf{I}_d)$ *is independent of* $\nabla f(w_r, x_{i,j}^\tau)$ *for all* $i \in [N], j \in [K]$. *Then*

$$\mathbb{E}\|\widetilde{g}_r - \nabla F(w_r)\|^2 \leqslant \frac{\phi^2}{MK} + \left(1 - \frac{M-1}{N-1}\right) \frac{v^2}{M} \mathbb{1}_{\{N>1\}} + \frac{d\sigma^2}{M}.$$

The three terms on the right-hand side correspond (from left to right) to the variances of: local minibatch sampling within each silo, the draw of the silo set $S_r$ of size $M_r$ under Assumption **3**, and the Gaussian noise. We now turn to the proof of Lemma E.3.

*Proof of Lemma E.3.* First, fix the randomness due to the size of the silo set $M_r$. Now $\widetilde{g}_r = g_r + \bar{u}_r$, where $\bar{u}_r = \frac{1}{M_r} \sum_{i=1}^{M_r} u_i \sim N(0, \frac{\sigma^2}{M_r} \mathbf{I}_d)$ and $\bar{u}_r$ is independent of $g_r := \frac{1}{M_r} \sum_{i \in S_r} \frac{1}{K_i} \sum_{j \in [K]} \nabla f(w_r, x_{i,j}^r)$. Hence,

$$\mathbb{E}[\|\widetilde{g}_r - \nabla F(w_r)\|^2 | M_r] = \mathbb{E}[\|g_r - \nabla F(w_r)\|^2 | M_r] + \mathbb{E}[\|\bar{u}\|^2 | M_r]$$

$$= \mathbb{E}[\|g_r - \nabla F(w_r)\|^2 | M_r] + d \frac{\sigma^2}{M_r}.$$

Let us drop the $r$ subscripts for brevity (denoting $g = g_r$, $w = w_r$, $S = S_r$, and $M_r = M_1$ since they have the same distribution) and denote $h_i := \frac{1}{K_i} \sum_{j=1}^{K_i} \nabla f(w, x_{i,j})$. Now, we have (conditionally on $M_1$)

$$\mathbb{E}[\|g - \nabla F(w)\|^2 | M_1] = \mathbb{E}\left[\left\|\frac{1}{M_1} \sum_{i \in S} \frac{1}{K_i} \sum_{j=1}^{K_i} \nabla f(w, x_{i,j}) - \nabla F(w)\right\|^2 \Bigg| M_1\right]$$

$$= \mathbb{E}\left[\left\|\frac{1}{M_1} \sum_{i \in S} (\nabla h_i - \nabla F_i(w)) + \frac{1}{M_1} \sum_{i \in S} \nabla F_i(w) - \nabla F(w)\right\|^2 \Bigg| M_1\right]$$

$$= \frac{1}{M_1^2} \underbrace{\mathbb{E}\left[\|\sum_{i \in S} h_i(w) - \nabla F_i(w)\|^2 \Big| M_1\right]}_{\text{(a)}}$$

$$+ \frac{1}{M_1^2} \underbrace{\mathbb{E}\left[\|\sum_{i \in S} \nabla F_i(w) - \nabla F(w)\|^2 \Big| M_1\right]}_{\text{(b)}},$$

since, conditional on $S$, the cross-terms vanish by (conditional) independence of $h_i$ and the non-random $\sum_{i' \in S} \nabla F_{i'}(w) - \nabla F(w)$ for all $i \in S$. Now we bound (a):

$$\text{(a)} = \mathbb{E}_S\left[\mathbb{E}_{h_i} \|\sum_{i \in S} h_i(w) - \nabla F_i(w)\|^2 \Big| S, M_1\right]$$

$$= \mathbb{E}_S\left[\sum_{i \in S} \mathbb{E}_{h_i} \|h_i(w) - \nabla F_i(w)\|^2 \Big| S, M_1\right]$$

$$\leqslant \mathbb{E}_S\left[\sum_{i \in S} \frac{\phi^2}{K}\right]$$

$$\leqslant \mathbb{E}_S\left[\frac{M_1 \phi^2}{K}\right],$$

by conditional independence of $h_i - \nabla F_i$ and $h_{i'} - \nabla F_{i'}$ given $S$. Hence

$$\frac{1}{M_1^2} \mathbb{E}\left[\| \sum_{i \in S} h_i(w) - \nabla F_i(w)\|^2 \Big| M_1\right] \leqslant \frac{\phi^2}{M_1 K}$$

Next we bound ⓑ. Denote $y_i := \nabla F_i(w)$ and $\bar{y} := \frac{1}{N}\sum_{i=1}^N y_i = \nabla F(w)$. We claim ⓑ $=$ $\mathbb{E}\left[\|\sum_{i \in S} y_i - \bar{y}\|^2 \Big| M_1\right] \leqslant M_1\left(\frac{N-M_1}{N-1}\right)v^2$. Assume WLOG that $\bar{y} = 0$ (otherwise, consider $y_i' = y_i - \bar{y}$, which has mean 0). In what follows, we shall omit the "conditional on $M_1$" notation (but continue to condition on $M_1$) and denote by $\Omega$ the collection of all $\binom{N}{M_1}$ subsets of $[N]$ of size $M_1$. Now,

$$
\begin{aligned}
ⓑ &= \frac{1}{\binom{N}{M_1}} \sum_{S \in \Omega} \left\| \sum_{i \in S} y_i \right\|^2 \\
&= \frac{1}{\binom{N}{M_1}} \sum_{S \in \Omega} \left( \sum_{i \in S} \|y_i\|^2 + 2 \sum_{i,i' \in S, i < i'} \langle y_i, y_{i'} \rangle \right) \\
&= \frac{1}{\binom{N}{M_1}} \left( \binom{N-1}{M_1-1} \sum_{i=1}^N \|y_i\|^2 + 2\binom{N-2}{M_1-2} \sum_{1 \leqslant i < i' \leqslant N} \langle y_i, y_{i'} \rangle \right) \\
&= \frac{M_1}{N} \sum_{i=1}^N \|y_i\|^2 + 2\frac{M_1(M_1-1)}{N(N-1)} \sum_{1 \leqslant i < i' \leqslant N} \langle y_i, y_{i'} \rangle \\
&= \frac{M_1}{N} \left( \frac{M_1-1}{N-1} + \frac{N-M_1}{N-1} \right) \sum_{i=1}^N \|y_i\|^2 + \frac{2M_1(M_1-1)}{N(N-1)} \sum_{1 \leqslant i < i' \leqslant N} \langle y_i, y_{i'} \rangle \\
&= \frac{M_1}{N} \frac{M_1-1}{N-1} \left\| \sum_{i=1}^N y_i \right\|^2 + \frac{M_1}{N} \frac{N-M_1}{N-1} \sum_{i=1}^N \|y_i\|^2 \\
&= \frac{M_1}{N} \frac{N-M_1}{N-1} \sum_{i=1}^N \|y_i\|^2 \\
&\leqslant M_1 \left( \frac{N-M_1}{N-1} \right) v^2.
\end{aligned}
$$

Hence $\frac{1}{M_1^2}\mathbb{E}\left[\| \sum_{i \in S} \nabla F_i(w) - \nabla F(w)\|^2 \Big| M_1\right] \leqslant \frac{N-M_1}{N-1} \frac{v^2}{M_1}$. Finally, we take expectation over the randomness in $M_1$ and use $\mathbb{E}[1/M_1] = 1/M$ to arrive at the lemma. Also, the result clearly holds when $N = 1$ since the ⓑ term is zero when there is no variance in silo sampling (which is the case when $N = 1$). $\qquad\square$

*Proof of Theorem E.1.* **Privacy:** By post-processing (Dwork & Roth, 2014, Proposition 2.1), it suffices to show that the $R = n/K$ noisy stochastic gradients computed in line 7 of Algorithm 1 are $(\epsilon_0, \delta_0)$-ISRL-DP. Further, since the batches sampled locally are disjoint (because we sample locally *without replacement*), parallel composition (McSherry, 2009) implies that if each update in line 7 is $(\epsilon_0, \delta_0)$-ISRL-DP, then the full algorithm is $(\epsilon_0, \delta_0)$-ISRL-DP. Now recall that the Gaussian mechanism provides $(\epsilon_0, \delta_0)$-DP if $\sigma^2 \geqslant \frac{8\Delta_2^2 \ln(1.25/\delta_0)}{\epsilon_0^2}$, where $\Delta_2 = \sup_{w, X_i \sim X_i'} \|\frac{1}{K}\sum_{j=1}^K \nabla f(w, x_{i,j}) - \nabla f(w, x_{i,j}')\| = \sup_{w,x,x'} \frac{1}{K}\|\nabla f(w,x) - \nabla f(w,x')\| \leqslant \frac{2L}{K}$ is the $L_2$ sensitivity of the non-private gradient update in line 7 of Algorithm 1: this follows from (Bun & Steinke, 2016, Propositions 1.3 and 1.6) and our assumption $\epsilon_0 \leqslant 8 \ln(1/\delta_0)$. Therefore, conditional on the private transcript of all other silos, our choice of $\sigma^2$ implies that silo $i$'s transcript is $(\epsilon_0, \delta_0)$-DP for all $i \in [N]$, which means that One-Pass Noisy Accelerated Distributed MB-SGD is $(\epsilon_0, \delta_0)$-ISRL-DP.

**Excess loss:** 1. For the convex case, we choose $\lambda = \frac{V}{2D\sqrt{R}}$, where $V^2 = \frac{\phi^2}{MK} + \frac{v^2}{M}\mathbb{1}_{\{N>1\}}\left(\frac{N-M}{N-1}\right) + \frac{d\sigma^2}{M}$ is the variance of the noisy stochastic minibatch gradients, by Lemma E.3 for our noise with variance $\sigma^2 = \frac{32L^2\ln(1.25/\delta_0)}{K^2\epsilon_0^2}$. Now plugging $V^2$ into Lemma E.1, setting $R = n/K$, and $\lambda := \frac{V}{2D\sqrt{R}}$ yields

$$\mathbb{E}F(w_R^{ag}) - F^* \lesssim \frac{\beta D^2 K^2}{n^2} + \frac{\phi D}{\sqrt{nM}} + \frac{LD\sqrt{d\ln(1/\delta_0)}}{\epsilon_0\sqrt{KnM}} + \frac{\sqrt{K}vD}{\sqrt{nM}}\sqrt{\frac{N-M}{N-1}}\mathbb{1}_{\{N>1\}}.$$

Choosing $K = \left(\frac{L}{\beta D}\right)^{2/5}\frac{n^{3/5}(d\ln(1/\delta_0)^{1/5}}{\epsilon_0^{2/5}M^{1/5}}$ implies (30).

2. For strongly convex loss, we plug the same estimate for $V^2$ used above into Lemma E.2 with $R = n/K$ to obtain

$$\mathbb{E}F(w_R^{ag}) - F^* \lesssim \Delta\exp\left(\frac{-n}{K\sqrt{\kappa}}\right) + \frac{\phi^2}{\mu nM} + \frac{v^2 K}{\mu nM}\left(1 - \frac{M-1}{N-1}\right)\mathbb{1}_{\{N>1\}} + \frac{L^2}{\mu}\frac{d\ln(1/\delta_0)}{Kn\epsilon_0^2 M}.$$

Choosing $K = \frac{n}{\sqrt{\kappa}\ln\left(\mu\Delta\min\left\{\frac{\epsilon_0^2 n^2 M}{L^2 d\ln(1/\delta_0)}, \frac{nM}{\phi^2}\right\}\right)}$ yields (31). $\qquad\square$

# F   PROOFS AND SUPPLEMENTARY MATERIAL FOR SECTION 4

## F.1   PROOF OF THEOREM 4.1

We begin by considering $\beta$-smooth $f$.

**Theorem F.1** (Smooth ERM Upper Bound). *Assume $f(\cdot, x)$ is $\beta$-smooth for all $x$. Let $\epsilon_0 \leq 2\ln(2/\delta_0), \delta_0 \in (0,1)$, choose $K \geq \frac{\epsilon_0 n}{4\sqrt{2R\ln(2/\delta_0)}}$, and $\sigma^2 = \frac{256L^2 R\ln(\frac{2.5R}{\delta_0})\ln(2/\delta_0)}{n^2\epsilon_0^2}$. Then Algorithm 1 is $(\epsilon_0, \delta_0)$-ISRL-DP. Further:*

*1. If $f_\lambda(\cdot, x)$ is convex, then running Algorithm 1 on the regularized objective $\widetilde{f}(w, x) = f(w, x) + \frac{\lambda}{2}\|w - w_0\|^2$ with $R = \max\left(\left(\sqrt{\frac{\beta D}{L}}\frac{\sqrt{M}\epsilon_0 n}{\sqrt{d\ln(1/\delta_0)}}\right)^{1/2}, \mathbb{1}_{\{MK<Nn\}}\frac{\epsilon_0^2 n^2}{Kd\ln(1/\delta_0)}\right)$ (and $\lambda$ specified in the proof) yields*

$$\mathbb{E}\widehat{F}_{\mathbf{X}}(w_R^{ag}) - \widehat{F}_{\mathbf{X}}^* = \widetilde{\mathcal{O}}\left(LD\frac{\sqrt{d\ln(1/\delta_0)}}{\epsilon_0 n\sqrt{M}}\right). \tag{32}$$

*2. If $f(\cdot, x)$ is $\mu$-strongly convex, then running the multi-stage implementation (Appendix E.1) of Algorithm 1 with $R = \max\left\{\sqrt{\frac{\beta}{\mu}}\ln\left(\frac{\Delta_{\mathbf{X}}\mu M\epsilon_0^2 n^2}{L^2 d}\right), \mathbb{1}_{\{MK<Nn\}}\frac{\epsilon_0^2 n^2}{Kd\ln(1/\delta_0)}\right\}$ and $\Delta_{\mathbf{X}} \geq \widehat{F}_{\mathbf{X}}(w_0) - \widehat{F}_{\mathbf{X}}^*$ yields*

$$\mathbb{E}\widehat{F}_{\mathbf{X}}(w_R^{ag}) - \widehat{F}_{\mathbf{X}}^* = \widetilde{\mathcal{O}}\left(\frac{L^2}{\mu}\frac{d\ln(1/\delta_0)}{\epsilon_0^2 n^2 M}\right). \tag{33}$$

To prove Theorem F.1, we will require the following lemma:

**Lemma F.1** ((Lei et al., 2017)). *Let $\{a_l\}_{l\in[\widetilde{N}]}$ be an arbitrary collection of vectors such that $\sum_{l=1}^{\widetilde{N}} a_l = 0$. Further, let $\mathcal{S}$ be a uniformly random subset of $[\widetilde{N}]$ of size $\widetilde{M}$. Then,*

$$\mathbb{E}\left\|\frac{1}{\widetilde{M}}\sum_{l\in\mathcal{S}}a_l\right\|^2 = \frac{\widetilde{N}-\widetilde{M}}{(\widetilde{N}-1)\widetilde{M}}\frac{1}{\widetilde{N}}\sum_{l=1}^{\widetilde{N}}\|a_l\|^2 \leq \frac{\mathbb{1}_{\{\widetilde{M}<\widetilde{N}\}}}{\widetilde{M}\widetilde{N}}\sum_{l=1}^{\widetilde{N}}\|a_l\|^2.$$

*Proof of Theorem F.1.* **Privacy:** ISRL-DP of Algorithm 2 follows from ISRL-DP of the stochastic gradients $\{\widetilde{g}_r^i\}_{r=1}^R$ in line 7 of the algorithm (which was established in Theorem D.1) and the post-processing property of DP (Dwork & Roth, 2014, Proposition 2.1). Namely, since the choices of $\sigma^2, K$ given in Theorem F.1 ensure that silos' local stochastic minibatch gradients are ISRL-DP

and the iterates in Algorithm 1 are functions of these private noisy gradients (which do not involve additional data queries), it follows that the iterates themselves are ISRL-DP.

**Excess risk:** 1. For $\lambda = \frac{V_{\mathbf{X}}}{\|w_0 - w^*\| \sqrt{R}}$, Lemma E.1 implies

$$\mathbb{E}\widehat{F}_{\mathbf{X}}(\widehat{w}_R) - \widehat{F}_{\mathbf{X}}^* \lesssim \frac{\beta D^2}{R^2} + \frac{VD}{\sqrt{R}}, \tag{34}$$

where $V_{\mathbf{X}}^2 := \sup_{r \in [R]} \mathbb{E}\|\frac{1}{M_r}\sum_{i \in S_r} \tilde{g}_r^i - \nabla \widehat{F}_{\mathbf{X}}(w_r^{md})\|^2$ for $\tilde{g}_r^i = \frac{1}{K}\sum_{j=1}^{K} \nabla f(w_r^{md}, x_{i,j}^r) + u_i$ defined in line 7 of Algorithm 1. Now Lemma F.1 and $L$-Lipschitzness of $f(\cdot, x)$ implies that

$$V_{\mathbf{X}}^2 \leqslant \frac{\mathbb{1}_{\{M_r K < nN\}}}{M_r K n N} \sum_{i=1}^{N}\sum_{j=1}^{n} \sup_{w \in \mathcal{W}} \|\nabla f(w, x_{i,j}) - \nabla \widehat{F}_{\mathbf{X}}(w)\|^2 \leqslant \mathbb{1}_{\{M_r K < nN\}}\frac{4L^2}{M_r K},$$

conditional on $M_r$. Hence, taking total expectation with respect to $M_r$ and plugging this bound into (34) yields:

$$\mathbb{E}\widehat{F}_{\mathbf{X}}(\widehat{w}_R) - \widehat{F}_{\mathbf{X}}^* \lesssim \frac{\beta D^2}{R^2} + LD\left(\frac{\mathbb{1}_{\{MK<nN\}}}{\sqrt{MKR}} + \frac{\sqrt{d\ln^2(R/\delta_0)}}{\sqrt{M}\epsilon_0 n}\right),$$

by our choice of $\sigma^2$ and independence of noises $\{u_i\}_{i \in S_r}$ across silos. Then one verifies that the prescribed choice of $R$ yields (32).
2. Invoking Lemma E.2 with the same estimate for $V_{\mathbf{X}}^2$ obtained above gives

$$\mathbb{E}\widehat{F}_{\mathbf{X}}(\widehat{w}_R) - \widehat{F}_{\mathbf{X}}^* \lesssim \Delta_{\mathbf{X}} \exp\left(-\sqrt{\frac{\mu}{\beta}}R\right) + \frac{L^2}{\mu}\left(\frac{\mathbb{1}_{\{MK<nN\}}}{MKR} + \frac{d\ln^2(R\delta_0)}{M\epsilon_0^2 n^2}\right),$$

and plugging in the prescribed $R$ completes the proof. $\qquad\square$

We now re-state a precise form of Theorem 4.1 before providing its proof:

**Theorem F.2** (Precise Re-statement of Theorem 4.1). *Let $\epsilon_0 \leqslant 2\ln(2/\delta_0), \delta_0 \in (0,1)$, choose* $\sigma^2 = \frac{256L^2 R \ln(\frac{2.5R}{\delta_0})\ln(2/\delta_0)}{n^2\epsilon_0^2}$ *and* $K \geqslant \frac{\epsilon_0 n}{4\sqrt{2R\ln(2/\delta_0)}}$. *Then, Algorithm 1 is $(\epsilon_0, \delta_0)$-ISRL-DP. Further, there exist choices of $\beta > 0$ such that for Algorithm 2 run on $f_\beta(w, x) := \min_{v \in \mathcal{W}}\left(f(v, x) + \frac{\beta}{2}\|w - v\|^2\right)$, we have:*

*1. If $f(\cdot, x)$ is convex, then setting $R = \max\left(\frac{\sqrt{M}\epsilon_0 n}{\sqrt{d\ln(1/\delta_0)}}, \mathbb{1}_{\{MK<Nn\}}\frac{\epsilon_0^2 n^2}{Kd\ln(1/\delta_0)}\right)$ yields*

$$\mathbb{E}\widehat{F}_{\mathbf{X}}(w_R^{ag}) - \widehat{F}_{\mathbf{X}}^* = \tilde{\mathcal{O}}\left(LD\frac{\sqrt{d\ln(1/\delta_0)}}{\epsilon_0 n\sqrt{M}}\right). \tag{35}$$

*2. If $f(\cdot, x)$ is $\mu$-strongly convex and $R = \tilde{\mathcal{O}}\left(\max\left\{\frac{\sqrt{M}\epsilon_0 n}{\sqrt{d\ln(1/\delta_0)}}, \mathbb{1}_{\{MK<Nn\}}\frac{\epsilon_0^2 n^2}{Kd\ln(1/\delta_0)}\right\}\right)$ (in the multi-stage implementation in Appendix E.1), then*

$$\mathbb{E}\widehat{F}_{\mathbf{X}}(w_R^{ag}) - \widehat{F}_{\mathbf{X}}^* = \tilde{\mathcal{O}}\left(\frac{L^2}{\mu}\frac{d\ln(1/\delta_0)}{\epsilon_0^2 n^2 M}\right). \tag{36}$$

*Proof.* We established ISRL-DP in Theorem F.1.
**Excess risk:** 1. By Lemma D.7 and Theorem F.1, we have

$$\mathbb{E}\widehat{F}_{\mathbf{X}}(\widehat{w}_R) - \widehat{F}_{\mathbf{X}}^* \lesssim LD\frac{\sqrt{d\ln^2(R/\delta_0)}}{\sqrt{M}\epsilon_0 n} + \frac{L^2}{\beta},$$

if $R = \max\left(\left(\sqrt{\frac{\beta D}{L}}\frac{\sqrt{M}\epsilon_0 n}{\sqrt{d\ln(1/\delta_0)}}\right)^{1/2}, \mathbb{1}_{\{MK<Nn\}}\frac{\epsilon_0^2 n^2}{Kd\ln(1/\delta_0)}\right)$. Now choosing $\beta := \frac{\epsilon_0 n\sqrt{M}L}{\sqrt{d}D}$ yields both the desired excess risk and communication complexity bound.
2. By Lemma D.7 and Theorem F.1, we have

$$\mathbb{E}\widehat{F}_{\mathbf{X}}(\widehat{w}_R) - \widehat{F}_{\mathbf{X}}^* \lesssim \frac{L^2}{\mu}\frac{d\ln^2(R/\delta_0)}{M\epsilon_0^2 n^2} + \frac{L^2}{\beta},$$

if $R = \max\left\{\sqrt{\frac{\beta}{\mu}}\ln\left(\frac{\Delta_{\mathbf{x}}\mu M \epsilon_0^2 n^2}{L^2 d}\right), \mathbb{1}_{\{MK<Nn\}}\frac{\epsilon_0^2 n^2}{Kd\ln(1/\delta_0)}\right\}$. Now choosing $\beta := \frac{\mu M \epsilon_0^2 n^2}{d}$ yields both the desired excess risk and communication complexity bound. $\qquad\square$

**Remark F.1.** *The algorithm of (Girgis et al., 2021) requires $R = \widetilde{\Omega}(\epsilon_0^2 n^2 M/d)$ communications, making Algorithm 1 faster by a factor of $\min(\sqrt{M}\epsilon_0 n/\sqrt{d}, MK)$. If $M = N$ and full batches are used, the advantage of our algorithm over that of (Girgis et al., 2021) is even more significant.*

### F.2 Lower bounds for ISRL-DP Federated ERM

Formally, define the algorithm class $\mathbb{B}_{(\epsilon_0,\delta_0),C}$ to consist of those (sequentially interactive or $C$-compositional, ISRL-DP) algorithms $\mathcal{A} \in \mathbb{A}_{(\epsilon_0,\delta_0),C}$ such that for any $\mathbf{X} \in \mathbb{X}$, $f \in \mathcal{F}_{L,D}$, the expected empirical loss of the shuffled algorithm $\mathcal{A}_s$ derived from $\mathcal{A}$ is upper bounded by the expected loss of $\mathcal{A}$: $\mathbb{E}_{\mathcal{A},\{\pi_r\}_r}\widehat{F}(\mathcal{A}_s(\mathbf{X})) \lesssim \mathbb{E}_{\mathcal{A}}\widehat{F}(\mathcal{A}(\mathbf{X}))$. Here $\mathcal{A}_s$ denotes the algorithm that applies the randomizer $\mathcal{R}_r^{(i)}$ to $X_{\pi_r(i)}$ for all $i, r$, but otherwise behaves exactly like $\mathcal{A}$. This is not a very constructive definition but we will describe examples of algorithms in $\mathbb{B}_{(\epsilon_0,\delta_0),C}$. $\mathbb{B}_{(\epsilon_0,\delta_0),C}$ includes all $C$-compositional or sequentially interactive ISRL-DP algorithms that are symmetric with respect to each of the $N$ silos, meaning that the aggregation functions $g_r$ are symmetric (i.e. $g_r(Z_1, \cdots, Z_N) = g_r(Z_{\pi(1)}, \cdots Z_{\pi(N)})$ for all permutations $\pi$) and in each round $r$ the randomizers $\mathcal{R}_r^{(i)} = \mathcal{R}_r$ are the same for all silos $i \in [N]$. ($\mathcal{R}_r^{(i)}$ can still change with $r$ though.) For example, Algorithm 2 and Algorithm 1 are both in $\mathbb{B}_{(\epsilon_0,\delta_0)} := \mathbb{B}_{(\epsilon_0,\delta_0),1}$. This is because the aggregation functions used in each round are simple averages of the $M = N$ noisy gradients received from all silos (and they are compositional) and the randomizers used by every silo in round $r$ are identical: each adds the same Gaussian noise to the stochastic gradients. $\mathbb{B}$ also includes sequentially interactive algorithms that choose the order in which silos are processed uniformly at random. This is because the distributions of the updates of $\mathcal{A}$ and $\mathcal{A}_s$ are both averages over all permutations of $[N]$ of the conditional (on $\pi$) distributions of the randomizers applied to the $\pi$-permuted database. If $\mathcal{A} \in \mathbb{B}_{(\epsilon_0,\delta_0),C}$ is sequentially interactive or compositional, we write $\mathcal{A} \in \mathbb{B}$. We now state and prove our ERM lower bounds:

**Theorem F.3.** *Let $\epsilon_0 \in (0, \sqrt{N}]$, $\delta_0 = o(1/nN)$ and $\mathcal{A} \in \mathbb{B}_{(\epsilon_0,\delta_0),C}$ such that in every round $r \in [R]$, the local randomizers $\mathcal{R}_r^{(i)}(\mathbf{Z}_{(1:r-1)}, \cdot) : \mathcal{X}^n \to \mathcal{Z}$ are $(\epsilon_0^r, \delta_0^r)$-DP for all $i \in [N]$, $\mathbf{Z}_{(1:r-1)} \in \mathcal{Z}^{r-1\times N}$, with $\epsilon_0^r \leqslant \frac{1}{n}$, $\delta_0^r = o(1/nNR)$, and $N \geqslant 16\ln(2/\delta_0^r n)$. Then:*
*1. there exists a (linear, hence $\beta$-smooth $\forall \beta \geqslant 0$) loss function $f \in \mathcal{F}_{L,D}$ and a database $\mathbf{X} \in \mathcal{X}^{nN}$ (for some $\mathcal{X}$) such that:*

$$\mathbb{E}\widehat{F}_{\mathbf{X}}(\mathcal{A}(\mathbf{X})) - \widehat{F}_{\mathbf{X}}^* = \widetilde{\Omega}\left(LD\min\left\{1, \frac{\sqrt{d}}{\epsilon_0 n\sqrt{N}C^2}\right\}\right).$$

*2. There exists a ($\mu$-smooth) $f \in \mathcal{G}_{\mu,L,D}$ and database $\mathbf{X} \in \mathcal{X}^{nN}$ such that*

$$\mathbb{E}\widehat{F}_{\mathbf{X}}(\mathcal{A}(\mathbf{X})) - \widehat{F}_{\mathbf{X}}^* = \widetilde{\Omega}\left(LD\min\left\{1, \frac{d}{\epsilon_0^2 n^2 NC^4}\right\}\right).$$

*Further, if $\mathcal{A} \in \mathbb{B}$, then the above lower bounds hold with $C = 1$.*

*Proof.* **Step 1** is identical to Step 1 of the proof of Theorem 2.2. **Step 2** is very similar to Step 2 in the proof of Theorem 2.2, but now we use Theorem F.4 (below) instead of Theorem D.6 to lower bound the excess empirical loss of $\mathcal{A}_s$. **Step 3:** Finally, the definition of $\mathbb{B}$ implies that the excess risk of $\mathcal{A}$ is the same as that of $\mathcal{A}_s$, hence the lower bound also applies to $\mathcal{A}$. $\qquad\square$

**Theorem F.4.** *(Bassily et al., 2014) Let $\mu, D, \epsilon > 0$, $L \geqslant \mu D$, and $\delta = o(1/nN)$. Consider $\mathcal{X} := \{\frac{-D}{\sqrt{d}}, \frac{D}{\sqrt{d}}\}^d \subset \mathbb{R}^d$ and $\mathcal{W} := B_2(0, D) \subset \mathbb{R}^d$. Let $\mathcal{A} : \mathcal{X}^{nN} \to \mathcal{W}$ be any $(\epsilon, \delta)$-CDP algorithm. Then:*
*1. There exists a ($\mu = 0$) convex, linear ($\beta$-smooth for any $\beta$), L-Lipschitz loss $f : \mathcal{W} \times \mathcal{X} \to \mathbb{R}$ and a database $\mathbf{X} \in \mathcal{X}^{nN}$ such that the expected empirical loss of $\mathcal{A}$ is lower bounded as*

$$\mathbb{E}\widehat{F}_{\mathbf{X}}(\mathcal{A}(\mathbf{X})) - \widehat{F}_{\mathbf{X}}^* = \widetilde{\Omega}\left(LD\min\left\{1, \frac{\sqrt{d}}{\epsilon nN}\right\}\right).$$

2. *There exists a $\mu$-strongly convex, $\mu$-smooth, L-Lipschitz loss $f : \mathcal{W} \times \mathcal{X} \to \mathbb{R}$ and a database $\mathbf{X} \in \mathcal{X}^{nN}$ such that the expected empirical loss of $\mathcal{A}$ is lower bounded as*

$$\mathbb{E}\widehat{F}_{\mathbf{X}}(\mathcal{A}(\mathbf{X})) - \widehat{F}_{\mathbf{X}}^* = \widetilde{\Omega}\left(LD\min\left\{1, \frac{d}{\epsilon^2 n^2 N^2}\right\}\right).$$

# G PROOFS AND SUPPLEMENTARY MATERIALS FOR SECTION 5

## G.1 PROOF OF THEOREM 5.1

**Theorem G.1** (Precise version of Theorem 5.1). *Let $\epsilon \leqslant \ln(2/\delta)$, $\delta \in (0,1)$, and $M \geqslant 16\ln(18RM^2/N\delta)$ for (polynomial) $R$ specified in the proof. Then, there is a constant $C > 0$ such that setting $\sigma^2 := \frac{CL^2RM\ln(RM^2/N\delta)\ln(R/\delta)\ln(1/\delta)}{n^2N^2\epsilon^2}$ ensures that the shuffled version of Algorithm 2 is $(\epsilon, \delta)$-CDP. Moreover, there exist $\eta_r = \eta$ and $\{\gamma_r\}_{r=0}^{R-1}$ such that the shuffled version of Algorithm 2 achieves the following upper bounds:*
*1. If $f(\cdot, x)$ is convex, then*

$$\mathbb{E}F(\widehat{w}_R) - F^* = \widetilde{\mathcal{O}}\left(LD\left(\frac{1}{\sqrt{nM}} + \frac{\sqrt{d\ln(1/\delta)}}{\epsilon nN}\right)\right). \tag{37}$$

*2. If $f(\cdot, x)$ is $\mu$-strongly convex, then*

$$\mathbb{E}F(\widehat{w}_R) - F^* = \widetilde{\mathcal{O}}\left(\frac{L^2}{\mu}\left(\frac{1}{nM} + \frac{d\ln(1/\delta)}{\epsilon^2 n^2 N^2}\right)\right). \tag{38}$$

*Proof.* We fix $K = 1$ for simplicity, but note that $K > 1$ can also be used (see Lemma 3 in (Girgis et al., 2021)). We shall also assume WLOG that $f(\cdot, x)$ is $\beta$-smooth: the reduction to non-smooth $f$ follows by Nesterov smoothing, as in the proofs of Theorems 2.1 and 4.1. Our choices of $R$ shall be: $R := \max\left(\frac{n^2N^2\epsilon^2}{M}, \frac{N}{M}, \min\left\{n, \frac{\epsilon^2 n^2 N^2}{dM}\right\}, \frac{\beta D}{L}\min\left\{\sqrt{nM}, \frac{\epsilon nN}{\sqrt{d}}\right\}\right)$ for convex $f$; and $R := \max\left(\frac{n^2N^2\epsilon^2}{M}, \frac{N}{M}, \frac{8\beta}{\mu}\ln\left(\frac{\beta D^2\mu\epsilon^2 n^2 N^2}{dL^2}\right), \min\left\{n, \frac{\epsilon^2 n^2 N^2}{dM}\right\}\right)$ for strongly convex.

**Privacy:** Observe that in each round $r$, the model updates of the shuffled algorithm $\mathcal{A}_s^r$ can be viewed as post-processing of the composition $\mathcal{M}_r(\mathbf{X}) = \mathcal{S}_M \circ samp_{M,N}(Z_r^{(1)}, \cdots, Z_r^{(N)})$, where $\mathcal{S}_M$ uniformly randomly shuffles the $M$ received reports, $samp_{M,N}$ is the mechanism that chooses $M$ reports uniformly at random from $N$, and $Z_r^{(i)} = samp_{1,n}(\widehat{\mathcal{R}}_r(x_{i,1}), \cdots \widehat{\mathcal{R}}_r(x_{i,n}))$, where $\widehat{\mathcal{R}}_r(x) := \nabla f(w_r, x) + u$ and $u \sim \mathcal{N}(0, \sigma^2 \mathbf{I}_d)$. Recall (Theorem A.1 in (Dwork & Roth, 2014)) that $\sigma^2 = \frac{8L^2\ln(2/\widehat{\delta_0})}{\widehat{\epsilon_0}^2}$ suffices to ensure that $\widehat{\mathcal{R}}_r$ is $(\widehat{\epsilon_0}, \widehat{\delta_0})$-DP if $\widehat{\epsilon_0} \leqslant 1$. Now note that $\mathcal{M}_r(\mathbf{X}) = \widetilde{\mathcal{R}}^M(\mathcal{S}_M \circ samp_{M,N}(X_1, \cdots X_N))$, where $\widetilde{\mathcal{R}} : \mathcal{X}^n \to \mathcal{Z}$ is given by $X \mapsto samp_{1,n}(\widehat{\mathcal{R}}(x_1), \cdots, \widehat{\mathcal{R}}(x_n))$ and $\widetilde{\mathcal{R}}^M : \mathcal{X}^{nM} \to \mathcal{Z}^M$ is given by $\mathbf{X} \mapsto (\widetilde{\mathcal{R}}(X_1), \cdots \widetilde{\mathcal{R}}(X_M))$ for any $\mathbf{X} = (X_1, \cdots, X_M) \in \mathcal{X}^{nM}$. This is because we are applying the same randomizer (same additive Gaussian noise) across silos and the operators $\mathcal{S}_M$ and $\widetilde{\mathcal{R}}^M$ commute. (Also, applying a randomizer to all $N$ silos and then randomly choosing $M$ reports is equivalent to randomly choosing $M$ silos and then applying the same randomizer to all $M$ of these silos.) Therefore, conditional on the random subsampling of $M$ out of $N$ silos (denoted $(X_1, \cdots, X_M)$ for convenience), Theorem 3.8 in (Feldman et al., 2020b) implies that $(\widehat{\mathcal{R}}(x_{\pi(1),1}), \cdots, \widehat{\mathcal{R}}(x_{\pi(1),n}), \cdots, \cdots, \widehat{\mathcal{R}}(x_{\pi(M),1}), \cdots, \widehat{\mathcal{R}}(x_{\pi(M),n}))$ is $(\widehat{\epsilon}, \widehat{\delta})$-CDP, where $\widehat{\epsilon} = \mathcal{O}\left(\frac{\widehat{\epsilon_0}\sqrt{\ln(1/M\widehat{\delta_0})}}{\sqrt{M}}\right)$ and $\widehat{\delta} = 9M\widehat{\delta_0}$, provided $\widehat{\epsilon_0} \leqslant 1$ and $M \geqslant 16\ln(2/\widehat{\delta_0})$ (which we will see is satisfied by our assumption on $M$). Next, privacy amplification by subsampling (see (Ullman, 2017) and Lemma 3 in (Girgis et al., 2021)) silos and local samples implies that $\mathcal{M}_r$ is $(\epsilon^r, \delta^r)$-CDP, where $\epsilon^r = \frac{2\widehat{\epsilon}M}{nN} = \mathcal{O}\left(\widehat{\epsilon_0}\frac{\sqrt{M\ln(1/M\widehat{\delta_0})}}{nN}\right)$ and $\delta^r = \frac{M}{nN}\widehat{\delta} = \frac{9M^2}{nN}\widehat{\delta_0}$. Finally, by the advanced composition theorem (Theorem 3.20 in (Dwork & Roth, 2014)), to ensure $\mathcal{A}_s$ is $(\epsilon, \delta)$-CDP, it suffices to make each round $(\epsilon^r := \frac{\epsilon}{2\sqrt{2R\ln(1/\delta)}}, \delta^r := \delta/2R)$-CDP. Using the two equations to solve for $\widehat{\epsilon_0} = \frac{CnN\epsilon}{\sqrt{R\ln(1/\delta)\ln(RM/nN\delta)M}}$ for some $C > 0$ and $\widehat{\delta_0} = \frac{nN\delta}{18RM^2}$, we see that

$\sigma^2 = \mathcal{O}\left(\frac{L^2 \ln(RM^2/N\delta)\ln(R/\delta)\ln(1/\delta))RM}{n^2 N^2 \epsilon^2}\right)$ ensures that $\mathcal{A}_s$ is $(\epsilon, \delta)$-CDP, i.e. that $\mathcal{A}$ is $(\epsilon, \delta)$-SDP. Note that our choices of $R$ in the theorem (specifically $R \geqslant N/M$ and $R \geqslant \frac{n^2 N^2 \epsilon^2}{M}$) ensure that $\widehat{\delta}_0, \delta \leqslant 1$ and $\widehat{\epsilon}_0 \lesssim 1$, so that Theorem 3.8 in (Feldman et al., 2020b) indeed gives us the amplification by shuffling result used above.

**Excess risk:** Note that shuffling does not affect the uniform stability of the algorithm. So we proceed similarly to the proof of Theorem 2.1, except $\sigma^2$ is now smaller.

1. *Convex case:* Set $\gamma_r = \gamma = 1/R$ for all $r$. Now Lemma D.2, Lemma D.1, and Lemma D.6 (with $\sigma^2$ in the lemma replaced by the $\sigma^2$ prescribed here) together imply for any $\eta \leqslant 1/\beta$ that

$$\mathbb{E}F(\widehat{w}_R) - F^* \lesssim \frac{L^2 R\eta}{nM} + \frac{D^2}{\eta R} + \eta\left(L^2/M + \frac{d\sigma^2}{M}\right).$$

Now plugging in $\eta := \min\left\{1/\beta, \frac{D\sqrt{M}}{LR}\min\left\{\sqrt{n}, \frac{\epsilon n N}{\sqrt{dM \ln(RM^2/N\delta)\ln(R/\delta)\ln(1/\delta)}}\right\}\right\}$ yields

$$\mathbb{E}F(\widehat{w}_R) - F^* \lesssim LD\left(\max\left\{\frac{1}{\sqrt{nM}}, \frac{\sqrt{d\ln(RM^2/N\delta)\ln(R/\delta)\ln(1/\delta)}}{\epsilon n N}\right\}\right)$$
$$+ \frac{LD}{R\sqrt{M}}\min\left\{\sqrt{n}, \frac{\epsilon n N}{\sqrt{dM \ln(RM^2/N\delta)\ln(R/\delta)\ln(1/\delta)}}\right\} + \frac{\beta D^2}{R}.$$

Then one can verify that plugging in the prescribed $R$ yields the stated excess population loss bound.

2. *$\mu$-strongly convex case:*

By Lemma D.6 (and its proof), we know there exists $\eta \leqslant 1/\beta$ and $\{\gamma_r\}_{r=1}^R$ such that

$$\mathbb{E}\widehat{F}_{\mathbf{X}}(\widehat{w}_R) - \widehat{F}_{\mathbf{X}}^* = \widetilde{\mathcal{O}}\left(\beta D^2 \exp\left(-\frac{R}{2\kappa}\right) + \frac{L^2}{\mu R} + \frac{d\sigma^2}{\mu M R}\right).$$

Hence Lemma D.1 and Lemma D.2 imply

$$\mathbb{E}F(\widehat{w}_R) - F^* = \widetilde{\mathcal{O}}\left(\beta D^2 \exp\left(-\frac{R}{2\kappa}\right) + \frac{L^2}{\mu R} + \frac{L^2 d \ln(1/\delta)}{\mu\epsilon^2 n^2 N^2} + \frac{L^2}{\mu M n}\right).$$

Then one verifies that the prescribed $R$ is large enough to achieve the stated excess population loss bound. $\qquad\square$

## G.2 SDP One-Pass Accelerated Noisy MB-SGD and the Proof of Theorem 5.2

To develop an SDP variation of One-Pass Accelerated Noisy MB-SGD, we will use the binomial noise-based protocol of (Cheu et al., 2021) (described in Algorithm 4) instead of using the Gaussian mechanism and amplification by shuffling. This is because for our one-pass algorithm, amplification by shuffling would result in an impractical restriction on $\epsilon$. Algorithm 4 invokes the SDP scalar summation subroutine Algorithm 5.

We recall the privacy and accuracy guarantees of Algorithm 4 below:

**Lemma G.1** ((Cheu et al., 2021)). *For any $0 < \epsilon \leqslant 15, 0 < \delta < 1/2, d, N \in \mathbb{N}$, and $L > 0$, there are choices of parameters $b, g \in \mathbb{N}$ and $p \in (0, 1/2)$ for $\mathcal{P}_{ID}$ (Algorithm 5) such that, for $\mathbf{X} = (\mathbf{x}_1, \cdots \mathbf{x}_N)$ containing vectors of maximum norm $\max_{i\in[N]}\|\mathbf{x}_i\| \leqslant L$, the following holds: 1) $\mathcal{P}_{vec}$ is $(\epsilon, \delta)$-SDP; and 2) $\mathcal{P}_{vec}(\mathbf{X})$ is an unbiased estimate of $\sum_{i=1}^N \mathbf{x}_i$ with bounded variance*

$$\mathbb{E}\left[\left\|\mathcal{P}_{vec}(\mathbf{X}; \epsilon, \delta; L) - \sum_{i=1}^N \mathbf{x}_i\right\|^2\right] = \mathcal{O}\left(\frac{dL^2 \log^2\left(\frac{d}{\delta}\right)}{\epsilon^2}\right).$$

With these building blocks in hand, we provide our SDP One-Pass Accelerated Noisy MB-SGD algorithm in Algorithm 6. We now provide the general version of Theorem 5.2 for $M \leqslant N$:

---

**Algorithm 4** $\mathcal{P}_{\text{vec}}$, a shuffle protocol for vector summation (Cheu et al., 2021)

---

1: **Input:** database of $d$-dimensional vectors $\mathbf{X} = (\mathbf{x}_1, \cdots, \mathbf{x}_N)$; privacy parameters $\epsilon, \delta$; $L$.
2: **procedure:** Local Randomizer $\mathcal{R}_{\text{vec}}(\mathbf{x}_i)$
3:     **for** $j \in [d]$ **do**
4:         Shift component to enforce non-negativity: $\mathbf{w}_{i,j} \leftarrow \mathbf{x}_{i,j} + L$
5:         $\mathbf{m}_j \leftarrow \mathcal{R}_{1D}(\mathbf{w}_{i,j})$
6:     **end for**
7:     Output labeled messages $\{(j, \mathbf{m}_j)\}_{j \in [d]}$
8: **end procedure**
9: **procedure: Analyzer** $\mathcal{A}_{\text{vec}}(\mathbf{y})$
10:     **for** $j \in [d]$ **do**
11:         Run analyzer on coordinate $j$'s messages $z_j \leftarrow \mathcal{A}_{1D}(\mathbf{y}_j)$
12:         Re-center: $o_j \leftarrow z_j - L$
13:     **end for**
14:     Output the vector of estimates $\mathbf{o} = (o_1, \cdots o_d)$
15: **end procedure**

---

**Algorithm 5** $\mathcal{P}_{1D}$, a shuffle protocol for summing scalars (Cheu et al., 2021)

---

1: **Input:** Scalar database $X = (x_1, \cdots x_N) \in [0, L]^N$; $g, b \in \mathbb{N}$; $p \in (0, \frac{1}{2})$.
2: **procedure: Local Randomizer** $\mathcal{R}_{1D}(x_i)$
3:     $\bar{x}_i \leftarrow \lfloor x_i g / L \rfloor$.
4:     Sample rounding value $\eta_1 \sim \mathbf{Ber}(x_i g / L - \bar{x}_i)$.
5:     Set $\hat{x}_i \leftarrow \bar{x}_i + \eta_1$.
6:     Sample privacy noise value $\eta_2 \sim \mathbf{Bin}(b, p)$.
7:     Report $\mathbf{y}_i \in \{0, 1\}^{g+b}$ containing $\hat{x}_i + \eta_2$ copies of 1 and $g + b - (\hat{x}_i + \eta_2)$ copies of 0.
8: **end procedure**
9: **procedure: Analyzer** $\mathcal{A}_{1D}(\mathcal{S}(\mathbf{y}))$
10:     Output estimator $\frac{L}{g}((\sum_{i=1}^{N} \sum_{j=1}^{b+g} (\mathbf{y}_i)_j) - pbn)$.
11: **end procedure**

---

---

**Algorithm 6** SDP Accelerated Noisy MB-SGD

---

1: **Input:** Data $X_i \in \mathcal{X}_i^n$, $i \in [N]$, strong convexity modulus $\mu \geqslant 0$, privacy parameters $(\epsilon, \delta)$, iteration number $R \in \mathbb{N}$, batch size $K \in [n]$, step size parameters $\{\eta_r\}_{r \in [R]}, \{\alpha_r\}_{r \in [R]}$.

2: Initialize $w_0^{ag} = w_0 \in \mathcal{W}$ and $r = 1$.

3: **for** $r \in [R]$ **do**

4:      Server updates and broadcasts $w_r^{md} = \frac{(1-\alpha_r)(\mu+\eta_r)}{\eta_r + (1-\alpha_r^2)\mu} w_{r-1}^{ag} + \frac{\alpha_r[(1-\alpha_r)\mu + \eta_r]}{\eta_r + (1-\alpha_r^2)\mu} w_{r-1}$

5:      **for** $i \in S_r$ **in parallel do**

6:          Silo $i$ draws $\{x_{i,j}^r\}_{j=1}^K$ from $X_i$ (without replacement) and computes $Z_i^r := \{\nabla f(w_r^{md}, x_{i,j}^r)\}_{j=1}^K$.

7:      **end for**

8:      Server receives $\widetilde{g}_r := \frac{1}{M_r K} \mathcal{P}_{\text{vec}}(\{Z_i^r\}_{i \in S_r}; \epsilon, \delta; L)$.

9:      Server computes $w_r := \text{argmin}_{w \in \mathcal{W}} \left\{ \alpha_r \left[ \langle \widetilde{g}_r, w \rangle + \frac{\mu}{2} \|w_r^{md} - w\|^2 \right] + \left[ (1-\alpha_r)\frac{\mu}{2} + \frac{\eta_r}{2} \right] \|w_{r-1} - w\|^2 \right\}$.

10:      Server updates and broadcasts $w_r^{ag} = \alpha_r w_r + (1-\alpha_r)w_{r-1}^{ag}$.

11: **end for**

12: **Output:** $w_R^{ag}$.

---

**Theorem G.2** (Complete version of Theorem 5.2). *Let $f(\cdot, x)$ be $\beta$-smooth $\forall x \in \mathcal{X}$. Assume $\epsilon \leqslant 15, \delta \in (0, \frac{1}{2})$. Then, Algorithm 6 is $(\epsilon, \delta)$-SDP. Moreover, there are choices of stepsize, batch size, and $\lambda > 0$ such that (for $\upsilon$ defined in (29)):*
*1. Running Algorithm 6 on $\widetilde{f}(w, x) := f(w, x) + \frac{\lambda}{2}\|w - w_0\|^2$ (where $w_0 \in \mathcal{W}$) yields*

$$\mathbb{E}F(w_R^{ag}) - F^* = \mathcal{O}\left( \frac{\phi D}{\sqrt{nM}} + \left( \frac{\beta^{1/4} L D^{3/2}\sqrt{d}\ln(d/\delta)}{\epsilon nM} \right)^{4/5} + \sqrt{\frac{N-M}{N-1}} \mathbb{1}_{\{N>1\}} \frac{\upsilon L^{1/5} D^{4/5}}{\beta^{1/5}} \left( \frac{\sqrt{d}\ln(d/\delta)}{\epsilon nM^{3.5}} \right)^{1/5} \right).$$
(39)

*2. If $f(\cdot, x)$ is $\mu$-strongly convex $\forall x \in \mathcal{X}$ and $\kappa = \frac{\beta}{\mu}$, then running the Multi-Stage Implementation of Algorithm 6 (recall Appendix E.1) directly on $f$ yields*

$$\mathbb{E}F(w_R^{ag}) - F^* = \widetilde{\mathcal{O}}\left( \frac{\phi^2}{nM} + \frac{L^2}{\mu} \frac{\sqrt{\kappa}d\ln(1/\delta)}{\epsilon^2 n^2 M^2} + \frac{\upsilon^2}{\mu\sqrt{\kappa}M} \left(1 - \frac{M-1}{N-1}\right) \mathbb{1}_{\{N>1\}} \right).$$
(40)

**Remark G.1.** *1. For convex $f$, if $M = N$ or*

$$\upsilon \lesssim \sqrt{\frac{N-1}{N-M}} \left[ (L^3 D^2 \beta^2)^{1/5} \frac{(d\ln(d/\delta))^{3/10}}{M^{1/10}n^{3/5}\epsilon^{3/5}} + \phi \left(\frac{\beta D}{L}\right)^{1/5} \left(\frac{\epsilon^2 M^2}{n^3 d\ln^2(d/\delta)}\right)^{1/10} \right],$$

*then (39) recovers the bound (10) in Theorem 3.1 ($M = N$ version), with $N$ replaced by $M$.*
*2. For $\mu$-strongly convex $f$, if $M = N$ or*

$$\upsilon^2 \lesssim \left(\frac{N-1}{N-M}\right) \sqrt{\kappa} \left(\frac{\phi^2}{n} + \frac{\sqrt{\kappa}L^2 d\ln(1/\delta)}{M\epsilon^2 n^2}\right),$$

*then (40) recovers the bound (11) in Theorem 3.1, with $N$ replaced by $M$.*
*Also, note that appealing to privacy amplification by subsampling would result in tighter excess risk bounds than those stated in Theorem G.2 when $M < N$, but would require a restriction $\epsilon \lesssim M/N$. To avoid this restriction, we do not invoke privacy amplification by subsampling in our analysis.*

*Proof of Theorem G.2.* **Privacy:** By post-processing (Dwork & Roth, 2014, Proposition 2.1), it suffices to show that the $R = n/K$ noisy stochastic gradients computed in line 8 of Algorithm 6 are $(\epsilon, \delta)$-SDP. Further, since the batches sampled locally are disjoint (because we sample locally *without replacement*), parallel composition (McSherry, 2009) implies that if each update in line 8 is $(\epsilon, \delta)$-SDP, then the full algorithm is $(\epsilon, \delta)$-SDP. Since $f(\cdot, x)$ is $L$-Lipschitz, it follows directly from Lemma G.1 that each update in line 8 is $(\epsilon, \delta)$-SDP.

**Excess loss:** 1. For the convex case, we choose $\lambda = \frac{V}{2D\sqrt{R}}$, where $V^2 = \frac{\phi^2}{MK} + \frac{\upsilon^2}{M}\mathbb{1}_{\{N>1\}}\left(\frac{N-M}{N-1}\right) + \text{Var}(\widetilde{g}_r)$ is the conditional variance of the noisy stochastic minibatch gradients given $M_r$, by Lemma E.3. Also, conditional on $M_r$, we have $\text{Var}(\widetilde{g}_r) \lesssim \frac{dL^2 \ln^2(d/\delta)}{\epsilon^2 M_r^2 K^2}$

by Lemma G.1 and independence of the data. Hence, taking total expectation over $M_r$, we get $V^2 = \frac{\phi^2}{MK} + \frac{v^2}{M}\mathbb{1}_{\{N>1\}}\left(\frac{N-M}{N-1}\right) + \frac{dL^2\ln^2(d/\delta)}{\epsilon^2M^2K^2}$. Now plugging $V^2$ into Lemma E.1, setting $R = n/K$, and $\lambda := \frac{V}{2D\sqrt{R}}$ yields

$$\mathbb{E}F(w_R^{ag}) - F^* \lesssim \frac{\beta D^2 K^2}{n^2} + \frac{\phi D}{\sqrt{nM}} + \frac{LD\sqrt{d}\ln(d/\delta)}{\epsilon M\sqrt{Kn}} + \frac{\sqrt{K}vD}{\sqrt{nM}}\sqrt{\frac{N-M}{N-1}}\mathbb{1}_{\{N>1\}}.$$

Choosing $K = \left(\frac{L}{\beta D}\right)^{2/5}\frac{n^{3/5}(d\ln(1/\delta_0)^{1/5}}{\epsilon_0^{2/5}M^{2/5}}$ implies (39).

2. For strongly convex loss, we plug the same estimate for $V^2$ used above into Lemma E.2 with $R = n/K$ to obtain

$$\mathbb{E}F(w_R^{ag}) - F^* \lesssim \Delta\exp\left(\frac{-n}{K\sqrt{\kappa}}\right) + \frac{\phi^2}{\mu nM} + \frac{v^2K}{\mu nM}\left(1 - \frac{M-1}{N-1}\right)\mathbb{1}_{\{N>1\}} + \frac{L^2}{\mu}\frac{d\ln^2(d/\delta)}{Kn\epsilon^2M^2},$$

where $\Delta \geqslant F(w_0) - F^*$. Choosing $K = \frac{n}{\sqrt{\kappa}\ln\left(\mu\Delta\min\left\{\frac{\epsilon^2n^2M^2}{L^2d\ln(1/\delta_0)},\frac{nM}{\phi^2}\right\}\right)}$ yields (40). $\square$

# H   ISRL-DP UPPER BOUNDS WITH UNBALANCED DATA SET SIZES AND DIFFERING PRIVACY NEEDS ACROSS SILOS

In order to state the generalized versions of our upper bounds (for arbitrary $n_i, \epsilon_i, \delta_i, i \in [N]$), we will require some additional notation and assumptions.

## H.1   ADDITIONAL NOTATION AND ASSUMPTIONS

First, we define a generalization of $(\epsilon_0, \delta_0)$-ISRL-DP (as it was formally defined in Appendix B) that allows for differing privacy parameters across silos:

**Definition 10.** *(Generalized Inter-Silo Record-Level Differential Privacy)   Let $\rho_i(X_i, X_i') := \sum_{j=1}^{n_i}\mathbb{1}_{\{x_{i,j}\neq x_{i,j}'\}}$, $i \in [N]$. A randomized algorithm $\mathcal{A}$ is $\{(\epsilon_i, \delta_i)\}_{i=1}^N$-ISRL-DP if for all silos $i$ and all $\rho_i$-adjacent $X_i, X_i'$,*

$$(\mathcal{R}_1^{(i)}(X_i), \mathcal{R}_2^{(i)}(\mathbf{Z}_1, X_i), \cdots, \mathcal{R}_R^{(i)}(\mathbf{Z}_{1:R-1}, X_i)) \underset{(\epsilon_i,\delta_i)}{\simeq} (\mathcal{R}_1^{(i)}(X_i'), \mathcal{R}_2^{(i)}(\mathbf{Z}_1', X_i'), \cdots, \mathcal{R}_R^{(i)}(\mathbf{Z}_{1:R-1}', X_i')),$$

*where $\mathbf{Z}_r := \{\mathcal{R}_r^{(i)}(\mathbf{Z}_{1:r-1}, X_i)\}_{i=1}^N$ and $\mathbf{Z}_r' := \{\mathcal{R}_r^{(i)}(\mathbf{Z}_{1:r-1}', X_i')\}_{i=1}^N$.*

We also allow for the weights put on each silo in the FL objective to differ and consider:

$$\min_{w\in\mathcal{W}}\left\{F(w) := \sum_{i=1}^N p_iF_i(w)\right\},$$

where $p_i \in [0,1]$ and $\sum_{i=1}^N p_i = 1$. However, we will present our results for the case where $p_i = \frac{1}{N}$ for all $i \in [N]$. This is without loss of generality: given any $\widetilde{F}(w) = \sum_{i=1}^N p_iF_i(w)$, we have $\widetilde{F}(w) = \sum_{i=1}^N p_iF_i(w) = \frac{1}{N}\sum_{i=1}^N \widetilde{F}_i(w)$, where $\widetilde{F}_i(w) = Np_iF_i(w) = \mathbb{E}_{x_i\sim\mathcal{D}_i}[Np_if(w, x_i)] := \mathbb{E}_{x_i\sim\mathcal{D}_i}[\widetilde{f}_i(w, x_i)]$. Thus, our results for the case of $p_i = 1/N$ apply for general $p_i$, but $L$ gets replaced with $\widetilde{L} = \max_{i\in[N]}p_iNL$, $\mu$ gets replaced with $\widetilde{\mu} = \min_{i\in[N]}p_iN\mu$, and $\beta$ gets replaced with $\widetilde{\beta} = \max_{i\in[N]}p_iN\beta$.

We will choose batch sizes $K_i$ such that $K_i/n_i = K_l/n_l$ for all $i, l \in [N]$ in each round, and denote $K = \min_{i\in[N]}K_i$. In addition to the assumptions we made in the main body, we also refine the assumption on $\{M_r\}$ to include a description of the second moment of $1/M_r$:

**Assumption 2.** *In each round $r$, a uniformly random subset $S_r$ of $M_r \in [N]$ distinct silos is available to communicate with the server, where $\{M_r\}_{r\geqslant 0}$ are independent random variables with $\frac{1}{M} := \mathbb{E}(\frac{1}{M_r})$ and $\frac{1}{M'} := \sqrt{\mathbb{E}(\frac{1}{M_r^2})}$.*

For $M \in [N]$, denote

$$\overline{\sigma_M^2} := \frac{1}{M} \sum_{i=1}^{M} \sigma_{(i)}^2,$$

where $\sigma_{(1)}^2 := \sigma_{\max}^2 := \max_{i \in [N]} \sigma_i^2 \geqslant \sigma_{(2)}^2 \geqslant \cdots \geqslant \sigma_{(N)}^2 := \sigma_{\min}^2 := \min_{i \in [N]} \sigma_i^2,$. (More generally, whenever a bar and $M$ subscript are appended to a parameter, it denotes the average of the $M$ largest values.) Also, define

$$\Sigma^2 := \sqrt{\mathbb{E}(\overline{\sigma_{M_1}^2})^2}$$

for any $\{\sigma_i^2\}_{i=1}^{N} \subseteq [0, \infty)$.

Next, recall the heterogeneity parameter from (29):

$$\upsilon^2 := \sup_{w \in \mathcal{W}} \frac{1}{N} \sum_{i=1}^{N} \|\nabla F_i(w) - \nabla F(w)\|^2.$$

Lastly, for given parameters, denote

$$\xi_i := \left( \frac{1}{n_i \epsilon_i} \right)^2 \ln(2.5R/\delta_i) \ln(2/\delta_i)$$

for $i \in [N]$,

$$\xi_{\max} = \max(\xi_1, \cdots \xi_N),$$

and

$$\Xi := \sqrt{\mathbb{E}_{M_1} \left( \frac{1}{M_1} \sum_{i=1}^{M_1} \xi_{(i)} \right)^2}.$$

In the case of balanced data and same parameters across silos, we have $\xi_i = \xi = \Xi$ for all $i$. In the general case, we have $\xi_{\min} \leqslant \Xi \leqslant \xi_{\max}$.

## H.2  Pseudocode for Noisy ISRL-DP MB-SGD in the Unbalanced Case

The generalized version of Noisy ISRL-DP MB-SGD is described in Algorithm 7.

---
**Algorithm 7** Noisy ISRL-DP MB-SGD
---
1: **Input:** $N, d, R \in \mathbb{N}$, $\{\sigma_i\}_{i \in [N]} \subset [0, \infty)$, $X_i \in \mathcal{X}_i^{n_i}$ for $i \in [N]$, loss function $f(w, x)$, $\{K_i\}_{i=1}^{N} \subset \mathbb{N}$, $\{\eta_r\}_{r \in [R]}$ and $\{\gamma_r\}_{r \in [R]}$.
2: Initialize $w_0 \in \mathcal{W}$.
3: **for** $r \in \{0, 1, \cdots, R-1\}$ **do**
4:     **for** $i \in S_r$ **in parallel do**
5:         Server sends global model $w_r$ to silo $i$.
6:         Silo $i$ draws $K_i$ samples $x_{i,j}^r$ uniformly from $X_i$ (for $j \in [K_i]$) and noise $u_i \sim \mathcal{N}(0, \sigma_i^2 \mathbf{I}_d)$.
7:         Silo $i$ computes $\tilde{g}_r^i := \frac{1}{K_i} \sum_{j=1}^{K_i} \nabla f(w_r, x_{i,j}^r) + u_i$ and sends to server.
8:     **end for**
9:     Server aggregates $\tilde{g}_r := \frac{1}{M_r} \sum_{i \in S_r} \tilde{g}_r^i$.
10:    Server updates $w_{r+1} := \Pi_{\mathcal{W}}[w_r - \eta_r \tilde{g}_r]$.
11: **end for**
12: **Output:** $\hat{w}_R = \frac{1}{\Gamma_R} \sum_{r=0}^{R-1} \gamma_r w_r$, where $\Gamma_R := \sum_{r=0}^{R-1} \gamma_r$.
---

## H.3  General Unbalanced Version of Theorem 2.1

We first state the formal version of Theorem 2.1 for arbitrary $n_i, \epsilon_i, \delta_i$, using the notation defined in Appendix H.1.

**Theorem H.1** (Generalized Version of Theorem 2.1). *Let $\epsilon_i \leqslant 2\ln(2/\delta_i)$, $\delta_i \in (0, 1)$. Then, Algorithm 2 is $\{(\epsilon_i, \delta_i)\}_{i=1}^N$-ISRL-DP if $\sigma_i^2 = \frac{256 L^2 R \ln(\frac{2.5R}{\delta_i})\ln(2/\delta_i)}{n_i^2 \epsilon_i^2}$ and $K_i \geqslant \frac{\epsilon_i n}{4\sqrt{2R\ln(2/\delta_i)}}$. Moreover, with notation as in Appendix H.1, there are choices of algorithmic parameters such that:*

*1. If $f(\cdot, x)$ is convex, then*

$$\mathbb{E}F(\widehat{w}_R) - F^* \lesssim LD\left(\frac{1}{\sqrt{n_{\min}M}} + \sqrt{d\min\left\{\frac{\Xi}{M'}, \frac{\xi_{\max}}{M}\right\}}\right).$$

*2. If $f(\cdot, x)$ is $\mu$-strongly convex, then*

$$\mathbb{E}F(\widehat{w}_R) - F^* = \widetilde{\mathcal{O}}\left(\frac{L^2}{\mu}\left(\frac{1}{Mn_{\min}} + d\min\left\{\frac{\Xi}{M'}, \frac{\xi_{\max}}{M}\right\}\right)\right). \tag{41}$$

**Remark H.1.** *Note that $1/M' \geqslant 1/M$ by the Cauchy-Schwartz inequality. Both of the upper bounds in Theorem H.1 involve minima of the terms $\Xi/M'$ and $\xi_{\max}/M$, which trade off the unbalancedness of silo data and privacy needs with the variance of $1/M_r$. In particular, if the variance of $1/M_r$ is small enough that $\frac{\Xi}{M'} \leqslant \frac{\xi_{\max}}{M}$, then the excess risk bounds in Theorem H.1 depend on averages of the parameters across silos, rather than maximums. In FL problems with unbalanced data and disparate privacy needs across a large number of silos, the difference between "average" and "max" can be substantial. On the other hand, if data is balanced and privacy needs are the same across silos, then $\xi_i = \xi_{\max} = \Xi = \ln(2.5R/\delta_0)\ln(2/\delta_0)/n^2\epsilon_0^2$ for all $i$ and $\frac{\Xi}{M'} \geqslant \frac{\xi_{\max}}{M}$, so we recover Theorem 2.1, with dependence only on the mean $1/M$ of $1/M_r$ and not the square root of the second moment $1/M'$.*

To prove Theorem H.1, we need the following empirical loss bound for Algorithm 2, which generalizes Lemma D.6 to the unbalanced setting:

**Lemma H.1.** *Let $f : \mathcal{W} \times \mathcal{X} \to \mathbb{R}$ be $\mu$-strongly convex (with $\mu = 0$ for convex case), L-Lipschitz, and $\beta$-smooth in $w$ for all $x \in \mathcal{X}$, where $\mathcal{W}$ is a closed convex set in $\mathbb{R}^d$ s.t. $\|w - w'\| \leqslant D$ for all $w, w' \in \mathcal{W}$. Let $\mathbf{X} \in \mathbb{X}$. Then Algorithm 2 with $\sigma_i^2 = \frac{256 L^2 R \ln(\frac{2.5R}{\delta_i})\ln(2/\delta_i)}{n_i^2 \epsilon_i^2}$ attains the following empirical loss bounds as a function of step size and the number of rounds:*
*1. (Convex) For any $\eta \leqslant 1/\beta$ and $R \in \mathbb{N}$, $\gamma_r := 1/R$, we have*

$$\mathbb{E}\widehat{F}_{\mathbf{X}}(\widehat{w}_R) - \widehat{F}_{\mathbf{X}}^* \leqslant \frac{D^2}{\eta R} + \frac{\eta}{2}\left(d\min\left\{\frac{\Sigma^2}{M'}, \frac{\sigma_{\max}^2}{M}\right\} + L^2\right)$$

*2. (Strongly Convex) There exists a constant stepsize $\eta_r = \eta \leqslant 1/\beta$ such that if $R \geqslant 2\kappa\ln\left(\frac{\beta\mu D^2}{L^2 d\min\left(\Xi/M', \frac{\xi_{\max}}{M}\right)}\right)$, then*

$$\mathbb{E}\widehat{F}_{\mathbf{X}}(\widehat{w}_R) - \widehat{F}_{\mathbf{X}}^* = \widetilde{\mathcal{O}}\left(\frac{L^2}{\mu}\left(\frac{1}{R} + d\min\left\{\frac{\Xi}{M'}, \frac{\xi_{\max}}{M}\right\}\right)\right). \tag{42}$$

*Proof.* By the proof of Lemma D.6, we have:

$$\mathbb{E}\left[\|w_{r+1} - w^*\|^2 \Big| M_r\right] \leqslant (1 - \mu\eta_r)\mathbb{E}\left[\|w_r - w^*\|^2 \Big| M_r\right] - 2\eta_r \mathbb{E}[\widehat{F}_{\mathbf{X}}(w_r) - \widehat{F}_{\mathbf{X}}^*|M_r]$$
$$+ \eta_r^2 \mathbb{E}\left[\left\|\bar{u}_r + \frac{1}{M_r}\sum_{i \in S_r}\frac{1}{K_i}\sum_{j=1}^{K_i}\nabla f(w_r, x_{i,j}^r)\right\|^2 \Big| M_r\right]$$

which implies

$$\mathbb{E}\left[\|w_{r+1} - w^*\|^2\right] \leqslant (1 - \mu\eta_r)\mathbb{E}\left[\|w_r - w^*\|^2\right] - 2\eta_r \mathbb{E}[\widehat{F}_{\mathbf{X}}(w_r) - \widehat{F}_{\mathbf{X}}^*] + \eta_r^2 d\min\left\{\frac{\Sigma^2}{M'}, \frac{\sigma_{\max}^2}{M}\right\} + L^2, \tag{43}$$

since $\mathbb{E}[\|\bar{u}_r\|^2 | M_r] = \frac{1}{M_r^2} \sum_{i \in S_r}$ and hence

$$\mathbb{E}[\|\bar{u}_r\|^2] \leqslant d\mathbb{E}\left[\frac{\overline{\sigma_{M_r}^2}}{M_r}\right] \leqslant d\min\left(\frac{\sigma_{\max}^2}{M}, \frac{\Sigma^2}{M'}\right),$$

using linearity of expectation for the first term in the minimum and Cauchy-Schwartz inequality for the second term in the minimum. Now we consider the convex ($\mu = 0$) and strongly convex ($\mu > 0$) cases separately.

**Convex ($\mu = 0$) case:** Re-arranging (43), we get

$$\mathbb{E}[\widehat{F}_{\mathbf{X}}(w_r) - \widehat{F}_{\mathbf{X}}^*] \leqslant \frac{1}{2\eta_r}\left(\mathbb{E}[\|w_r - w^*\|^2 - \|w_{r+1} - w^*\|^2]\right) + \frac{\eta_r}{2}\left(d\min\left\{\frac{\Sigma^2}{M'}, \frac{\sigma_{\max}^2}{M}\right\} + L^2\right).$$

Then for $\eta_r = \eta$, the average iterate $\bar{w}_R$ satisfies:

$$\begin{aligned}
\mathbb{E}[\widehat{F}_{\mathbf{X}}(\bar{w}_R) - \widehat{F}_{\mathbf{X}}^*] &\leqslant \frac{1}{R}\sum_{r=0}^{R-1}\mathbb{E}[\widehat{F}_{\mathbf{X}}(w_r) - \widehat{F}_{\mathbf{X}}^*] \\
&\leqslant \frac{1}{R}\sum_{r=0}^{R-1}\frac{1}{2\eta}(\mathbb{E}[\|w_r - w^*\|^2 - \|w_{r+1} - w^*\|]) \\
&\quad + \frac{\eta}{2}\left(d\min\left\{\frac{\Sigma^2}{M'}, \frac{\sigma_{\max}^2}{M}\right\} + L^2\right) \\
&\leqslant \frac{\|w_0 - w^*\|^2}{\eta R} + \frac{\eta}{2}\left(d\min\left\{\frac{\Sigma^2}{M'}, \frac{\sigma_{\max}^2}{M}\right\} + L^2\right),
\end{aligned}$$

which proves part 1 of the lemma.

**Strongly convex ($\mu > 0$) case:** Note that (43) satisfies the conditions for Lemma D.5, with sequences

$$r_t = \mathbb{E}\|w_t - w^*\|^2, \, s_t = \mathbb{E}[\widehat{F}_{\mathbf{X}}(w_t) - \widehat{F}_{\mathbf{X}}^*]$$

and parameters

$$a = \mu, \, b = 2, \, c = d\min\left\{\frac{\Sigma^2}{M'}, \frac{\sigma_{\max}^2}{M}\right\} + L^2, \, g = 2\beta, \, T = R.$$

Then Lemma D.5 and Jensen's inequality imply

$$\mathbb{E}\widehat{F}_{\mathbf{X}}(\widehat{w}_R) - \widehat{F}_{\mathbf{X}}^* = \tilde{\mathcal{O}}\left(\beta D^2 \exp\left(\frac{-R}{2\kappa}\right) + \frac{L^2}{\mu}\left(\frac{1}{R} + d\min\left\{\frac{\Xi}{M'}, \frac{\xi_{\max}}{M}\right\}\right)\right),$$

where $\kappa = \beta/\mu$. Finally, plugging in $R$ completes the proof. □

We are prepared to prove Theorem H.1.

*Proof of Theorem H.1.* **Privacy:** The proof follows exactly as in the balanced case, since $\sigma_i^2$ is now calibrated to $(\epsilon_i, \delta_i)$ for all $i \in [N]$.

**Excess loss:** We shall prove the results for the case when $f(\cdot, x)$ is $\beta$-smooth. The non-smooth case follows by Nesterov smoothing, as in the proof of Theorem 2.1.
**1. Convex case:** By Lemma D.2 (and its proof), Lemma D.1, and Lemma H.1, we have:

$$\begin{aligned}
\mathbb{E}F(\widehat{w}_R) - F^* &\leqslant \alpha + \mathbb{E}\widehat{F}_{\mathbf{X}}(\widehat{w}_R) - \widehat{F}_{\mathbf{X}}^* \\
&\leqslant \frac{2L^2 R\eta}{n_{\min}M} + \frac{D^2}{\eta R} + \frac{\eta}{2}\left(d\min\left\{\frac{\Sigma^2}{M'}, \frac{\sigma_{\max}^2}{M}\right\} + L^2\right) \\
&\leqslant 2\eta L^2\left(\frac{R}{n_{\min}M} + dR\min\left\{\frac{\Xi}{M'}, \frac{\xi_{\max}}{M}\right\} + 1\right) + \frac{D^2}{\eta R},
\end{aligned}$$

for any $\eta \leqslant \frac{1}{\beta}$. Choosing $\eta = \min\left(1/\beta, \frac{D}{L\sqrt{R}}\min\left(\frac{\sqrt{n_{\min}M}}{\sqrt{R}}, 1, \sqrt{\frac{1}{dR}\max\left\{\frac{M'}{\Xi}, \frac{M}{\xi_{\max}}\right\}}\right)\right)$ yields

$$\mathbb{E}F(\widehat{w}_R) - F^* \lesssim \frac{\beta D^2}{R} + LD\left(\frac{1}{\sqrt{n_{\min}M}} + \frac{1}{\sqrt{R}} + \sqrt{d\min\left\{\frac{\Xi}{M'}, \frac{\xi_{\max}}{M}\right\}}\right).$$

Choosing $R \geqslant \frac{\beta D}{L}\min\left(\sqrt{Mn_{\min}}, \sqrt{d\min\left\{\frac{\Xi}{M'}, \frac{\xi_{\max}}{M}\right\}}\right) + \min\left(n_{\min}M, \frac{1}{d}\max\left\{\frac{M'}{\Xi}, \frac{M}{\xi_{\max}}\right\}\right)$ completes the proof of the convex case.

**2. $\mu$-strongly convex case:** By Lemma D.2 (and its proof), Lemma D.1, and Lemma H.1, we have:

$$\mathbb{E}F(\widehat{w}_R) - F^* \leqslant \alpha + \mathbb{E}\widehat{F}_{\mathbf{X}}(\widehat{w}_R) - \widehat{F}_{\mathbf{X}}^*$$
$$\leqslant \frac{4L^2}{\mu(Mn_{\min}-1)} + \tilde{\mathcal{O}}\left(\frac{L^2}{\mu}\left(\frac{1}{R} + d\min\left\{\frac{\Xi}{M'}, \frac{\xi_{\max}}{M}\right\}\right)\right)$$

for the $\eta \leqslant 1/\beta$ prescribed in the proof of Lemma H.1 and any $R \geqslant 2\kappa\ln\left(\frac{\beta\mu D^2}{L^2 d\min(\Xi/M', \frac{\xi_{\max}}{M})}\right)$.

Thus, choosing $R = 2\kappa\ln\left(\frac{\beta\mu D^2}{L^2 d\min(\Xi/M', \frac{\xi_{\max}}{M})}\right) + Mn_{\min}$ completes the proof. $\square$

**Remark H.2.** *Generalized versions of the other upper bounds in this paper can also be easily derived with the techniques used above. The key takeaways are: a) the excess empirical risk (and the private term in the SCO bounds) involve a minimum of two terms that trade off the degree of unbalancedness with the variance of $1/M_r$. In particular, if the variance of $1/M_r$ is sufficiently small (e.g. if $M_r = M$, which is what most existing works on FL assume), then the refined excess risk bounds depend on averages of the parameters across silos, rather than worst-case maximums. b) the generalization error scales with $\min_{i\in[N]} n_i$.*

## I  NUMERICAL EXPERIMENTS: DETAILS AND ADDITIONAL RESULTS

In some plots in this section, we include a parameter describing the heterogeneity of the FL problem:

$$\upsilon_*^2 := \frac{1}{N}\sum_{i=1}^{N}\|\nabla F_i(w^*)\|^2,$$

which has appeared in (Khaled et al., 2019; Koloskova et al., 2020; Karimireddy et al., 2020; Woodworth et al., 2020b). If the data is homogeneous, then all $F_i$ share the same minimizers, so $\upsilon_*^2 = 0$, but the converse is false.

ISRL-DP Local SGD runs as follows: in round $r$, each silo $i \in S_r$ receives the global model $w_r$ and takes $K$ steps of noisy SGD (with one sample per step) with their local data: $w_r^{i,0} = w_r$, $w_r^{i,t} = w_r^{i,t-1} - \eta(\nabla f(w_r^{i,t-1}, x_{i,t}^r) + u_i^t)$ for $t \in [K]$, where $x_{i,t}^r$ is drawn uniformly at random from $X_i$ and $u_i^t \sim \mathcal{N}(0, \sigma^2 \mathbf{I}_d)$ for $\sigma^2 = \frac{8L^2 RK\log(1/\delta)}{\epsilon_0^2 n^2}$. Then silo $i$ sends its $K$-th iterate, $w_r^{i,K}$ to the server; the server averages the iterates across all silos and updates the global model to $w_{r+1} = \frac{1}{M_r}\sum_{i\in S_r} w_r^{i,K}$.

### I.1  LOGISTIC REGRESSION WITH MNIST

The data set can be downloaded from `http://yann.lecun.com/exdb/mnist`. Our code does this for you automatically.

**Experimental setup:** To divide the data into $N = 25$ silos and for preprocessing, we borrow code from (Woodworth et al., 2020b), which can be downloaded from: `https://papers.nips.cc/paper/2020/hash/45713f6ff2041d3fdfae927b82488db8-Abstract.html`. It is available under a Creative Commons Attribution-Share Alike 3.0 license. There are $n_i = n = 8673$ training and $2168$ test examples per silo; to expedite training, we use only $1/7$

of the MNIST samples ($n = 1238$ training examples per silo). We fix $\delta_i = \delta = 1/n^2$ and test $\epsilon \in \{0.75, 1.5, 3, 6, 12, 18\}$. The maximum $\upsilon_*^2$ is about $0.17$ for this problem (corresponding to each silo having disjoint local data sets/pairs of digits).

**Preprocessing:** We used PCA to reduce the dimensionality to $d = 50$. We used an $80/20$ train/test split for all silos. To improve numerical stability, we clipped the input $\langle w, x \rangle$ (i.e. projected it onto $[-15, 15]$) before feeding into logistic loss.

**Hyperparameter tuning:** For each algorithm and each setting of $\epsilon, R, K, \upsilon_*^2$, we swept through a range of constant stepsizes and ran 3 trials to find the (approximately) optimal stepsize for that particular algorithm and experiment. We then used the corresponding $w_R$ (averaged over the 3 runs) to compute test error. For (ISRL-DP) MB-SGD, the stepsize grid consisted of 10 evenly spaced points between $e^{-6}$ and 1. For (ISRL-DP) Local SGD, the stepsizes were between $e^{-8}$ and $e^{-1}$. We repeated this entire process 20 times for fresh train/test splits of the data and reported the average test error in our plots.

**Choice of $\sigma^2$ and $K$:** We used smaller noise (compared to the theoretical portion of the paper) to get better utility (at the cost of larger $K$/larger computational cost, which is needed for privacy): $\sigma^2 = \frac{8L^2 \ln(1/\delta)R}{n^2 \epsilon^2}$, which provides ISRL-DP by Theorem 1 of (Abadi et al., 2016) if $K = \frac{n\sqrt{\epsilon}}{2\sqrt{R}}$ (c.f. Theorem 3.1 in (Bassily et al., 2019)). Here $L = 2 \max_{x \in X} \|x\|$ is an upper bound on the Lipschitz parameter of the logistic loss and was computed directly from the training data.

To estimate $\upsilon_*^2$, we followed the procedure used in (Woodworth et al., 2020b), using Newton's method to compute $w^*$ and then averaging $\|\nabla \widehat{F}_i(w^*)\|^2$ over all $i \in [N]$.

**Additional experimental result :** In Fig. 7, we show an additional experiment with $M = 18$ available to communicate in each round. The results are qualitatively similar to the results presented in the main body for MNIST with $M = 25, 12$.

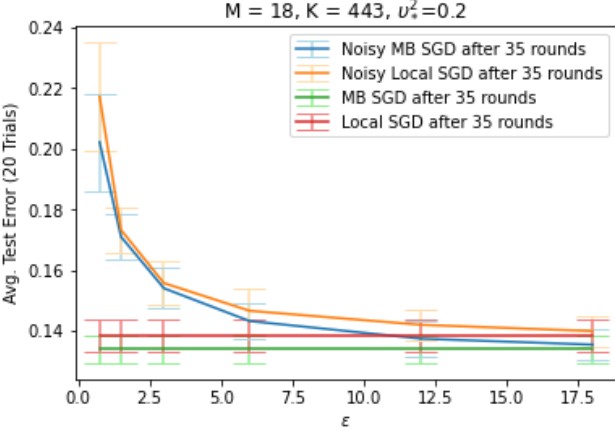

Figure 7: Test error vs. $\epsilon$ for binary logistic regression on MNIST. $\delta = 1/n^2$. We show 90% error bars over the 20 trials (train/test splits).

**Limitations of Experiments:** It is important to note that pre-processing and hyperparameter tuning (and estimation of $L$) were not done in a ISRL-DP manner, since we did not want to detract focus from evaluation of ISRL-DP FL algorithms.[16] As a consequence, the overall privacy loss for the entire experimental process is higher than the $\epsilon$ indicated in the plots, which solely reflects the privacy loss from running the FL algorithms with fixed hyperparameters and (pre-processed) data. Similar remarks apply for the linear regression experiments (see Appendix I.2).

---

[16]See (Abadi et al., 2016; Liu & Talwar, 2019; Papernot & Steinke, 2021) and the references therein for discussion of DP PCA and DP hyperparameter tuning.

## I.2 LINEAR REGRESSION WITH HEALTH INSURANCE DATA

**Data set:** The data (`https://www.kaggle.com/mirichoi0218/insurance`), which is available under an Open Database license, consists of $\widetilde{N} = 1338$ observations. The target variable $y$ is medical charges. There are $d - 1 = 6$ features: age, sex, BMI, number of children, smoker, and geographic region.

**Experimental setup:** For a given $N$, we grouped data into $N$ (almost balanced) silos by sorting $y$ in ascending order and then dividing into $N$ groups, the first $N - 1$ of size $\lceil 1338/N \rceil$ and the remaining points in the last silo. For each $N$, we ran experiments with $R = 35$. We ran 20 trials, each with a fresh random train/test (80/20) split. We tested $\epsilon \in \{.125, .25, .5, 1, 2, 3\}$ and fixed $\delta_i = 1/n_i^2$ for all experiments.

To estimate $\upsilon_*^2$, we followed the procedure used in (Woodworth et al., 2020b), using Newton's method to compute $w^*$ and then averaging $\|\nabla \widehat{F}_i(w^*)\|^2$ over all $i \in [N]$.

**Preprocessing:** We first numerically encoded the categorical variables and then standardized the numerical features *age* and *BMI* to have zero mean and unit variance.

**Gradient clipping:** In the absence of a reasonable a priori bound on the Lipschitz parameter of the squared loss (as is typical for unconstrained linear regression problems with potentially unbounded data), we incorporated gradient clipping (Abadi et al., 2016) into the algorithms. We then calibrated the noise to the clip threshold $L$ to ensure LDP. For fairness of comparison, we also allowed for clipping for the non-private algorithms (if it helped their performance).

**Hyperparameter tuning:** For each trial and each algorithm, we swept through a log-scale grid of 10 stepsizes and 5 clip thresholds 3 times, selected the parameter $w$ that minimized average (over 3 repetitions) training error (among all $10 \times 5 = 50$), and computed the corresponding average test error. The stepsize grids we used ranged from $e^{-8}$ and $e^1$ for (ISRL-DP) MB-SGD and from $e^{-10}$ to 1 for (ISRL-DP) Local SGD. The excess risk (train and test) we computed was for the normalized objective function $F(w, X, Y) = \|Y - wX\|^2/2N_0$ where $N_0 \in \{1070, 268\}$ (1070 for train, 268 for test) and $X$ is $N_0 \times d$ with $d = 7$ (including a column of all 1s) and $Y \in \mathbb{R}^{N_0}$. The clip threshold grids were $\{100, 10^4, 10^6, 10^8, 10^{32}\}$, with the last element corresponding to effectively no clipping.

**Choice of $\sigma^2$ and $K$:** We used the same $\sigma^2$ and $K = \frac{n\sqrt{\epsilon}}{2\sqrt{R}}$ as in the logistic regression experiments described in Appendix I.1. However, here $L$ is the clip threshold instead of the Lipschitz parameter.

**Relative Root Mean Square Error (RMSE):** We scale our reported errors (in the plots) to make them more interpretable. We define the Relative (test) RMSE of an algorithm to be $\sqrt{MSE/NMSE} = \sqrt{\frac{\sum_{i=1}^{N_{\text{test}}} (y_i - \hat{y}_i)^2}{\sum_{i=1}^{N_{\text{test}}} (y_i - \bar{y}_{\text{train}})^2}}$, where $NMSE$ ("Naiive Mean Square Error") is the (test) MSE incurred by the non-private naiive predictor that predicts $y_i$ to be $\bar{y}_{\text{train}}$, the average of the training labels, for all test data $i \in [N_{\text{test}}]$. Here $\hat{y}_i$ is the predicted label of the algorithm. Relative RMSE can be thought of as the Coefficient of Variation or Relative Standard Deviation of the predictions of the algorithm. Note that even though the naiive predictor is completely non-private and allowed to access the entire training data even when $M < N$ (violating the decentralized data principle of FL), the ISRL-DP FL algorithms still outperform this predictor for most values of $\epsilon$ (except for some experiments when $\epsilon \approx 0$), as evidenced by values of Relative RMSE being below 1. For $\epsilon \approx 1$, ISRL-DP MB-SGD tends to outperform the naiive predictor by $30 - 40\%$.

**Additional experimental results:** In Fig. 8 and Fig. 9, we present results for experiments with additional settings of $N$ and $M$. We observe qualitatively similar behavior as in the plots presented in the main body of the paper. In particular, ISRL-DP MB-SGD continues to outperform ISRL-DP Local SGD in most tested setups/privacy levels (especially the high-privacy regime). On the other hand, for some settings of $M, N, K$, we observe that ISRL-DP Local SGD outperforms ISRL-DP MB-SGD as $\epsilon \to 3$ (e.g. Fig. 9, and $N = M = 3$). In general, we see that the utility impact of ISRL-DP is relatively insignificant for this problem when $\epsilon \approx 3$.

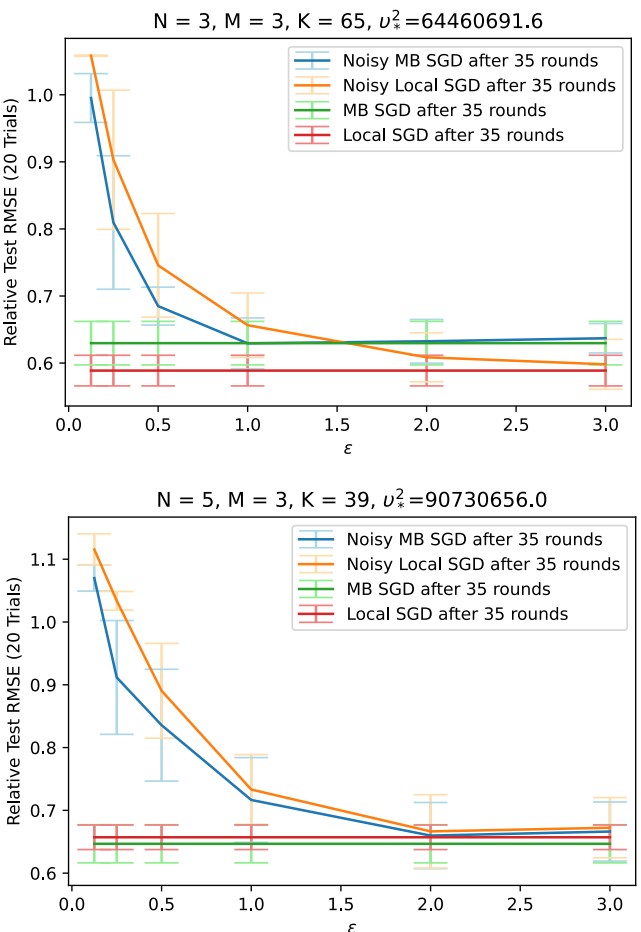

Figure 8: Test error vs. $\epsilon$ for linear regression on heterogeneous health insurance data. $\delta = 1/n^2$. 90% error bars are shown.

### I.3 SOFTMAX REGRESSION WITH OBESITY DATASET

The data set can be freely downloaded from `https://archive.ics.uci.edu/ml/datasets/Estimation+of+obesity+levels+based+on+eating+habits+and+physical+condition+`. The data contains 17 attributes (such as age, gender, physical activity, eating habits, etc.) and 2111 records.

**Experimental setup:** We divide the data into $N = 7$ heterogeneous silos based on the value of the target variable, obesity level, which is categorical and takes 7 values: Insufficient Weight, Normal Weight, Overweight Level I, Overweight Level II, Obesity Type I, Obesity Type II and Obesity Type III. We fix $\delta_i = \delta = 1/n^2$ and test $\epsilon \in \{0.5, 1, 3, 6, 9\}$. We ran three trials with a new train/test split in each trial and reported the average test error.

**Preprocessing:** We numerically encode the categorical variables and standardize the continuous numerical features to have mean zero and unit variance. We used an $80/20$ train/test split. We discarded a small number of (randomly selected) training samples from some silos in order to obtain balanced silo sets, to ease implementation of the noisy methods.

**Hyperparameter tuning:** For each algorithm and each setting of $\epsilon, R, K$, we swept through a range of constant stepsizes to find the (approximately) optimal stepsize for that particular algorithm and experiment. We then used the corresponding $w_R$ to compute test error for that trial. For (ISRL-DP) MB-SGD, the stepsize grid consisted of 8 evenly spaced points between $e^{-7}$ and $e^{-1}$ For (ISRL-DP)

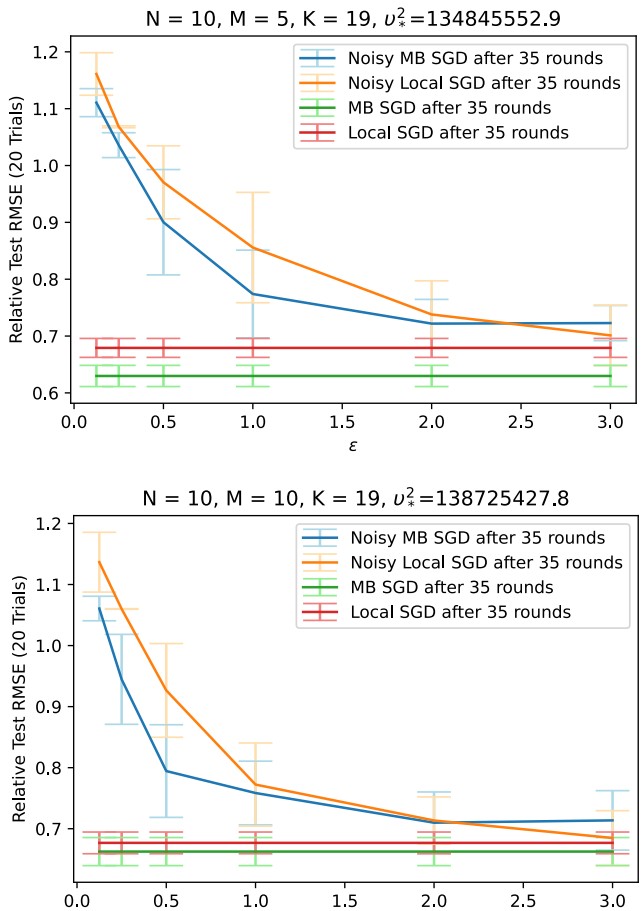

Figure 9: Test error vs. $\epsilon$ for linear regression on heterogeneous health insurance data. $\delta = 1/n^2$. 90% error bars are shown.

Local SGD, we started with the same stepsize grid, but sometimes required additional tuning with smaller stepsizes (especially for small $\epsilon$) to find the optimal one.

**Choice of $\sigma^2$ and $K$:** We used smaller noise (compared to the theoretical portion of the paper) to get better utility (at the cost of larger $K$/larger computational cost, which is needed for privacy): $\sigma^2 = \frac{8L^2 \ln(1/\delta)R}{n^2 \epsilon^2}$ for ISRL-DP MB-SGD, which provides ISRL-DP by Theorem 1 of (Abadi et al., 2016) if $K = \frac{n\sqrt{\epsilon}}{2\sqrt{R}}$ (c.f. Theorem 3.1 in (Bassily et al., 2019)). For ISRL-DP Local SGD, one needs $\sigma^2 = \frac{8L^2 \ln(1/\delta)RK}{n^2 \epsilon^2}$ since the sensitivity and number of gradient evaluations are both larger by a factor of $K$. Here $L = 2\max_{x \in X} \|x\| = 20$ is an upper bound on the Lipschitz parameter of the softmax loss and was estimated directly from the pre-processed training data.

## J   LIMITATIONS AND SOCIETAL IMPACTS

### J.1   LIMITATIONS

Our results rely on certain assumptions (e.g convex, Lipschitz loss), which may be violated in certain practical applications. Moreover, our theoretical results require an *a priori* bound on the Lipschitz parameter (for noise calibration). While such a bound is fairly easy to obtain for models such as logistic regression with data that is (known to be) bounded (e.g. our MNIST experiments), it is unrealistic for models such as unconstrained linear regression with potentially unbounded data (e.g. our medical cost data experiments). In practice, in such situations, gradient clipping can be incorporated into our algorithms; we have shown empirically (medical cost data experiments) that MB-SGD still performs well with clipping. However, we did not obtain our theoretical results with gradient clipping. An interesting direction for future work would be to determine optimal rates for ISRL-DP FL without Lipschitzness and/or without convexity. Further, as we explained in Appendix I, pre-processing and hyperparameter tuning (and estimation of $L$) were not done in a ISRL-DP manner in our numerical experiments, since we did not want to detract focus from evaluation of ISRL-DP FL algorithms. As a consequence, the overall privacy loss for the entire experimental process is higher than the $\epsilon$ indicated in the plots, which solely reflects the privacy loss from running the FL algorithms with fixed hyperparameters and (pre-processed) data.

### J.2   SOCIETAL IMPACTS

Our work provides algorithms for protecting the privacy of users during federated learning. Privacy is usually thought of favorably: for example, the right to privacy is an element of various legal systems. However, it is conceivable that our algorithms could be misused by corporations and governments to justify malicious practices, such as collecting personal data without users' permission. Also, the (necessarily) lower accuracy from privately trained models could have negative consequences: e.g. if a ISRL-DP model is used to predict the effects of climate change and the model gives less accurate and more optimistic results, then a government might use this as justification to improperly eliminate environmental protections. Nevertheless, we believe the dissemination of privacy-preserving machine learning algorithms and greater knowledge about these algorithms provides a net benefit to society.

