# OpenReview forum: "Private Federated Learning Without a Trusted Server: Optimal Algorithms for Convex Losses"
_ICLR.cc/2023/Conference — ICLR 2023 poster_

### Official Review · Reviewer_beiC · 2022-10-16

**Confidence:** 3
**Correctness:** 4
**Technical Novelty And Significance:** 3
**Empirical Novelty And Significance:** 2
**Recommendation:** 6

**Clarity, Quality, Novelty And Reproducibility:**

Clarity: This paper is clear. The theoretical results are clearly stated with description of assumptions and risk bounds. Experimental details are also provided.

Quality: The theory is solid and covers several different settings, and is corroborated with extensive experiments. I did not check all the proof details but they look credible.

Novelty: There is technical novelty because this paper addresses unique technical challenges during the proof of the lower bound and novel analysis framework is used. The authors discuss this point clearly in the paper. So I think this paper has novelty. Furthermore, although the paper is restricted to the convex loss setting, this is not a big problem in my opinion since the paper provides an analysis of ISRL-DP which is lack from the literature.


Reproducibility:
I did not check all the proof details but they do look credible to me. As for the experimental results, details are included in the appendix. Based on these two points, I believe this paper has good reproducibility.


**Strength And Weaknesses:**

Strength:

1. Meaningful problem setting: the ISRL-DP is a reasonable notion of DP with practical applications. This paper also explains this point clearly by listing scenarios under which ISRL-DP is appropriate (i.e. when a client only trust its own silo). Therefore, establishing theoretical guarantees for ISRL-DP is a meaningful problem.

2. Solid theory: under the general setting of convex loss function, this paper considers a various of different settings including a (not complete) combination of the following: homogeneous versus heterogeneous agents; convex and strongly convex loss functions; smooth and general loss functions; existence of trusted shuffler or not. This paper gives algorithms for these settings with convergence rate guarantees and ISRL-DP.

The solid theory is accompanied by rather extensive experimental results. The performance of the proposed optimization algorithm is comparable to that of FedAvg which has no DP guarantee.

---
Weakness:

There is no major technical weakness of this paper in my opinion. Still, I think the following is worth mentioning.

1. The homogeneous setting is over-simplified and not very realistic. Basically, since $f$ is the same for all agents, when the distribution $\mathcal{D}_i$ is the same for all $i$, it means every agent is facing the same learning problem. This seems not very practical.

2. The main text of this paper is not easy to read. The reason is that there are quite a few theoretical results with different assumptions. So it is a little hard to follow.

---
Minor comments/suggestions:

1. It is confusing when the authors mentioned in the bottom of page 3 that in section 4 they consider a setting where $F$ is an empirical loss. Since from equation 1 and 2 the definition of $F$ involves an expectation over data distribution, the ‘empirical loss’ confused me when I read for the first time.

2. I suggest making a table of results (i.e. convergence rate, etc.) and assumptions (i.e. smooth or not; convex or strongly convex, etc.). This table can help readers compare different results and follow the settings.


**Summary Of The Paper:**

This paper considers a realistic DP notion called Inter-silo Record-level DP (ISRL-DP) which can address the shortcoming of central DP and LDP. An algorithm satisfying ISRL-DP requires that the output of this algorithm cannot be distinguished between two adjacent dataset of any agent, given all other agents’ dataset being fixed. The paper then gives a series of risk bounds for federated learning with ISRL-DP. The loss function is assumed to be convex in this paper.

---
First, under the homogeneous setting (i.e. the data distribution is the same across all agents), the paper shows noisy MiniBatch SGD is ISRL-DP. It then gives upper bounds of excess risk for convex and strongly convex loss function. It also gives lower bounds under the case of full participation (i.e. all agents participate in each round), which almost matches the upper bound. Furthermore, the lower bounds are quite general and can be reduced to existing lower bound for central-DP and local-DP.

Second, the paper considers a heterogeneous setting with full participation, and proposes an one-pass accelerated noisy MB-SGD algorithm with convergence guarantee under the convex and strongly convex loss function. Notably, the upper bound has a nearly optimal rate when the strongly convex loss function is well-conditioned, and is sub-optimal when the loss function is convex.

Third, this paper considers the empirical excess risk minimization, and proposes an algorithm that achieves optimal rate.

Fourth, this paper considers a case with a trusted shuffler that can permute the datasets from all agents. It proposes an algorithm that can achieve optimal rate with central-DP under homogeneous agents. Under heterogeneous agents, nearly optimal rate is achieved under the extra assumption that the loss is smooth.

Several experiments are conducted and the results corroborate the theory.


**Summary Of The Review:**

I recommend accept.

---
My reasons are the following:
The problem considered is meaningful and well-motivated. The ISRL-DP is an interesting topic and a theoretical analysis would be valuable to the community.
The theory looks solid and covers a variety of settings, most of which are practical. Although the homogeneous setting is not that interesting, I do not think this is a huge problem because it is only part of the paper, and even for the convex homogeneous case this paper seems to give (one of) the first analysis.
The proof is complete and looks credible. The proof (I read some of it) is well-written with plenty of explanations.
There are experimental results to support the theory.

---
There is something that needs improvement.
Presentation: The main text should be organized better to make results and assumptions easier to follow.

---

> ### Author Response · Authors · 2022-11-19
> **Response to Reviewer beiC**
>
> Thank you for appreciating our contributions and for your detailed feedback! We respond to your questions/comments below:
>
> >*The homogeneous setting is over-simplified and not very realistic*
>
> We agree that the heterogeneous setting considered in Section 3 is usually more realistic than the homogeneous setting of Section 2. However, we still feel that it is worth studying the homogeneous setting. First, on a theoretical level, in the homogeneous setting, we are able to characterize the optimal excess risk bounds (up to logarithms). This informs the fundamental privacy-accuracy tradeoff of ISRL-DP. For example, we prove that the cost of ISRL-DP is intermediate between the costs of CDP and LDP, affirming our intuition (due to the intermediate privacy notion). Secondly, on a practical level, some special FL problems can be well-approximated by homogeneous problems. For example, suppose the FL problem is to train a model that predicts household electricity usage, where the silos are households in the same neighborhood. Then, in certain cities, on any fixed day, it is likely that the electricity usage patterns of neighboring households follow similar distributions (e.g. since temperature is roughly the same outside of each house).
>
> >*The main text of this paper is not easy to read. The reason is that there are quite a few theoretical results with different assumptions…I suggest making a table of results (i.e. convergence rate, etc.) and assumptions (i.e. smooth or not; convex or strongly convex, etc.). This table can help readers compare different results and follow the settings.*
>
> Figure 3 summarizes our results and assumptions. Also, we revised page 2 of the paper to clarify the privacy notions and added simpler discussions. Please let us know if there is anything else we can do to improve readability.
>
> >*It is confusing when the authors mentioned in the bottom of page 3 that in section 4 they consider a setting where $F$ is an empirical loss. Since from equation 1 and 2 the definition of $F$ involves an expectation over data distribution, the ‘empirical loss’ confused me when I read for the first time.*
>
> Note that empirical risk minimization is a special case of the FL problem defined in Equations (1) and (2): by letting $\mathcal{D}^i$ be the empirical distribution on dataset $X_i$, we recover the case where $F = \widehat{F}_{\mathbf{X}}$ is an empirical loss. We clarified this in our revision (footnote 5).

---

> > ### Comment · Reviewer_beiC · 2022-11-22
> > **Thank you for your rebuttal.**
> >
> > Thank you for taking time to writing the response!
> >
> > I agree that the homogeneous case is simple but still has its own theoretical interest.
> >
> > I think the current paper is more readable given your revision. Appreciate the effort.
> >
> > I have no further question.

---

> > > ### Author Response · Authors · 2022-11-23
> > > **Thank you for your feedback**
> > >
> > > Thank you very much for your feedback and for helping us improve the quality of the manuscript. We are glad that you were satisfied with our response and found the revision to be more readable. Finally, given our response and revision, we kindly ask you to increase your score if you find it appropriate. Thank you again for your time and effort in giving us such constructive feedback!

---

### Official Review · Reviewer_SUpG · 2022-10-24

**Confidence:** 3
**Correctness:** 4
**Technical Novelty And Significance:** 3
**Empirical Novelty And Significance:** 3
**Recommendation:** 6

**Clarity, Quality, Novelty And Reproducibility:**

To the best of my knowledge, this work provides the first convergence results for convex optimization and ERM under ISRL-DP
The results, contributions, and techniques are clearly stated.
The authors provide code for their algorithms and experiments which helps with reproducing the results claimed in the paper.

**Strength And Weaknesses:**

Strengths
- This work provides the first convergence results for convex optimization and ERM under ISRL-DP to the best of my knowledge.
- Near-optimal rates are obtained for stochastic convex optimization and ERM for convex and strongly convex functions.
- The practical setting where client data distributions are heterogenous is considered and convergence results are provided.
- The authors provided empirical results for the proposed algorithms, and the performance is encouraging.

Weakness
- The lower bound proof uses an indirect approach using privacy amplification by shuffling. However, privacy amplification holds under certain conditions on $\epsilon_0, \delta_0$ and may lead to extra log factors, so tight lower bounds may not be obtained for all parameter regimes. What's the main difficulty in directly proving lower bounds for ISRL-DP?
- Presentation: the table summarizing the results (figure 3) can be made larger and more readable.

**Summary Of The Paper:**

This work studies Inter-silo record-level DP in cross-silo federated learning, where all communications to the server should satisfy item-level DP. The authors investigate convex optimization and design algorithms that achieve near-optimal performance for both convex and strongly convex cases. With amplification by shuffling, the error rate matches the optimal rate for central differential privacy. The results are supported by empirical evidence.

**Summary Of The Review:**

This work provides convergence rates for SCO and ERM with convex losses under ISRL-DP, verified by experiment results. A minor issue is that the lower bound technique is indirectly proved and may not lead to tight rates for all parameter regimes.

---

> ### Author Response · Authors · 2022-11-19
> **Response to Reviewer SUpG**
>
> Thank you for appreciating our contributions and for your great questions/feedback! We respond to your questions/comments below:
>
> >*The lower bound proof uses an indirect approach using privacy amplification by shuffling… What's the main difficulty in directly proving lower bounds for ISRL-DP?*
>
> We discuss this a bit on page 6. Essentially, lower bounds for ISRL-DP are more challenging to prove because the definition of the privacy notion is more complicated and nuanced than LDP, user-level DP, and CDP. Further, since no non-trivial ISRL-DP lower bounds were known prior to our work–-even for simple problems like mean estimation, we needed to come up with the first framework for proving lower bounds under ISRL-DP.
>
> LDP lower bounds are easier to prove due to the simplicity of the privacy notion, especially for sequentially interactive algorithms (see e.g. Duchi et al., FOCS 2013). The LDP proof techniques are heavily tailored to the simple characterization of sequentially interactive LDP algorithms (in terms of privacy of conditionally independent local randomizers), and do not appear to be applicable in the more complex ISRL-DP setting with fully interactive FL algorithms.  We attempted to prove the necessary extension of the information-theoretic tools in Duchi et al. (2013) to ISRL-DP FL, but it does not seem like such extension is possible: these tools hinge on conditional independence of local randomizers, which is not satisfied in the fully interactive ISRL-DP FL setting. Further, the CDP lower bound proof frameworks–often consisting of packing arguments and a reduction to mean estimation (e.g. Bassily et al., 2014)–are specific to centralized notions of DP (CDP and user-level DP).
>
>
> Given that ISRL-DP is an intermediate privacy notion between LDP and CDP and lower bounds for both of these notions are known, a natural approach is to try to relate ISRL-DP to one of these other privacy notions somehow. By proving a non-trivial extension of privacy amplification by shuffling from sequentially interactive/LDP to fully interactive/ISRL-DP, we were able to do this, and obtain nearly-tight lower bounds. Obtaining bounds that are exactly tight (no logarithmic slack) in all parameter regimes, and developing other ISRL-DP lower bound proof techniques could be interesting directions for future work.
>
> >*Presentation: the table summarizing the results (figure 3) can be made larger and more readable.*
>
> In the rebuttal revision, we have increased the size of the table. We really appreciate your detailed feedback in helping us improve the manuscript.

---

> > ### Author Response · Authors · 2022-12-06
> > **Have we addressed your questions/concerns?**
> >
> > Thank you again for reading our paper and providing such great feedback. We know it is a busy time of the year for everyone. We were wondering if we've addressed your concerns with our responses. Please let us know if any questions remain.
> >
> > Thank you,
> >
> > Authors

---

### Official Review · Reviewer_Gj6i · 2022-10-24

**Confidence:** 4
**Clarity, Quality, Novelty And Reproducibility:** The quality, clarity, and originality…
**Correctness:** 4
**Technical Novelty And Significance:** 3
**Empirical Novelty And Significance:** Not applicable
**Recommendation:** 6

**Strength And Weaknesses:**

Strength:

1. This paper fills some gaps in the field of (record-level) differentially private FL convergence analysis. In particular, the authors present the first tight convergence analysis of FL with record-level approximation DP.

2. This work provides the first convergence analysis when the shuffle model is incorporated in FL to achieve DP.

Weaknesses

1. The motivation for Inter-Silo Record-Level Differential Privacy (ISRL-DP) is not clear. If I understand correctly, ISRL-DP is identical to the setting where record-level DP is incorporated into local model training process, e.g., train local models by DP-SGD [1], and then submit local model parameters to the (untrusted) server. In other words, it treats each FL party as a database, and answer the query from the server (i.e., submit local models) in a differentially private manner. Therefore, It is unclear to me what really is the difference between ISRL-DP and record-level DP.

2. The proposed algorithm (Algorithm 1) seems impractical. The Accelerated Noisy MB-SGD algorithm samples batches without replacement. In this way, each individual record is only used exactly once for the gradient computation, and thus parallel composition can be applied to account the overall privacy loss. While this modification can help achieve a convergence rate that is close to the i.i.d. setting, I am afraid that this algorithm will perform poorly in practice, because each record is only used once during the entire FL training process. A more general and practical setting is that each record would be sampled multiple times (e.g., DP-SGD algorithm [1]).

3. For the analysis of the non-i.i.d. setting, I do not understand how this paper quantifies the degree of non-i.i.d. In previous work [2], the non-i.i.d. degree is explicitly characterized by introducing a parameter ($\Gamma$).

======== Reference ========

[1] M. Abadi, A. Chu, I. Goodfellow, H. B. McMahan, I. Mironov, K. Talwar, and L. Zhang, “Deep learning with differential privacy,” in CCS, 2016.

[2] X. Li, K. Huang, W. Yang, S. Wang, and Z. Zhang, “On the convergence of fedavg on non-iid data,” in ICLR, 2019.

**Summary Of The Paper:**

This paper analyzes the convergence of FL with record-level DP. It establishes nearly tight upper and lower bound for Noisy-Distributed-MB-SGD in the i.i.d. setting. For the non-i.i.d. setting, it proposes a new DP-FL algorithm (a modified version of Noisy-SGD), and proves the upper bound of its convergence rate. In particular, the upper bound of the proposed algorithm nearly matches the optimal i.i.d. bound when the loss function is strongly convex. This paper further analyzes the upper bound of the convergence rate when the shuffle model is applied into FL. Finally, it conducts numerical experiments to examine the theoretical results.

**Summary Of The Review:**

This paper presents some interesting theoretical results. In particular, it proves the tight upper and lower bounds for individual-level DP FL algorithms when silo data is i.i.d. On the other hand, the proposed non-i.i.d. algorithm is impractical, thus the empirical novelty and significance is somehow limited.

---

> ### Author Response · Authors · 2022-11-19
> **Response to Reviewer Gj6i**
>
> Thank you for appreciating our contributions and for your great questions/feedback! We respond to your questions/suggestions below:
>
> >*Difference between ISRL-DP and record-level DP, and motivation for ISRL-DP*
>
> Based on your comments, we believe that the source of confusion might simply be a matter of terminology: to distinguish inter-silo record-level DP from classical centralized record-level DP, **we refer to (classical, centralized) record-level DP as “central DP (CDP).”**
> We revised the paper (particularly the violet part on page 2) to further clarify the ISRL-DP notion. We have an example in Figure 1, and the difference between ISRL-DP and other DP notions is explained in the text and summarized in Figure 2. We believe this revision should clarify/answer your concern.  We briefly summarize the differences between ISRL-DP and central DP below:
>
> **Inter-silo record-level DP (ISRL-DP)** requires each individual silo to provide *record-level DP for their own local data.* Adjacent (distributed) databases are those which differ in $N$ samples: one sample in each silo. Thus, ISRL-DP protects the privacy of each individual person’s record in all $N$ silos, even when the server/other silos are adversarial eavesdroppers. Two motivations for ISRL-DP are: a) in cross-silo FL (e.g. silos are hospitals), each sample (e.g. health record) corresponds to a person (e.g. person), so a record-level DP guarantee is desired for each specific silo; and b) people (e.g. patients)/laws (e.g. HIPAA) may not want/allow silo’s local data to be shared with the server/other silos. You are right that one way to achieve ISRL-DP would be to train local models using DP-SGD and then submit private model updates to the server.
>
> By contrast, using classical **record-level DP**–which we refer to as **central DP (CDP)** in our paper–*would not adequately protect the privacy of every person’s data in cross-silo FL*. That is because:  a) the notion of adjacency in the definition of CDP only leads to a privacy guarantee for the aggregate data of all silos, but not for each individual silo’s data; and b) CDP allows the untrusted server and other silos to access silo $i$’s raw data and still satisfy the definition.
> Please let us know if we did not completely answer your question.
>
> >*The proposed algorithm (Algorithm 1) seems impractical. The Accelerated Noisy MB-SGD algorithm samples batches without replacement…empirical significance may be limited*
>
> We agree that sampling with replacement can be more practical in certain applications. For empirical risk minimization (Section 4), we propose with-replacement sampling (see **line 6 of Algorithm 1**). Also, in our numerical experiments, we use with-replacement sampling and show that our algorithm performs well empirically–even compared to *non-private* FedAvg/Local SGD. To clarify this in the revision, we added  “In practice, sampling with-replacement in line 6 of Algorithm 1 may be preferable. In this case, privacy noise should be calibrated as prescribed in the proof of Theorem 4.1” at the end of Section 3 (in violet).
>
> The main purpose of sampling without replacement for non-i.i.d. FL (Section 3) is to prove Theorem 3.1. Note that if we sampled with replacement, then stochastic gradients would no longer be independent across iterations, and heterogeneous data would prevent us from being able to obtain as tight a bound on the excess population risk as the bounds provided in Theorem 3.1.
>
> >*For the analysis of the non-i.i.d. setting, I do not understand how this paper quantifies the degree of non-i.i.d. In previous work [2], the non-i.i.d. degree is explicitly characterized…*
>
> An appealing feature of our bounds in Section 3 (for $M=N$) is that they are **immune to heterogeneity/non-i.i.d. data**. Thus, even for arbitrarily heterogeneous/non-i.i.d. silo data, we still get excess risk that nearly matches the optimal i.i.d. bound. In contrast, the convergence of FedAvg/Local SGD, which was studied in [2], necessarily suffers when silo data is heterogeneous. Indeed, the algorithm-specific lower bound in Woodworth et al. (NeurIPS 2020) proves that such dependence on heterogeneity is *necessary* for FedAvg.
>
> That being said, when $M < N$ silos are available to communicate in each round, the effect of silo heterogeneity kicks in for our algorithm, and the resulting excess risk bounds scale with a heterogeneity parameter $\upsilon^2 := \sup_{w \in \mathcal{W}} \frac{1}{N}\sum_{i=1}^N \||\nabla F_i(w) - \nabla F(w) \||^2$. This parameter is discussed in Appendix E.2 (Proof of the general $M \leq N$ version of Theorem 3.1). See also the paragraph immediately following the Algorithm 1 pseudocode (page 7).

---

> > ### Author Response · Authors · 2022-12-06
> > **Have we addressed your questions/concerns?**
> >
> > Thank you again for reading our paper and providing such great feedback. We know it is a busy time of the year for everyone. We were wondering if we've addressed your concerns with our responses. Please let us know if any questions remain.
> >
> > Thank you,
> >
> > Authors

---

### Official Review · Reviewer_aKXR · 2022-10-25

**Confidence:** 4
**Correctness:** 4
**Technical Novelty And Significance:** 3
**Empirical Novelty And Significance:** 3
**Recommendation:** 8

**Clarity, Quality, Novelty And Reproducibility:**

This paper is very well written and organized. The main body contains most information but some results are informal due to the abundance of results.

**Strength And Weaknesses:**

Strength:
1. The theoretical analysis of this paper is very thorough and informative, most cases with convex losses are considered and both upper bounds and lower bounds are derived.
2. The considered privacy notion is of high practical relevance.

Weakness:
1. It could help audience to appreciate the significance of this work better if the authors can highlight the implications of ISRL-DP, and maybe give some examples instead of pure mathematical languages to illustrate this notion.
2. It can be better if the authors can highlight the technical contributions of these analyses, like are there any technical challenges in obtaining these results comparing with other privacy notions, what make these results different from repeating the machinery of derivations in other DP notions.
3. For the secure shuffle part, can the authors provide some more discussions with existing works such as [1]?

[1] Cheu, A., Joseph, M., Mao, J., & Peng, B. (2021, September). Shuffle Private Stochastic Convex Optimization. In International Conference on Learning Representations.

**Summary Of The Paper:**

This paper proposes a notion called Inter-Silo Record-Level Differential Privacy that differentiates from the well studied central DP, client-level DP, and local DP.  Very detailed analyses for convex losses in various cases are conducted, and experiments are provided.

**Summary Of The Review:**

This paper conducts thorough analysis the privacy notion ISRL-DP in federated learning, the results are solid.

---

> ### Author Response · Authors · 2022-11-19
> **Response to Reviewer aKXR**
>
> Thank you for appreciating our contributions and for your great questions/feedback! We respond to your questions/suggestions below:
>
> >*...highlight the implications of ISRL-DP, and maybe give some examples instead of pure mathematical languages to illustrate this notion.*
>
> We revised the paper (particularly the violet part on page 2) to further clarify ISRL-DP. We have an example in Figure 1, and the differences between ISRL-DP and other DP notions are explained in the text and in Figure 2. We believe this revision should address your concern. We would be open to adding another section in the appendix to provide further clarification if you think it is needed.
>
>
> >*...highlight the technical contributions of these analyses…*
>
> Proving our lower bounds requires a significant amount of ingenuity and technical work. We explained this on page 6 (and we would be happy to move this discussion to the earlier parts of the paper if you find it necessary). Particularly, existing lower bound techniques for LDP and CDP ERM/SCO do not seem applicable to the more complex ISRL-DP FL problem. Thus, we take a completely different approach from these works, by using shuffling to relate ISRL-DP to CDP. The main technical contribution of our lower bound proof is **the first privacy amplification by shuffling bound for fully interactive FL algorithms and databases with any number of samples/silos** (see Theorem D.4). The proofs of Theorems D.4 and D.5 involve a non-trivial extension of the elegant “hiding among the clones” analysis of shuffling (Feldman et al., 2020) to ISRL-DP FL.
>
> For our upper bounds, our algorithmic contributions (especially our accelerated ISRL-DP algorithm for non-i.i.d FL) are arguably just as significant as our technical contributions. Some of our technical contributions include:
>
> -The tight uniform stability bound for strongly convex loss functions in Lemma D.2 (used to prove Theorem 2.1) is novel even for $N=1$, to the best of our knowledge. We added a comment on this in the revision on page 5.
>
> -Non-standard choice of stepsize in ISRL-DP MB-SGD enables us to remove the restriction on the smoothness parameter that appeared in Bassily et al. (2019) for the simpler CDP case with $N=1$; this is necessary for  extending our results to non-smooth loss via Nesterov smoothing. We mentioned this on page 5.
>
> -Lemma E.3 (used to prove the general version of Theorem 3.1) bounds the variance of stochastic gradients with $M < N$ heterogeneous silos using a careful combinatorial argument. This lemma is needed to obtain fast convergence rates for our algorithm when silo communications are unreliable. We mentioned this on page 7.
>
> >*For the secure shuffle part, can the authors provide some more discussions with existing works such as [1]?*
>
> Thank you for pointing this out. In the revision, we have expanded our discussion of [1]  in Appendix A (due to space constraints). Particularly, we mentioned “The main difference between our treatment of shuffle DP and [1] is that our results are much more general than [1]. For example,  [1] does not consider FL: instead they consider the simpler problem of stochastic convex optimization (SCO). SCO is a simple special case of FL in which each silo has only $n=1$ sample. Additionally, [1] only considers the i.i.d. case, but not the more challenging non-i.i.d. case. Further, [1] assumes perfect communication ($M=N$), while we also analyze the case when $M < N$ and some silos are unavailable in certain rounds (e.g. due to internet issues).  Note that our bounds in Theorem 5.1 recover the results in [1] in the special case considered in their work.”

---

> > ### Comment · Reviewer_aKXR · 2022-11-28
> > **Thank you for your rebuttal**
> >
> > I have read the revision and thanks for the effort to address my comments. The highlighted technical thrust help me better understand the contributions of this work. I vote to accept this work and have no other questions.

---

> > > ### Author Response · Authors · 2022-11-29
> > > **Thank you for your feedback**
> > >
> > > Thank you very much for your feedback and for helping us improve the quality of the manuscript. We are glad that you were satisfied with our response and revision. Thank you again for your time and effort in giving us such constructive feedback!

---

### Author Response · Authors · 2022-11-19
**Note to all reviewers**

Dear Reviewers,

Thank you all for your thoughtful reviews and detailed, constructive feedback. We have incorporated your suggestions in the rebuttal revision. Edits are highlighted in violet. We respond to each of your reviews individually below.

Sincerely,

Authors

---

### Decision · Program_Chairs · 2023-01-20

**Decision:**

Accept: poster

**Justification For Why Not Higher Score:**

It could've been an spotlight paper too. The results are solid. Though the results are not groundbreakingly new, surprising or interesting imho.

**Justification For Why Not Lower Score:**

I think it is more solid than most ICLR submissions and is clearly above the threshold for accept.

**Metareview: Summary, Strengths And Weaknesses:**

The paper studies differentially private federated learning in the multi-hospital collaboration setting where individual patients trust their hospital but not other collaborators nor the server federating the learning process.  This setting can be viewed in between the local DP setting and the central DP setting and is slightly different from the shuffle DP setting (though not much different technically). The results characterized the upper and lower bounds of learning with convex losses and upper bounds were obtained for the non-convex smooth cases. The paper clearly discussed the novelty in terms of the results and techniques and all reviewers vote for accept.

**Note From Pc:**

if the above contains the word "oral" or "spotlight" please see: "oral" presentation means -> notable-top-5% and "spotlight" means -> notable-top-25%. As stated in our emails, we are disassociating presentation type from AC recommendations